# Commensal production of a broad-spectrum and short-lived antimicrobial peptide polyene eliminates nasal *Staphylococcus aureus*

Benjamin O. Torres Salazar[1,2,3,10], Taulant Dema [4,10], Nadine A. Schilling[2,4], Daniela Janek[1,2,3], Jan Bornikoel [5], Anne Berscheid [3,5], Ahmed M. A. Elsherbini [1,2,3], Sophia Krauss[1,2,3], Simon J. Jaag [6], Michael Lämmerhofer [6], Min Li[7], Norah Alqahtani [8], Malcolm J. Horsburgh [8], Tilmann Weber [9], José Manuel Beltrán-Beleña [2,4], Heike Brötz-Oesterhelt [3,5], Stephanie Grond [2,4] ✉, Bernhard Krismer [1,2,3] ✉ & Andreas Peschel [1,2,3]

Antagonistic bacterial interactions often rely on antimicrobial bacteriocins, which attack only a narrow range of target bacteria. However, antimicrobials with broader activity may be advantageous. Here we identify an antimicrobial called epifadin, which is produced by nasal *Staphylococcus epidermidis* IVK83. It has an unprecedented architecture consisting of a non-ribosomally synthesized peptide, a polyketide component and a terminal modified amino acid moiety. Epifadin combines a wide antimicrobial target spectrum with a short life span of only a few hours. It is highly unstable under in vivo-like conditions, potentially as a means to limit collateral damage of bacterial mutualists. However, *Staphylococcus aureus* is eliminated by epifadin-producing *S. epidermidis* during co-cultivation in vitro and in vivo, indicating that epifadin-producing commensals could help prevent nasal *S. aureus* carriage. These insights into a microbiome-derived, previously unknown antimicrobial compound class suggest that limiting the half-life of an antimicrobial may help to balance its beneficial and detrimental activities.

The microbiomes of the human skin and upper airways play crucial roles in human health and predisposition to various diseases[1]. Microbiome compositions govern susceptibility to and severity of chronic diseases such as atopic dermatitis and acne, and microbiomes can include facultative bacterial pathogens such as *Staphylococcus aureus*, which colonizes the anterior nares of ca. 30% of the human population[2-4]. Nasal colonization is a major risk factor for severe and often fatal *S. aureus* infections, in particular when caused by difficult-to-treat methicillin-resistant *S. aureus*[5]. Microbiome dynamics are shaped by both antagonistic and mutualistic interactions between microbiome

[1]Department of Infection Biology, Interfaculty Institute of Microbiology and Infection Medicine, University of Tübingen, Tübingen, Germany. [2]Cluster of Excellence EXC 2124 Controlling Microbes to Fight Infections, Tübingen, Germany. [3]German Center for Infection Research (DZIF), partner site Tübingen, Tübingen, Germany. [4]Present address: Institute of Organic Chemistry, University of Tübingen, Tübingen, Germany. [5]Department of Microbial Bioactive Compounds, Interfaculty Institute of Microbiology and Infection Medicine, University of Tübingen, Tübingen, Germany. [6]Institute of Pharmaceutical Sciences, University of Tübingen, Tübingen, Germany. [7]Department of Laboratory Medicine, Renji Hospital, School of Medicine, Shanghai Jiaotong University, Shanghai, China. [8]Department of Infection Biology and Microbiomes, University of Liverpool, Liverpool, UK. [9]The Novo Nordisk Foundation Center for Biosustainability, Technical University of Denmark, Kongens Lyngby, Denmark. [10]These authors contributed equally: Benjamin O. Torres Salazar, Taulant Dema. ✉e-mail: stephanie.grond@uni-tuebingen.de; b.krismer@uni-tuebingen.de

members, but the elucidation of underlying molecular mechanisms has remained in its infancy[6].

*Staphylococcus epidermidis* is the most abundant member of the human skin and nasal microbiomes[7,8]. The mechanisms underlying its ecological success have remained largely unclear. *S. epidermidis* can modify its surface glycopolymers to alter its adhesive properties to different host surfaces[9]. It also produces phenol-soluble modulin peptides, which modulate inflammatory reactions in the skin and epithelia that seem to promote *S. epidermidis* persistence and may impair the fitness of major potential competitors such as *S. aureus, Corynebacterium* sp. or *Cutibacterium acnes*[10–12]. Moreover, *S. epidermidis* is a particularly frequent producer of bacteriocins, antibacterial peptides with highly variable structures and activity against potential target species[13–16]. Bacteriocins have traditionally been defined as ribosomally synthesized post-translationally modified peptides, but this term is increasingly used to encompass microbiome-derived antimicrobial small molecules including also non-ribosomally synthesized peptides (NRPs)[6].

Elimination of competitors by bacteriocin production is often beneficial but can also cause collateral damage to the producer when other bacteria within a functional mutualistic network are killed. Moreover, several bacteriocins can damage host cells, leading to inflammation and increased local antimicrobial defence[17,18]. Bacteriocin-mediated collateral damage could be limited, for instance, by bacteriocins with an extremely narrow and specific target range, such as most microcins, which would spare many important mutualists[19,20]. Another strategy relies on contact-dependent bacteriocins such as effectors secreted by type-V–VII secretion systems, which rely on specific interbacterial adhesion mechanisms, sparing mutualistic bacteria that do not physically bind to the bacteriocin producer[21,22]. Theoretically, bacteria could also produce bacteriocins with limited lifetimes, precluding their accumulation at a wider distance from the producer. The latter strategy would not rely on a high selectivity of the compound for specific competitors, which may be difficult to ensure in a dynamic microbiome context. However, it would ensure that the producer inhibits only bacteria in close proximity but maintains long-distance interaction networks and the integrity of host cells. Such a strategy can only work if community density is relatively low, and the microbiome composition and structure are sufficiently stable in time and space, which is usually the case in skin and anterior nares habitats[23].

In this article, we present a previously unknown antimicrobial small molecule, epifadin, produced by certain *S. epidermidis* strains, that combines an exceptionally wide target range with a very short half-life, thereby representing a previously unrecognized antimicrobial strategy. It is currently the only metabolite with antimicrobial activity produced by a common member of the human microbiota that combines moieties synthesized by non-ribosomal peptide synthetases (NRPSs) and polyketide synthases (PKSs). Epifadin allows *S. epidermidis* to eliminate its competitor *S. aureus* from their shared habitat, shown both in a lab-based setup and, more importantly, in vivo.

## Results

### *S. epidermidis* IVK83 produces an unknown antimicrobial

We previously reported that more than 95% of *S. epidermidis* isolates from the human nose produce antimicrobial molecules with activity against one or several other bacterial skin and nose microbiome members[13]. Whereas most inhibited only a limited number of test bacteria, isolate IVK83, isolated from a nasal swab of a healthy volunteer, was unique in its capacity to strongly inhibit most of the test strains, including representatives of Firmicutes, Actinobacteria and γ-Proteobacteria. A transposon mutant library of IVK83 was generated, and a mutant failing to inhibit the methicillin-resistant *S. aureus* strain USA300 and other test strains was identified. The transposon had disrupted an unknown biosynthetic gene cluster (BGC), composed of ten putative genes, which encompassed about 40 kb on a 55-kb plasmid (Fig. 1a), named pIVK83. The BGC encoded a set of putative biosynthetic

enzymes, namely, putative NRPS (EfiA), PKS (EfiB, EfiC and EfiD) and a combined PKS–NRPS (EfiE) along with an oxidoreductase (EfiO), a phosphopantetheinyl transferase (EfiP) and a thioesterase (EfiT). In addition, two genes encoded putative ABC transporter components, EfiF and EfiG (Fig. 1). The transposon had disrupted *efiA*, the first gene of the operon. Deleting *efiTP* in IVK83 abrogated antibacterial activity, which was restored by complementation with a plasmid-encoded *efiTP* copy, thereby confirming that the BGC is required for the capacity of IVK83 to inhibit other bacteria (Fig. 1b).

Whole-genome sequencing of various other inhibitory *S. epidermidis* strains from different strain collections identified four additional isolates, from Tübingen, Shanghai (China) and Liverpool (UK), with largely identical antimicrobial activities, plasmids and BGCs. They belonged to four different multi-locus sequence types (STs): ST575 (IVK83), ST549 and ST615 (Tübingen), and ST73 (Shanghai and Liverpool). The National Center for Biotechnology Information (NCBI) database contained further ten *S. epidermidis* isolates with an identical BGC, indicating that the ability to produce epifadin might be a sporadic but regularly found trait among *S. epidermidis* strains. Additionally, a nearly identical BGC, which differed only in the length of the initial NRPS gene, was present in five *Staphylococcus saccharolyticus* genomes[24] indicating that it is also an infrequent accessory genetic element in this species (Supplementary Fig. 1).

### Epifadin is a highly unstable antimicrobial small molecule

Culture filtrates of IVK83 displayed antibacterial activity, but initial typical isolation strategies via extraction and chromatography failed. The activity could be enriched by acidic precipitation with HCl (pH 2), drying under vacuum and dissolution in dimethyl sulfoxide (DMSO). However, within a short time, the antimicrobial activity of this extract decreased profoundly under standard laboratory conditions, which impeded further purification of the compound. However, the antibacterial activity remained largely constant when the DMSO extract was maintained under acidic conditions (pH 2), protected from oxygen and light and stored at −80 °C under argon atmosphere with addition of 0.05% of the oxidation inhibitor palmitoyl ascorbate (PA). After dilution of the extract in tryptic soy broth (TSB), loss of activity was observed under standard laboratory conditions (Fig. 2a). In contrast, other microbiome-derived antimicrobial peptides such as lugdunin from *Staphylococcus lugdunensis*[25], gallidermin from *Staphylococcus gallinarum*[26] or nisin A from *Lactococcus lactis*[27] did not lose activity under the same conditions within the 6-h observation period (Fig. 2a). Notably, at conditions simulating skin or nasopharyngeal habitats (pH 5.5, 30 °C or pH 7, 34 °C), respectively, and exposure to laboratory light, the antimicrobial activity quickly diminished and was completely lost within a few hours (Fig. 2b).

DMSO–PA extracts, obtained from acid-precipitated culture supernatants, were analysed by reversed-phase high-performance liquid chromatography with ultraviolet light detection (RP-HPLC-UV). Although the predicted sequences of the NRPS pointed to the presence of multiple peptide bonds, no obvious difference could be detected at the peptide bond-specific wavelength of 215 nm between the IVK83 wild type and Δ*efiTP* strain. However, the HPLC profile of the wild type displayed a major and some adjacent minor peaks at 383 nm, which were absent in that of the mutant, indicating a product possessing an expanded unsaturated system with double bonds (Fig. 3a–c). The antimicrobial activity corresponded well to the major 383-nm peak fraction. RP-HPLC coupled with high-resolution mass spectrometry (RP-HPLC–HR-MS) revealed a quasi-molecular ion ([M + H]$^+$, $m/z$ 964.4472) corresponding to $C_{51}H_{62}N_7O_{12}^+$ ($m/z$ 964.4451) and an elementary composition of $C_{51}H_{61}N_7O_{12}$ of the antibacterial agent (Fig. 4a). This formula was unknown in publicly available literature, indicating that IVK83 produces a yet unknown compound, which was named epifadin to reflect its origin from *S. epidermidis* and its rapidly fading activity at typical environmental conditions.

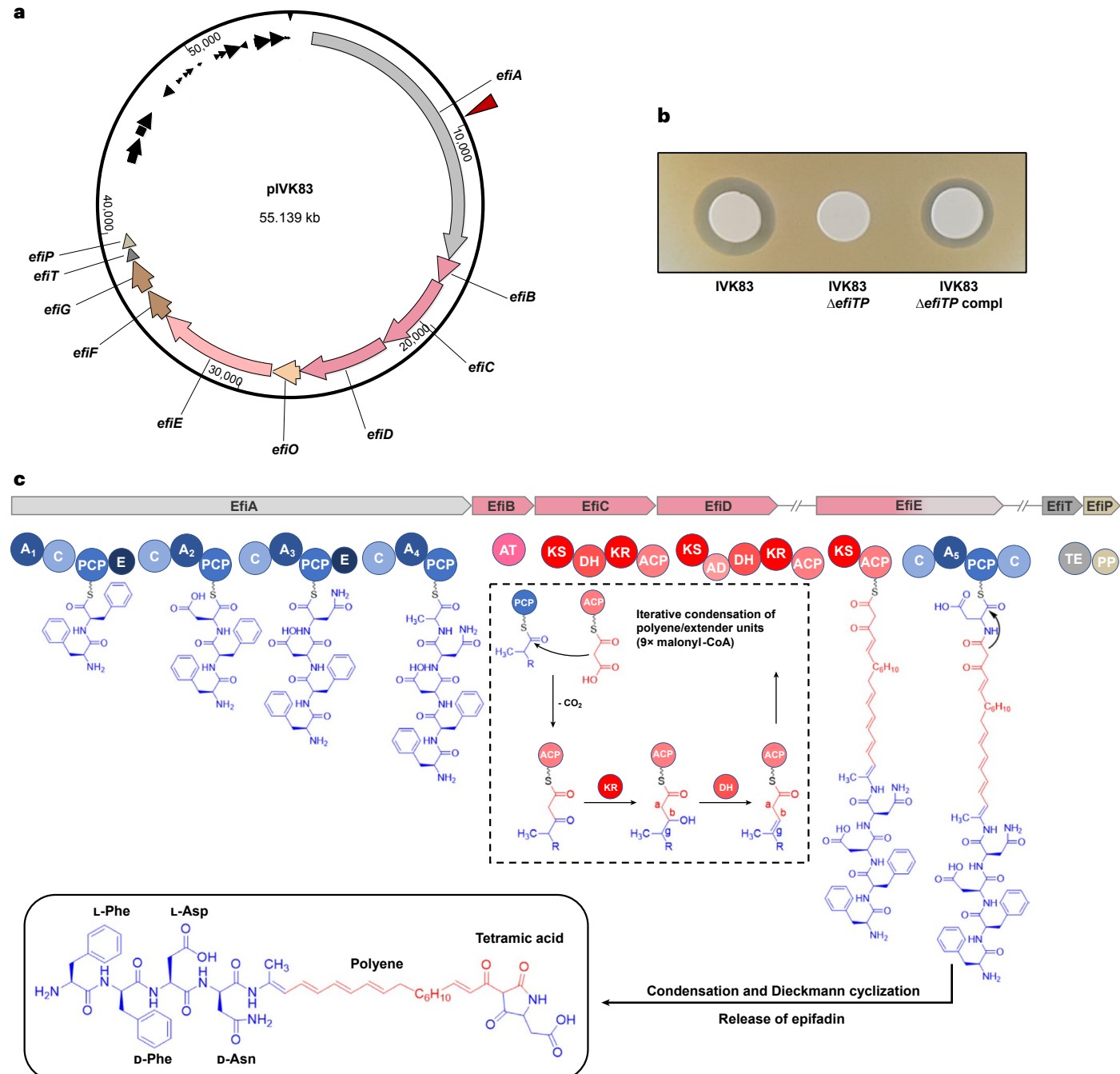

**Fig. 1 | Plasmid and epifadin BGC composition, biosynthetic modules and domain organization of the epifadin NRPS–PKS enzymes of IVK83. a**, Plasmid map of pIVK83 (~55 kbp) harbouring the epifadin operon (~40 kbp). The cluster consists of genes encoding a putative NRPS (*efiA*), three PKS (*efiB*, *efiC* and *efiD*), a hybrid PKS/NRPS (*efiE*), a putative NAD(P)- or FAD-dependent oxidoreductase (*efiO*) located between *efiD* and *efiE*, a thioesterase (*efiT*), a phosphopantetheinyl transferase (*efiP*) and two ABC transporter genes (*efiF* and *efiG*). Red arrow indicates the insertion site of Tn917, generating an epifadin-deficient mutant. **b**, Antimicrobial activity of IVK83 wild type (WT), isogenic Δ*efiTP* mutant and complemented strain towards *S. aureus* USA300. **c**, Domain organization of the NRPS and PKS EfiA, EfiB, EfiC, EfiD, EfiE, EfiT and EfiP involved in epifadin biosynthesis and proposed biosynthesis pathway, in which the first A-domain is responsible for the iterative integration of two phenylalanine residues in L- and D-conformation, respectively. Functional domains: A, adenylation; C, condensation; PCP, peptidyl carrier protein; E, epimerization; KS, ketosynthase; DH, dehydratase; KR, ketoreductase; AD, *trans*-AT docking; TE, thioesterase; PP, 4′-phosphopantetheinyl transferase. In the final structure of epifadin, the given tetramic acid additionally undergoes keto-tautomerization. NRPS and PKS modules and their domains as well as their products are depicted in blue and red, respectively.

## Epifadin is synthesized by a combination of NRPS and PKS enzymes

The epifadin BGC was analysed with antiSMASH 5.0 (bacterial settings)[28], which predicted the product of the cluster to have a three-partite composition with an N-terminal NRP part followed by a polyketide (PK) moiety and a C-terminal single amino acid residue (Fig. 1c). The order

of adenylation, condensation and epimerization domains in the NRPS enzyme EfiA suggested the biosynthesis of a pentapeptide sequence starting with aromatic amino acids, followed by aspartate, asparagine and an aliphatic amino acid (Supplementary Information) (Fig. 1c).

Epifadin was purified from more than 100 litre IVK83 culture supernatant as outlined above, immediately followed by preparative

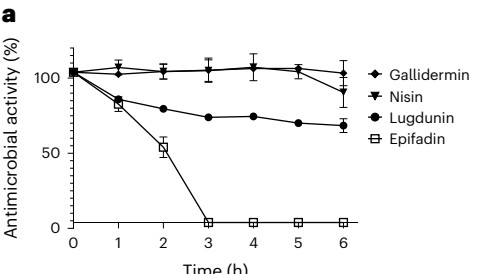

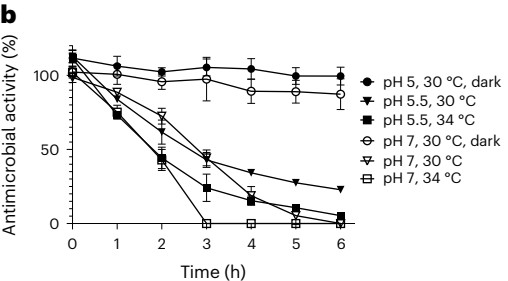

**Fig. 2 | High instability of epifadin under in vivo-like conditions.** *S. epidermidis* IVK83 culture supernatants containing epifadin were precipitated with HCl, freeze-dried and extracted with DMSO. This extract was diluted in TSB, incubated for the indicated time periods with constant shaking under the respective conditions and subsequently tested for activity towards *S. aureus* USA300 LAC. **a**, Stability of epifadin compared with other antibacterial agents under standard laboratory conditions (37 °C, pH 7, laboratory light exposure). **b**, In vivo-like conditions with laboratory light exposure at pH 7 (nasal condition) or pH 5.5 (skin condition) and reduced temperatures; incubation of epifadin under exclusion from light enabled further purification and activity determination. Conditions at pH 7, 37 °C and 0 h were used as reference for 100% stability. Data shown represent the mean ± s.d. of three independent experiments.

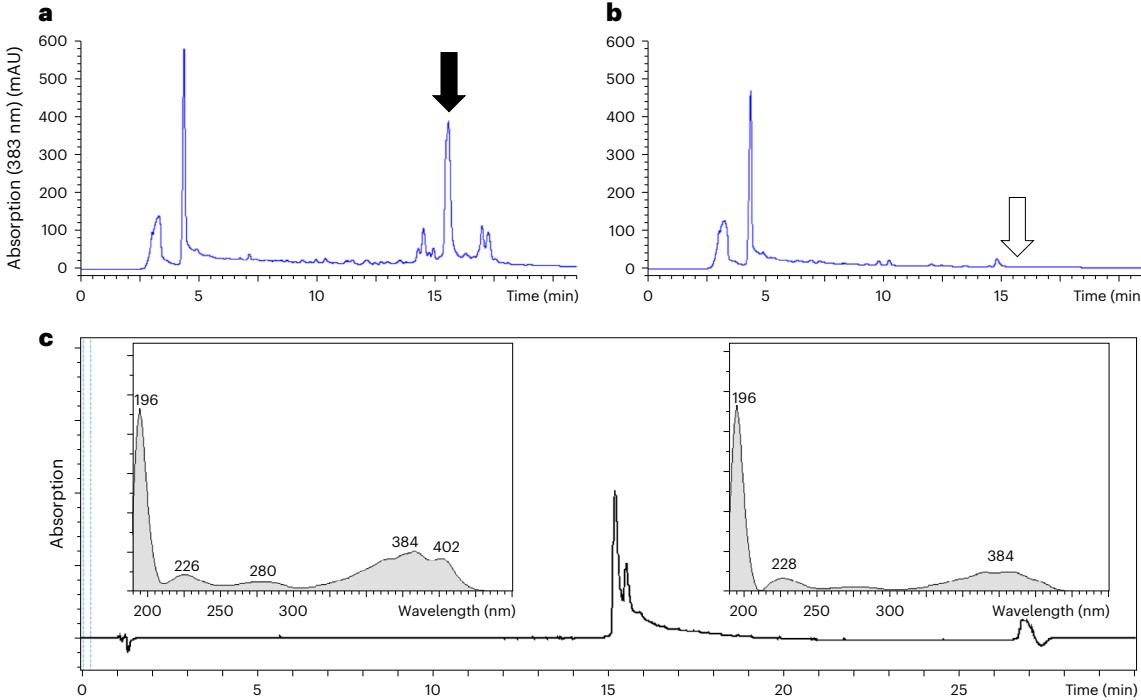

**Fig. 3 | Detection, isolation and absorption spectrum of epifadin.** Chromatogram and absorption spectrum of epifadin on RP-HPLC column. **a,b**, HPLC-UV chromatograms ($\lambda$ = 383 nm) of DMSO–PA extract of IVK83 wild type (**a**) and $\Delta efiTP$ mutant (**b**). A prominent peak (black arrow) of epifadin is only present in the wild-type sample with a retention time of ~15 min at 383 nm.

**c**, UV chromatogram (383 nm) of the epifadin peak is shown. UV spectra of the active compound at 15.2 min and 15.6 min from 190 nm to 450 nm are shown (methanol as a solvent causes a strong absorbance at 200 nm). The absorption maxima of the epifadin peak are indicated (peptide bond, 210–230 nm; phenylalanine, 280 nm; and polyene moiety, 330–410 nm).

RP-HPLC in the dark and collecting the eluate at −77 °C under inert argon gas, finally yielding 6 mg of the transiently intact pure compound. RP-HPLC–HR-MS confirmed the fast degradation of epifadin under regular laboratory conditions within a few hours. A peptide amide Phe–Phe–Asp–Asn–NH$_2$ with the neutral sum formula C$_{26}$H$_{32}$N$_6$O$_7$ ([M + H]$^+$, $m/z$ 541.2422, in accordance with C$_{26}$H$_{33}$N$_6$O$_7^+$ ($m/z$ 541.2405)) represented a dominant decomposition fragment, probably resulting from spontaneous, distinct chemical decomposition at the originally fifth amino acid (Extended Data Fig. 1a). Since this fragment was not accessible to common Marfey's analysis for the assignment of the D-/L-configuration of the phenylalanine moieties, we synthesized L-Phe–D-Phe–L-Asp–D-Asn–NH$_2$ (FfDn–NH$_2$). Synthetic FfDn–NH$_2$ showed identical physicochemical properties in RP-HPLC-UV and tandem mass spectrometry (MS/MS) analysis compared with the natural tetrapeptide amide fragment of epifadin (Extended Data Fig. 1). The L-D-L-D-amino acid configuration of the peptide amide

was also supported by antiSMASH prediction. Chemical synthesis of the tetrapeptide FfDn–NH$_2$ and its enantiomer with swapped L-/D-stereochemistry (fFdN–NH$_2$) and chiral HPLC–MS analysis unambiguously confirmed the L-D-L-D stereochemistry (Supplementary Fig. 2). Nuclear magnetic resonance (NMR) spectroscopy of the pure tetrapeptide amide confirmed the sequence, which lacked antimicrobial activity (Supplementary Information, Extended Data Fig. 2 and Supplementary Table 1). Additionally, multidimensional NMR experiments of purified, intact epifadin provided evidence that the fifth amino acid is a modified alanine (Extended Data Figs. 3 and 4a) with a carbon double bond (−NH−C(CH$_3$)=C) instead of the C-terminal carbonyl group (−NH−CH(CH$_3$)−C=O).

Two-dimensional NMR (Extended Data Fig. 3) revealed the direct linkage of the decarbonylated pentapeptide to a polyene structure, with a tetraene at C$_\alpha$ of the original alanine unit, linked to a saturated methylene unit, overall building an octaketide bridge (C$_{16}$H$_{20}$). This

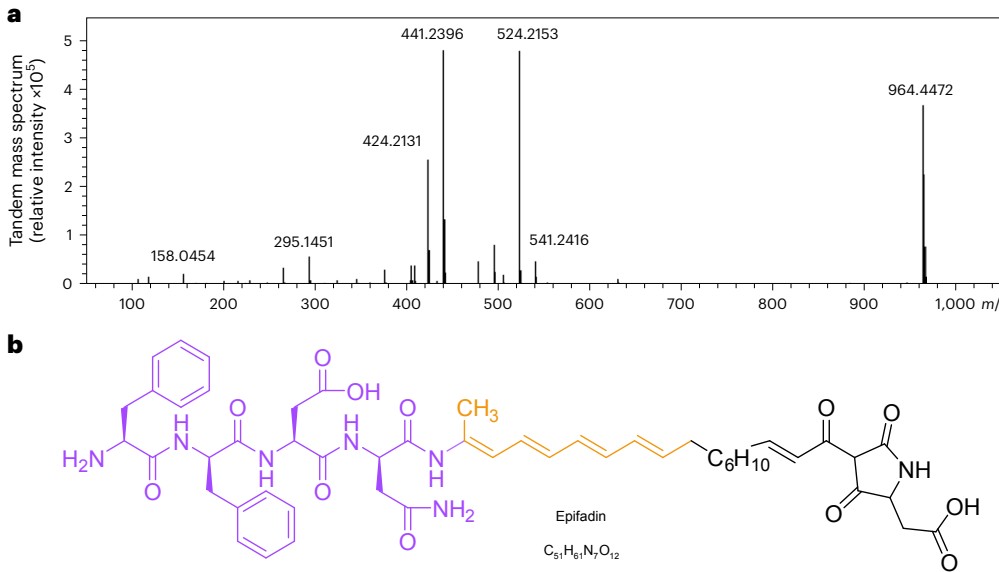

**Fig. 4 | Molecular mass and structure of epifadin. a**, MS/MS spectrum of intact epifadin (HR-ESI(+) TOF MS) indicates a mass of intact epifadin of 964.4472 Da. **b**, Purple moiety elucidated by NMR and MS, orange moiety by NMR and tetramic acid moiety (black, inferred from genetic, 1D-NMR, 2D-NMR and HR-MS data in accordance with detailed MS mechanistic considerations; the structure is not depicted in its keto-tautomeric form).

polyene is presumably formed by the *trans*-acetyltransferase (AT) type PKS modules of EfiC, EfiD and EfiE (Fig. 1c and Extended Data Fig. 4a). Detailed structure elucidation of the polyene part was achieved by sets of two-dimensional NMR yielding additional spatial information, despite limitations due to the instability of epifadin.

The NRPS domain of EfiE was predicted to link an aspartate as the sixth amino acid to the hydrophobic PK chain of epifadin and was suggested to allow for further modification, a terminal cyclization (Fig. 1c). HR-MS (MS/MS) analyses of epifadin combined with the predicted enzymatic properties of EfiE underline the Dieckmann-like cyclization of the terminal amino acid to form a tetramic acid (Supplementary Information). MS analyses with MS/MS fragmentation yielded a comprehensive mass signal pattern, which we assigned to distinct fragment ions. We state that these ion structures originate from a five-membered tetramate heterocycle (for example, $[M + H]^+$, *m/z* 158.0454, assigned to $C_6H_8NO_4^+$ (*m/z* 158.0448); $[M + H]^+$, *m/z* 140.0348, for a $C_6H_6NO_3^+$ moiety (*m/z* 140.0342) (Fig. 4a, Extended Data Fig. 5 and Supplementary Table 2). Typical for tetramic acids, 2D-NMR spectroscopy yielded solid key correlations for characteristic tetramate signals in epifadin. However, some distinct signals of the tetramic heterocycle were not detected, probably as a consequence of the unique assembly of the amphiphilic tetrapeptide-polyene-(Asp)tetramate (Supplementary Figs. 3 and 4). Chemical derivatization attempts did not provide stable epifadin tautomers (Supplementary Figs. 5–7). The tetramic acid moiety is further supported by the predicted adenylation domain specificity of $A_5$ (Asp) and the presence of a particular C-terminal condensation-like ($C_T$) domain in EfiE, which is involved in cyclization reactions[29]. Such a domain has been proposed to be responsible for the tetramic acid formation in the bacterial antibiotic malonomycin[30] and in fungal PKS–NRPS-derived products such as cAATrp[31].

Remarkably, none of the epifadin PKS modules contains a cognate AT domain. The single, discrete acyltransferase (EfiB) presumably loads the extender units to the acylcarrier protein (ACP) domains of EfiCDE. Since only three PKS modules are responsible for incorporation of nine malonyl-CoA building blocks, epifadin biosynthesis seems to involve an iterative *trans*-AT PKS system for the generation of the polyene moiety. To the best of our knowledge, such an iterative *trans*-AT PKS system has been described only once, for biosynthesis of the macrocyclic chejuenolide[32].

The provided results represent strong evidence for the given epifadin structure (Figs. 1c and 4b), which is derived from a combination of HPLC-UV coupled with HR-MS, yielding sum formulae, fragmentation patterns and UV spectrum (Figs. 3 and 4, Extended Data Figs. 1, 5, 6 and 9 and Supplementary Table 2), from multidimensional NMR analyses of full epifadin and synthetic epifadin fragments with stereo structure elucidations (Extended Data Figs. 2, 3 and 4a and Supplementary Figs. 3 and 4) and from sequence-based prediction of EfiA-P enzyme reactions (Fig. 1c and Supplementary Information). In conclusion, we demonstrate that epifadin is currently the only NRP–PK antimicrobial isolated from the human microbiome and is the founding member of a previously undescribed structural class of peptide–polyene–tetramic acids.

### Bactericidal epifadin perturbs membrane integrity

A large panel of microorganisms from human nasal microbiomes was analysed for susceptibility to epifadin by monitoring inhibition zones around the epifadin producer IVK83. Most of the tested Firmicutes (several *Staphylococcus* species and *Streptococcus pyogenes*) and Actinobacteria (several *Corynebacterium* species, *Cutibacterium acnes*, *Micrococcus luteus* and *Kocuria* sp.) were susceptible albeit with some intra-species variability (Fig. 5). Notably, all tested *S. aureus* strains were susceptible to epifadin, whereas 9 of 16 tested nasal *S. epidermidis* isolates were resistant. Most of the tested γ-Proteobacteria were resistant, but *Raoultella ornithinolytica* was inhibited. Since the antifungal agents amphotericin A, a natural derivative of amphotericin B, and nystatin $A_1$ contain polyene moieties similar to that of epifadin[33], *S. epidermidis* IVK83 was also tested for its impact on *Candida albicans* and *Saccharomyces cerevisiae*. Both fungal species were inhibited by the wild-type strain (Fig. 5) but not by the isogenic mutant Δ*efiTP*. Thus, epifadin has a very broad activity spectrum including Gram-positive and some Gram-negative bacteria and yeasts.

Antimicrobial activity was also tested against selected bacterial isolates using the limited amounts of purified, non-decomposed epifadin under dark conditions to sustain the stability of epifadin. It turned out to be a potent inhibitor of *S. aureus* with inhibitory concentration (IC) values between 0.9 and 1.5 µg ml⁻¹ (Supplementary Table 3). Interestingly, among all the tested *Staphylococcus* species, epifadin was most active against *S. aureus* and had substantially higher IC values (between 3.7 and 8.6 µg ml⁻¹) even for susceptible strains of

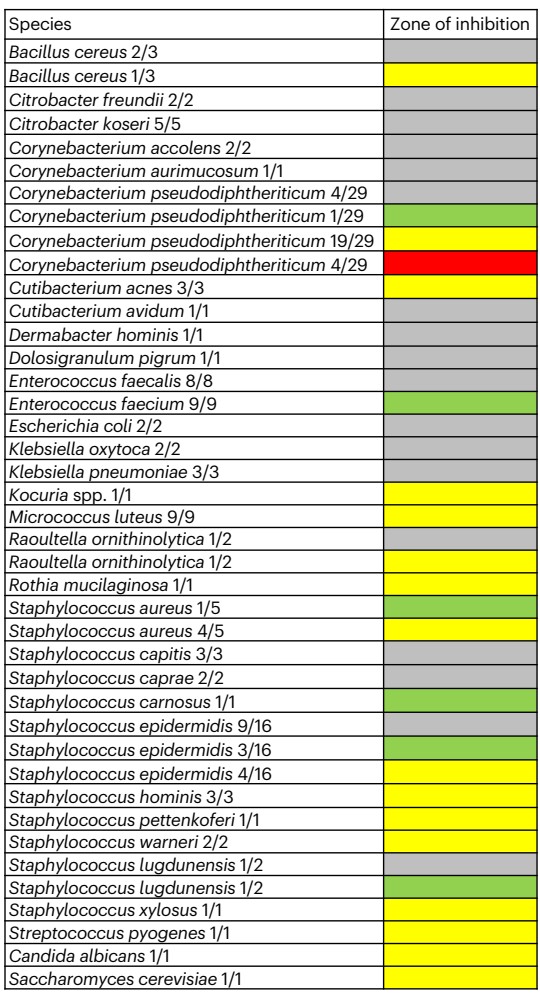

| Species | Zone of inhibition |
|---|---|
| *Bacillus cereus* 2/3 | grey |
| *Bacillus cereus* 1/3 | yellow |
| *Citrobacter freundii* 2/2 | grey |
| *Citrobacter koseri* 5/5 | grey |
| *Corynebacterium accolens* 2/2 | grey |
| *Corynebacterium aurimucosum* 1/1 | grey |
| *Corynebacterium pseudodiphtheriticum* 4/29 | grey |
| *Corynebacterium pseudodiphtheriticum* 1/29 | green |
| *Corynebacterium pseudodiphtheriticum* 19/29 | yellow |
| *Corynebacterium pseudodiphtheriticum* 4/29 | red |
| *Cutibacterium acnes* 3/3 | yellow |
| *Cutibacterium avidum* 1/1 | grey |
| *Dermabacter hominis* 1/1 | grey |
| *Dolosigranulum pigrum* 1/1 | grey |
| *Enterococcus faecalis* 8/8 | grey |
| *Enterococcus faecium* 9/9 | green |
| *Escherichia coli* 2/2 | grey |
| *Klebsiella oxytoca* 2/2 | grey |
| *Klebsiella pneumoniae* 3/3 | grey |
| *Kocuria* spp. 1/1 | yellow |
| *Micrococcus luteus* 9/9 | grey |
| *Raoultella ornithinolytica* 1/2 | grey |
| *Raoultella ornithinolytica* 1/2 | yellow |
| *Rothia mucilaginosa* 1/1 | yellow |
| *Staphylococcus aureus* 1/5 | green |
| *Staphylococcus aureus* 4/5 | yellow |
| *Staphylococcus capitis* 3/3 | grey |
| *Staphylococcus caprae* 2/2 | grey |
| *Staphylococcus carnosus* 1/1 | yellow |
| *Staphylococcus epidermidis* 9/16 | grey |
| *Staphylococcus epidermidis* 3/16 | green |
| *Staphylococcus epidermidis* 4/16 | yellow |
| *Staphylococcus hominis* 3/3 | yellow |
| *Staphylococcus pettenkoferi* 1/1 | yellow |
| *Staphylococcus warneri* 2/2 | yellow |
| *Staphylococcus lugdunensis* 1/2 | yellow |
| *Staphylococcus lugdunensis* 1/2 | green |
| *Staphylococcus xylosus* 1/1 | yellow |
| *Streptococcus pyogenes* 1/1 | yellow |
| *Candida albicans* 1/1 | yellow |
| *Saccharomyces cerevisiae* 1/1 | yellow |

Legend:
- No inhibition (grey)
- <1 mm (green)
- 1–3 mm (yellow)
- 3–5 mm (red)

**Fig. 5 | Broad antimicrobial activity of epifadin-producing *S. epidermidis* IVK83.** Sizes of inhibition zones caused by IVK83 on lawns of test strains listed on the left are given in colour code: grey (no inhibition), green (below 1 mm), yellow (1–3 mm) or red (3–5 mm). The numbers of isolates with a specific degree of susceptibility to epifadin among all tested isolates are given behind species names. The IVK Δ*efiTP* mutant mediated no inhibition.

*S. epidermidis*, *Staphylococcus hominis*, *Staphylococcus sciuri* (now *Mammaliicoccus sciuri*) and *Staphylococcus warneri*. Purified epifadin was bactericidal—a tenfold IC reduced the number of viable *S. aureus* cells by three orders of magnitude within 4 h of incubation (Extended Data Fig. 7a). In contrast, epifadin impaired human HeLa cells only at an IC ca. 20-fold higher than that for *S. aureus* (Extended Data Fig. 7b), suggesting minor effects on viability of human cells.

After exposure of *S. aureus* USA300 JE2 to epifadin extract, time-lapse microscopy showed Sytox Green penetration of the membrane within minutes, demonstrating a loss of membrane integrity, which was quickly followed by cell lysis (Extended Data Fig. 10a and Supplementary Videos 1 and 2). In contrast, cells exposed to a control extract from IVK83 Δ*efiTP* continued to divide without signs of impairment. In agreement with this notion, the epifadin-containing extract led to a substantial and dose-dependent breakdown of the membrane potential in *S. aureus* in a similar fashion as the protonophore and oxidative phosphorylation uncoupler carbonyl cyanide-*m*-chlorophenylhydrazone (CCCP) indicating an immediate impact on cytoplasmic membrane integrity (Extended Data Fig. 10b).

To gain more insights into the mechanisms governing the susceptibility or tolerance to epifadin, the producer *S. epidermidis* IVK83 was co-cultivated with *S. aureus* USA300 JE2 in daily subcultures for 3 days followed by isolation and analysis of viable *S. aureus* isolates. Several of these isolates had developed resistance to inhibition by IVK83 (Supplementary Fig. 8). Comparison of the genome sequences of five resistant isolates with that of the parental strain revealed point mutations in several virtually unrelated genes (Supplementary Table 4). One gene, *desK*, encoding the histidine kinase of the DesKR two-component regulatory system (TCS) (also described as TCS7)[34] was mutated in all five epifadin-resistant isolates. Two different missense mutations led to distinct amino acid changes (A162V, AV95DG; Supplementary Fig. 8 and Supplementary Table 4), suggesting an altered activity of the DesKR TCS, which controls a largely uncharacterized regulon encompassing about 14 genes with mostly unknown functions[34]. Collectively, these data indicate that epifadin impairs bacterial membrane integrity, an effect that can be alleviated by dysregulation of the *desKR* regulon in a currently unclear fashion.

## *S. aureus* is eliminated during co-cultivation

The particularly strong, bactericidal activity of epifadin against *S. aureus* raised the question of whether epifadin can help *S. epidermidis* to outcompete *S. aureus*. The mutant *S. epidermidis* Δ*efiTP* was overgrown by *S. aureus* during co-cultivation on agar plates or in liquid broth within 24 h (Fig. 6a,b and Extended Data Fig. 8), which is in agreement with the previously documented capacity of *S. aureus* to grow faster or utilize nutrients better than *S. epidermidis*[35]. In contrast, the IVK83 wild type eradicated *S. aureus* within 24 h, indicating that epifadin was effective enough under the used conditions to confer a considerable fitness advantage. *S. aureus* had almost completely vanished after 24 h, underlining the bactericidal activity of epifadin. Complementation of the mutant with a plasmid-encoded copy of *efiTP* restored the *S. aureus*-eradicating ability of IVK83 Δ*efiTP*, confirming that it is indeed epifadin that allowed *S. epidermidis* to outcompete *S. aureus* (Fig. 6c and Extended Data Fig. 8).

The ability of epifadin to promote the competitive capacity of *S. epidermidis* in vivo was analysed in the cotton rat-based model of nasal colonization[36]. The IVK83 wild type and Δ*efiTP* strains were similarly proficient in nasal colonization (Fig. 6d). When either the IVK83 wild type or the *efiTP* mutant was nasally instilled together with equal numbers of *S. aureus* Newman, the median number of viable *S. aureus* cells was significantly reduced in the IVK83 wild type-colonized animals compared with the Δ*efiTP*-colonized animals after 5 days (Fig. 6e), which indicates that epifadin-producing *S. epidermidis* can effectively interfere with *S. aureus* nasal colonization. From a subgroup of animals colonized with *S. aureus* and IVK83 wild type, higher *S. aureus*-to-*S. epidermidis* ratios were recovered compared with the rest of the group, albeit still at lower average ratios compared with animals colonized with *S. aureus* and IVK83 Δ*efiTP*. We confirmed that this behaviour was not caused by spontaneous development of epifadin resistance. It may result from different capacities of bacteria to colonize individual noses of the non-inbred and non-gnotobiotic cotton rats.

## Discussion

Recent metagenome- and cultivation-dependent screening approaches have revealed an increasing number of BGCs for complex natural products among bacteria from microbiomes[37,38]. These include several NRP compounds such as lugdunin[25] and PK compounds such as wexrubicin[38]. We here report epifadin, currently the only example of an antimicrobial peptide–polyene consisting of both NRP and PK moieties produced by a common member of the human microbiome. Epifadin is the founding member of a previously undescribed compound class. Previously, hybrid NRP–PK molecules have been associated predominantly with environmental or soil bacteria. Our study further supports the notion that the antibacterial compounds produced by environmental and host-associated bacterial communities are based on equally unique and complex chemical architectures[39].

Epifadin is an amphiphilic, charged compound with an unprecedented alternation of a polar peptide, a non-polar polyene and a polar

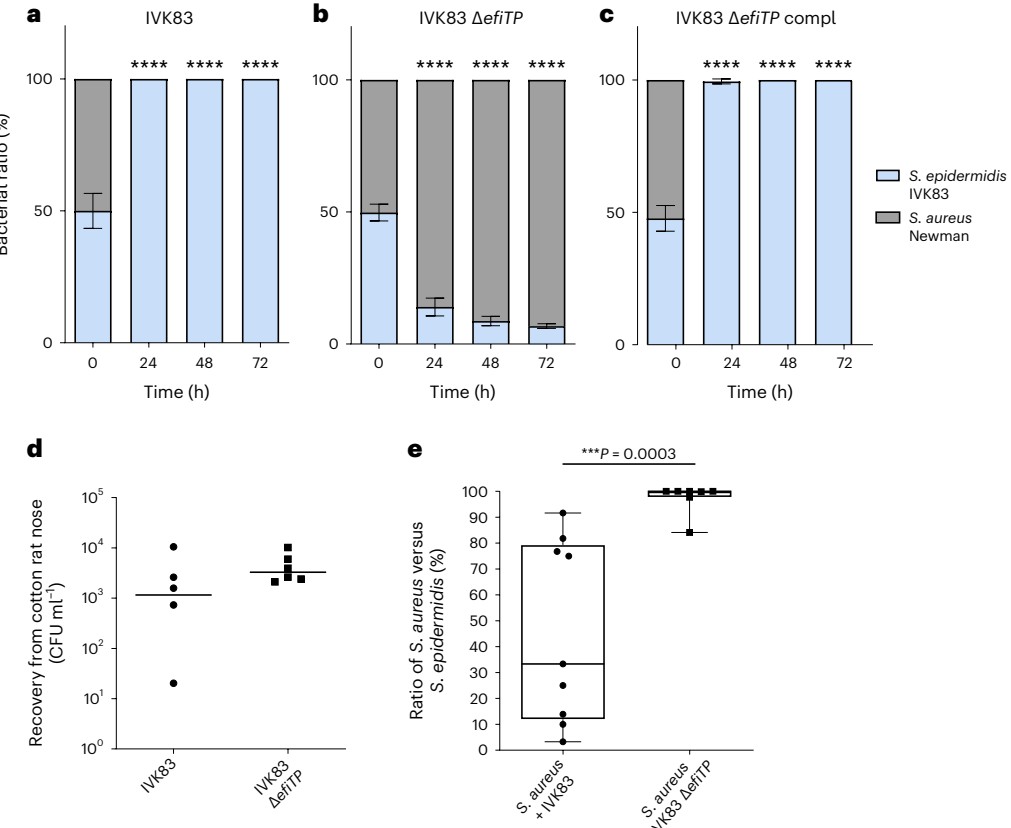

**Fig. 6 | Epifadin-producing *S. epidermidis* IVK83 restricts *S. aureus* growth in vitro and in vivo in cotton rats. a–c**, In vitro competition assays on TSA: *S. aureus* growth is inhibited by IVK83 wild type (WT) (grey or light blue bars, respectively) after 24 h of incubation on TSA (**a**); in contrast, the mutant IVK83 Δ*efiTP* is overgrown by *S. aureus* over time (**b**); complementation (pRB474Compl) restored the wild-type phenotype (**c**). Data points represent mean values ± s.d. of three independent experiments. Significant differences between starting condition and indicated timepoints were analysed by one-way analysis of variance (****$P$ < 0.0001). **d**, Nasal colonization capability of IVK83 wild type and IVK83 Δ*efiTP* in cotton rats. Bacterial CFU found in cotton rat noses 5 days after instillation are shown. Horizontal lines represent the median of each group. Data shown represent the median of six individually colonized animals per bacterial strain. **e**, Epifadin-producing IVK83 reduces *S. aureus* carriage in cotton rat noses. Percentage of *S. aureus* cells from cotton rat noses was significantly lower when *S. aureus* was co-colonized with IVK83 wild type compared with the Δ*efiTP* mutant 5 days after instillation. Centre lines in the boxplots represent median values of nine and seven individually co-colonized animals, respectively; box edges, 25th and 75th percentiles; whiskers, 1.5× the interquartile range. Significant differences were calculated by the Mann–Whitney test (***$P$ ≤ 0.001).

amino acid-derived building block. We will focus on a chemical total synthesis in the future to study structural details of the $C_6H_{10}$ bridge in the polyene moiety, which remains challenging due to the extraordinary instability of epifadin. Structure–activity relationship studies with synthetic epifadin analogues aim towards higher chemical stability with concurrent antimicrobial activity. While the full epifadin molecule may remain refractory to NMR analysis due to the labile enamide moiety, chemical total synthesis of epifadin analogues and comparison of chromatographic, spectroscopic and antimicrobial properties with those of native epifadin may confirm structural details. However, this approach will require the development of new strategies for the chemical synthesis of the unstable enamide polyene tetramic acid in epifadin. The recent elucidation of the structure of a fungal propargyl tetramic acid natural product on the sole basis of chemical synthesis exemplifies the feasibility of such an approach[40].

Although the exact antimicrobial mechanism remains to be elucidated, we provide evidence that epifadin compromises the integrity of the cytoplasmic membrane, a mechanism that could explain why epifadin also inhibits fungal cells. It remains unclear whether epifadin needs to bind to a specific molecular target to exert its antimicrobial activity. We demonstrate that mutations in the regulatory sensor kinase DesK cause tolerance to epifadin in *S. aureus*, presumably by altering the expression of either a target molecule or a resistance-conferring

factor. Elucidating the functions of the various members of the DesKR regulon will help to obtain better insights in the cellular interaction partners of epifadin. Future in-depth studies on the mode of action will also help to understand why HeLa cells appeared insensitive to epifadin and whether other human cell types may be more susceptible.

To our knowledge, the combination of peptide–polyene–tetramic acid moieties is unique to epifadin. Recently, isolates of *S. mutans* have been shown to produce reutericyclin A (ref. 41), an antimicrobial acyl tetramic acid with proton-ionophore activity[42], suggesting that the tetramic acid moiety could represent an antimicrobial 'war head'. Since the reutericyclin A derivative mutanocyclin, lacking the acyl chain, is not antimicrobial[41], linkage of a tetramic acid to a membrane-associating unpolar carbon chain may be required for the antimicrobial activity. In contrast to the ionophore activity of reutericyclin[42], epifadin completely disrupts *S. aureus* cells, leading to cell lysis. This difference clearly indicates an alternative or additional mode of action for epifadin, which will be studied in more detail in the future. Additionally, some isolates of *S. mutans* can produce the polyene peptide mutanofactin, which plays a role in biofilm formation by these strains[43]. From our genetic analyses, we postulate that specific *Streptococcus mutans*, *L. lactis* and various *Bacillus* isolates potentially produce epifadin-related peptide–polyene–tetramic acid compounds (Supplementary Fig. 1).

The extraordinary instability of epifadin may cause a higher impact on cells that take the compound up quickly, while slow uptake may spare cells from destruction. The types of immunity (or self-protection) proteins encoded in a BGC can help to propose the mode of action of a new antibiotic. However, the epifadin BGC encodes only the two ABC transporter components EfiFG as potential resistance proteins, which may act as drug exporters and do not shed light on a specific antibacterial target.

Previous approaches for the isolation of new microbiome-derived bacteriocins and related compounds have focused on stable compounds, which are comparably easy to purify and to characterize[6]. Short-lived antimicrobial compounds may have been overlooked in many of these studies, and purification attempts may have remained unsuccessful. The *Bacillus* antimicrobial bacillaene, for instance, is also highly unstable because of its polyene-enamide structure[44]. It might initially seem paradoxical for a bacterium to waste energy by producing an antimicrobial with such a short life span. However, limited stability may help to minimize collateral damage of bacterial mutualists and host cells. Mounting evidence indicates that bacterial interactions in complex communities are shaped not only by antagonistic but also by multifactorial mutualistic interdependencies[45]. Bacterial life on the mammalian skin and in the nares is challenging because nutrient supply is poor and many of the important cofactors and building blocks need to be released from larger polymers or synthesized de novo[35,46]. These constraints require that bacteria use antagonistic mechanisms with great care. Other strategies than using short-lived antimicrobials may be equally costly but even less effective. For instance, bacteriocins with a very narrow target spectrum may work only in a community with defined and stable composition, while its transfer to other, even slightly different microbiomes may easily lead to loss of a competitive advantage. Likewise, contact-dependent bacteriocin delivery requires a huge and complex secretion machinery, which are also costly to synthesize and maintain[47]. A short-lived but broad-spectrum antimicrobial such as epifadin may be much more versatile and advantageous in microbiomes with limited physical proximity between antimicrobial producer and potential target bacteria. Such low bacterial densities and sufficient spatial separation of individual bacterial microcolonies can be found, for instance, on the human skin and upper airways. It should be noted that the human antimicrobial host defence also relies on short-lived compounds, reactive oxygen and nitrogen derivatives, with broad toxicity to microbial and even host cells[48]. If produced at the right place and the right time, the beneficial consequences of such compounds outweigh the detrimental effects. It remains to be analysed under which conditions epifadin is produced by *S. epidermidis* as the BGC does not encode a potential regulator (Fig. 1a).

The epifadin BGC is located on a conjugative plasmid, which probably disseminates horizontally because it was found in different *S. epidermidis* clonal lineages and in *S. saccharolyticus* isolates[24]. The observation that epifadin is produced by five *S. epidermidis* isolates from three different geographical regions, and that the NCBI database contains ten further isolates, underscores its evolutionary success and fitness benefit. Related gene clusters have also been found in *L. lactis*[49] and *S. mutans*[50] isolates (Supplementary Fig. 1). Whereas the *S. epidermidis* or *S. saccharolyticus* BGCs could not be identified in the metagenomes of human oral and aerodigestive tract (eHOMD; http://www.ehomd.org), the closely related *L. lactis* and *S. mutans* BGCs could be identified in this database. However, antimicrobial activities have not been reported for these strains. Moreover, these two BGCs lack *efiO* and the gene order is different from that of the epifadin BGC, suggesting that the product is not identical to epifadin. Moreover, several *Bacillus amyloliquefaciens*, *Bacillus velezensis* and *Bacillus thuringiensis* genomes from databases contain BGCs with high overall organizational similarity to the epifadin BGC, suggesting that numerous isolates from these species might produce similar compounds, which may also belong to the here described NRPS–polyene–tetramic acid class of secondary metabolites (Supplementary Fig. 1). A few *Paenibacillus*,

*Clostridium* and *Lachnotalea* genomes also contain BGCs with moderate similarity to the epifadin BGC.

Epifadin-producing *S. epidermidis* IVK83 had a strong capacity to eliminate the opportunistic pathogen *S. aureus* in experimental nasal colonization studies similar to the recently reported lugdunin-producing *S. lugdunensis* strain IVK28 (ref. 25). Whereas the highly stable lugdunin acts as a protonophore leading to the breakdown of the membrane potential and subsequent slow killing of bacterial cells, the unstable epifadin seems to instantly disrupt the membrane of *S. aureus* leading to fast cell lysis, highlighting two different strategies to combat *S. aureus* in the human microbiome. Although epifadin is too unstable in its native form to be considered as a future drug, epifadin-producing commensal bacteria could be used as probiotics that would eliminate *S. aureus* from the nares of at-risk patients. It remains to be clarified how unstable epifadin is in an in vivo setting where mucus or mucosal components could influence its stability. MS-based imaging combined with organoid-based microbiome models, mimicking in vivo physiology, could help clarify this question. The search for fugacious bacteriocins should be extended to other microbes to identify further previously overlooked, short-lived antimicrobials, which could hold promise for the eradication of facultative pathogens in human microbiomes.

## Methods

### Data reporting
The animal experiments were not randomized, and the investigators were not blinded to allocation during experiments and outcome assessment. No statistical method was used to pre-determine sample size, but our sample sizes are similar to those reported in previous publications[25].

### Bacterial strains and growth conditions
The bacterial strains used in this study are listed in Supplementary Table 5. TSB and tryptic soy agar (TSA) or basic medium (BM: 1% soy peptone, 0.5% yeast extract, 0.5% NaCl, 0.1% glucose and 0.1% $K_2HPO_4$, pH 7.2) and BM agar (BM with 1.5% agar) were used for bacterial cultivation. For corynebacteria, 5% sheep blood (Oxoid) was added. Cutibacteria were incubated under anaerobic conditions using an anaerobic jar including an AnaeroGen bag (Thermo Fisher Scientific). When appropriate, antibiotics were used at concentrations of 250 µg ml$^{-1}$ for streptomycin, 10 µg ml$^{-1}$ chloramphenicol, 100 µg ml$^{-1}$ ampicillin or 2.5 µg ml$^{-1}$ erythromycin. All media were prepared with water purified by a VEOLIA PureLab Chorus2 water purification system.

### Antimicrobial spectrum of IVK83
The antimicrobial activity of *S. epidermidis* IVK83, which was isolated from a nasal swab of a healthy volunteer, was assessed by spotting *S. epidermidis* IVK83 on TSA agar plates containing lawns of the bacterial test strains listed in Supplementary Table 5. To this end, the test strains from fresh agar plates were resuspended in phosphate-buffered saline (PBS, Gibco), adjusted to an optical density (OD)$_{600}$ of 0.5 and streaked out evenly on TSA or TSA+ blood plates with a cotton swab. *S. epidermidis* IVK83 grown on TSA was resuspended in PBS, adjusted to an OD$_{600}$ of 1, and 10 µl was spotted on the bacterial lawn. Plates were incubated for 24 h to 48 h at 37 °C, anaerobic bacteria were grown under anaerobic conditions as above, and diameters of inhibition zones were measured.

### Transposon mutagenesis for the identification of the epifadin BGC
Identification of the epifadin BGC was performed via transposon mutagenesis as described earlier[51]. Briefly, *S. epidermidis* IVK83 was transformed with vector pTV1ts[52,53] containing the transposon Tn917 with an erythromycin resistance gene (*ermB*). After growing in TSB supplemented with 10 µg ml$^{-1}$ chloramphenicol at 30 °C overnight, the culture was diluted (1:1,000) in TSB containing 2.5 µg ml$^{-1}$ erythromycin and cultivated at 42 °C overnight. This step was repeated once

more with 2.5 μg/mL erythromycin and once without erythromycin and the cells were subsequently plated on TSA containing 2.5 μg ml⁻¹ erythromycin. Erythromycin-resistant but chloramphenicol-sensitive mutants, which probably had *Tn917* integrated, and the plasmid lost, were screened for loss of antimicrobial activity against *S. aureus*. The insertion sites were identified by sequencing upstream- and downstream-flanking regions of the transposon of genomic DNA isolated from non-inhibiting clones with primers Tn917 up and Ptn2 down.

### Whole-genome analysis of *S. epidermidis* IVK83

Whole-genome sequence of *S. epidermidis* IVK83 was determined by Illumina short-read and PacBio long-read sequencing. Illumina reads were de novo assembled by velvet (version 1.2.10)[54], and *S. epidermidis* IVK83 was also sequenced using the PacBio Sequel platform. SMRT-bell libraries were constructed using standard procedures (Pacific Biosciences) and the genome was de novo assembled using the Hierarchical Genome Assembly Process (HGAP4) workflow in SMRT Link (v.5.1.0.26411; Pacific Biosciences). Alignment of the two de novo assemblies with MAUVE[55] (version 2.4.0) and subsequent manual curation allowed us to generate the final genome, which was confirmed by mapping the Illumina reads to the final assembly. The circular chromosome and the plasmid were annotated using the NCBI Prokaryotic Genome Annotation Pipeline (version 5.3) and deposited at NCBI.

### Genome sequence analysis of epifadin-resistant *S. aureus*

Five clones of *S. aureus* USA300 JE2 were stored from either day 3 from mixed species culture experiments with *S. epidermidis* IVK83 or day 2 from mixed culture experiments with the mutant strain IVK83 Δ*efiTP* before its extinction on day 3. Cultured cells from each individual clone were added to tubes with DNA/RNA Shield solution (Zymo Research) and DNA extraction and sequencing was performed by MicrobesNG using Nextera XT library protocol on a HiSeq platform, generating 250-bp paired-end reads (Illumina). The resulting datasets are available from the SRA under BioProject number (PRJEB56114). From the obtained reads, Trimmomatic[56] (v0.39) was used to trim adapters and low-quality bases and read qualities were assessed using FastQC v0.11.7 (ref. [57]) and MultiQC v1.0 (ref. [58]). Genome sequences were assembled de novo and annotated using Unicycler v 0.4.7 (ref. [59]) with default parameters, using SPAdes v 3.15.4 (ref. [60]) and Prokka v 1.14.6 (ref. [61]).

### Construction of an epifadin production-deficient mutant and complementation

To generate a production-deficient mutant of *S. epidermidis* IVK83, the thermosensitive plasmid pBASE6 (ref. [62]) and the primers listed in Supplementary Tables 6 and 7 were used. For its construction, DNA fragments up- and downstream of the genes *efiP* (phosphopantetheinyl transferase) and *efiT* (thioesterase) were amplified by polymerase chain reaction using primers 83 KO Acc65I with 83 KO EcoRI and 83 KO BssHII with 83 KO SalI, respectively. After digestion of the upstream fragment with restriction enzymes EcoRI/Acc65I (Thermo) and of the downstream fragment with BssHII/SalI (Thermo), the two fragments and the Acc65I–BssHII-digested erythromycin resistance cassette from plasmid pEC2 (ref. [63]) were ligated into the EcoRI/SalI-digested vector pBASE6. The resulting plasmid pBASE6_KO83 was cloned in *Escherichia coli* DC10B, isolated and used to transform IVK83. Homologous recombination was monitored according to established procedures[62], resulting in the epifadin-deficient mutant IVK83 Δ*efiTP*.

For mutant complementation, the genes *efiTP* were amplified by polymerase chain reaction using primers 83compl BamHI and 83compl EcoRI. After digestion with BamHI/EcoRI (Thermo), the DNA fragment was ligated into BamHI/EcoRI-digested pRB474 (ref. [64]). After transformation and amplification of the resulting vector pRB474-83compl in *E. coli* DC10B, the vector was used to transform the epifadin-deficient IVK83 Δ*efiTP* and clones were screened for antimicrobial activity against *S. aureus*.

### Purification of epifadin

All steps of the purification process were performed under reduced light exposure. An overnight culture of IVK83 in TSB was used to inoculate fresh TSB 1:1,000, which was incubated for 4 h at 37 °C and constant shaking at 160 rpm to obtain bacteria in exponential growth phase. Subsequently, fresh TSB containing 5 g l⁻¹ glucose was inoculated with an $OD_{600}$ of 0.00125 of IVK83 and incubated for 44 h at 30 °C with constant shaking at 160 rpm. Cultures were centrifuged and the supernatant adjusted to pH 2 using 37% HCl (Thermo Fisher Scientific) for 2 h at 4 °C. The acidified supernatants were centrifuged at 8,000*g* for 15 min at 4 °C, the clear supernatant was discarded, and the obtained precipitate was resuspended in small volumes of de-ionized water and frozen at −80 °C. The precipitate was lyophilized at −20 °C and 1 mbar until water was completely removed. DMSO (Merck) supplemented with 0.05% PA was added 20:1 (ml vol/g weight) to the precipitate to extract the active compound from the insoluble particles of the precipitate. After gentle vortexing and centrifugation, the DMSO supernatant was transferred to a new vial. Extraction of the precipitate was repeated once more. DMSO extracts were combined and lyophilized, and the dry extract resuspended in a 1:2 mixture of system A buffer (99.95% $H_2O$, 0.05% trifluoroacetic acid (TFA)) and system B buffer (80% acetonitrile (Baker), 19.95% $H_2O$, 0.05% TFA). Subsequently, the solution was injected to a preparative, RP-HPLC column (Kromasil 100 C18, 7 μm, 250 × 4 mm, Dr. Maisch GmbH) and HPLC was performed with a gradient ranging from 50% system B to 100% system B within 30 min at 383 nm (diode-array detector). The product-containing fractions were collected at −77 °C (dry ice isopropyl alcohol cooling bath) under argon atmosphere, lyophilized and stored at −80 °C until further use.

### Isolation of the natural epifadin peptide moiety

The decomposed epifadin NMR sample was lyophilized and resolved in 1 ml system C:system D (2:3, C: water with 0.1% formic acid, D: acetonitrile with 0.1% formic acid). After centrifugation, the supernatant was injected to a semi-preparative RP-HPLC column (Kromasil 100 C18, 5 μm, 250 × 8 mm, Dr. Maisch GmbH) and HPLC was performed with a gradient ranging from 10% D to 100% D in water (C) within 30 min. The flow rate was set to 2 ml min⁻¹. The UV absorption was recorded at 210 nm. The peptide-containing fraction was frozen and lyophilized.

### Stability analysis of epifadin

The effect of different incubation conditions on the bioactivity of epifadin was analysed by agar diffusion assays on lawns of the sensitive indicator strain *S. aureus* USA300 LAC. To this end, liquid TSA was inoculated with an overnight culture of USA300 at an $OD_{600}$ of 0.00125. After solidification of the agar, a cork borer was used to punch wells into the agar with 5 mm diameter. TSB media with 50 mM sodium citrate (pH 5 and 6) or 50 mM tris(hydroxymethyl)aminomethane (pH 8 and 9) were prepared. Epifadin-containing precipitate was dissolved in DMSO (12 mg ml⁻¹), and insoluble particles were removed by centrifugation. The supernatant was added to the prepared TSB media at a final concentration of 750 μg precipitate per millilitre (37.5 μg in 50 μl). Media were incubated for 6 h at 21 °C, 30 °C or 37 °C with or without laboratory light exposure and constant shaking at 500 rpm in a thermomixer (Eppendorf). As controls, 70 μg ml⁻¹ (3.5 μg in 50 μl) lugdunin, 24 μg ml⁻¹ (1.2 μg in 50 μl) gallidermin and 100 μg ml⁻¹ (5 μg in 50 μl) nisin were used. At different timepoints, 50 μl of each sample was pipetted into the wells of the agar plates. After incubation at 37 °C overnight, zones of inhibition were analysed with ImageJ software (version 1.8.0_112).

### Analysis of epifadin by HPLC-UV−HR-MS

Mass spectra were recorded on a HPLC-UV−HR-MS (MaXis4G with Performance Upgrade kit with electrospray ionization (ESI)-Interface, Bruker Daltonics) with time-of-flight (TOF) detection. To obtain HR-MS data, DMSO extracts were lyophilized, resuspended in an acetonitrile−water mixture (1:1) and applied to a Dionex Ultimate 3000 HPLC

system (Thermo Fisher Scientific), coupled to the MaXis4G ESI-QTOF mass spectrometer (Bruker Daltonics). The ESI source was operated in ESI (+) mode at a nebulizer pressure of 2.0 bar, and dry gas was set to 8.0 l min$^{-1}$ at 200 °C. MS/MS spectra were recorded in auto-MS/MS mode with collision energy stepping enabled. Sodium formate was used as internal calibrant in each analysis. The routine gradient was from 90% de-ionized $H_2O$ with 0.1% formic acid and 10% methanol with 0.6% formic acid to 100% methanol with 0.6% formic acid in 20 min with a flow rate of 0.3 ml min$^{-1}$ on a Nucleoshell EC RP-C$_{18}$ (150 × 2 mm, 2.7 μm) from Macherey-Nagel.

## Proof of absolute configuration of the natural peptide amide FfDn–NH$_2$ by HPLC–ESI–QTOF-MS

MS detection was performed on a TripleTOF 5600+ mass spectrometer with a DuoSpray source (AB Sciex Instruments) and operated in positive ESI mode (general MS settings were as follows: curtain gas (N$_2$) 35 psi, nebulizer gas (zero-grade air) 50 psi, heater gas (zero-grade air) 40 psi, ion source voltage floating 5,000 V, source temperature 600 °C, scan window $m/z$ = 100 to 2,000, accumulation time 250 ms). The mass spectrometer was coupled to an Agilent 1290 series UHPLC pump and column thermostat equipped with a CTC-PAL HTS autosampler (CTC Analytics). The samples (synthetic peptides FfDn–NH$_2$ (L-D-L-D) and fFdN–NH$_2$ (D-L-D-L), a mixture of both and the natural peptide) were dissolved in the mobile phase and the concentration was set to 0.1 mg ml$^{-1}$. The mobile phase used was acetonitrile/methanol/water (49:49:2 (v/v/v)) with 50 mM formic acid/25 mM ammonia and isocratic elution was carried out with a flow rate of 0.5 ml min$^{-1}$ for 16 min on a Daicel Chiralpak ZWIX(+) (150 × 3.0 mm, 3 μm) from Chiral Technologies. The injection volume was 2 μl, and the column temperature was set to 25 °C.

## NMR spectroscopy

$^1$H-NMR was performed on Bruker AMX-600 (600 MHz) and Bruker AvanceIII-700 (700 MHz) instruments at 303 K. Chemical shifts were monitored as $\delta$ values (ppm) relative to the solvent DMSO-d$_6$ as internal standard. Coupling constants ($J$) were monitored in Hertz (Hz). Abbreviations for multiplicity description are as follows: s, singlet; d, duplet; dd, duplet of a duplet; and m, multiplet. $^{13}$C-NMR was performed on a Bruker AMX-600 (150.3 MHz) instrument at 303 K. Chemical shifts were monitored as $\delta$ values (ppm) relative to the solvent DMSO-d$_6$ as internal standard. Homo- and heteronuclear correlation experiments: $^1$H-$^1$H-COSY (correlated spectroscopy), $^1$H-$^1$H-ROESY (rotating frame overhauser effect spectroscopy) and $^1$H-$^{13}$C-HMBC (heteronuclear multiple bond correlation).

## Chemical SPPS of FfDn–NH$_2$

Fmoc-D-Asn(Trt) resin (loading: 0.23 mmol g$^{-1}$, 150 mg, 34.5 μmol scale, TG S RAM, Rapp Polymere) was swollen in 2 ml dimethylformamide (DMF) for 30 min. The Fmoc group was removed by treatment with a solution of 2% DBU/10% morpholine (v/v) in DMF (2 ml) for 3 min and additional 12 min. The resin-bound residue was submitted to iterative peptide assembly (Fmoc-solid-phase peptide synthesis (SPPS)) using 2% DBU/10% morpholine (v/v) in DMF (2 ml, 3 + 12 min) for Fmoc-deprotection and Fmoc-D/L-AAx-OH (Fmoc-L-Asp(OtBu)-OH, Fmoc-D-Phe-OH and Fmoc-L-Phe-OH) (6 equiv.), hexafluorophosphate azabenzotriazole tetramethyl uronium (HATU) (6 equiv.), 1-hydroxybenzotriazole (HOBt) (6 equiv.) and N-methylmorpholine (NMM) (8 equiv.) in N,N-dimethylformamide (DMF) (2 ml) for 45 min to couple each amino acid. After full assembly of linear peptide amide on the solid support, the resin was washed with DMF (3 × 2 ml), dichloromethane (3 × 2 ml), toluene (3 × 2 ml), isopropyl alcohol (3 × 2 ml) and diethylether (Et$_2$O) (3 × 2 ml) and dried under reduced pressure for 3 h. The peptide was cleaved by treatment with trifluoroacetic acid/triisopropylsilanol/water (TFA/TIPS/H$_2$O) (95:5:5 v/v/v, 2 ml) for 3 × 1 h and one washing step with TFA (2 ml) for 10 min. The solvents

were removed under reduced pressure, and the residue was washed with Et$_2$O (3 × 2 ml) and centrifuged. The precipitate was lyophilized using tert-butanol/water (1:1 v/v, 10 ml).

## Chemical solid-phase peptide synthesis of fFdN–NH$_2$

Fmoc-L-Asn(Trt) TG S RAM resin (loading: 0.23 mmol g$^{-1}$, 150 mg, 34.5 μmol scale, Rapp Polymere) was swollen in DMF (2 ml) for 30 min. The Fmoc group was removed by treatment with a solution of 2% DBU/10% morpholine (v/v) in DMF (2 ml) for 15 min. The resin-bound residue was submitted to iterative peptide assembly (Fmoc-SPPS) using 2% DBU/10% morpholine (v/v) in DMF (2 ml, 15 min) for Fmoc-deprotection. Protected amino acids (Fmoc-D/L-AAx-OH, each at a time: Fmoc-D-Asp (OtBu)-OH, Fmoc-L-Phe-OH and Fmoc-D-Phe-OH, each 6 equiv.), HATU (6 equiv.), HOBt (6 equiv.) and NMM (8 equiv.) in DMF (2 ml) were added to the reaction for 45 min to couple each amino acid. The remaining steps were performed just as described for the FfDn–NH$_2$ peptide.

## Acetylation of the tetrapeptide amide via chemical reaction with epifadin extract

To 10 mg precipitate of IVK83 wild-type culture supernatant, 200 μl 50 mM ammonium bicarbonate and 500 μL acetylation reagent (25% acetic acid anhydride in methanol) were added and stirred at room temperature for 1 h. One hundred microlitres of the reaction solution was taken, lyophilized and dissolved in 50 μl methanol (LC–MS grade) for HPLC–MS analytics.

## Esterification of the tetrapeptide amide via chemical reaction of epifadin extract with methanol

To 10 mg precipitate of IVK83 wild-type culture supernatant, 1 ml methanol and a droplet of sulfuric acid were added and stirred at room temperature overnight. After centrifugation of the reaction solution, 100 μl was taken for HPLC–MS analytics.

## Detailed methylation reaction conditions

To a solution of purified epifadin (**1**, 2.7 mg, 2.8 μmol) in toluene/methanol (8/2) was added a solution of trimethylsilyldiazomethane in diethyl ether (2 M, 100 μl, 200 μmol, 71 equiv.). The resulting yellow solution was stirred 2 h at room temperature. Acetic acid was added until the gas evolution ceased. The solvent was removed and a brown residue (0.9 mg) was obtained and further analysed by HPLC–HR-MS. To a solution of reutericyclin (**6**, 1.5 mg, 4.3 μmol) in toluene/methanol (8/2) was added a solution of trimethylsilyldiazomethane in diethyl ether (2 M, 43 μl, 86 μmol, 20 equiv.). The resulting yellow solution was stirred 24 h at room temperature. Acetic acid was added until the gas evolution ceased. The solvent was removed and a yellow residue was obtained and further analysed by HPLC–HR-MS. To a solution of kirromycin (**7**, 20.0 mg, 25.1 μmol) in toluene/methanol (8/2) was added a solution of trimethylsilyldiazomethane in diethyl ether (2 M, 50 μl, 100 μmol, 4 equiv.). The resulting yellow solution was stirred 4 h at room temperature. After adding an additional 8 equiv. of trimethylsilyldiazomethane the solution was stirred for 20 h. Acetic acid was added until the gas evolution ceased. The solvent was removed and a brown residue was obtained and further analysed by HPLC–HR-MS.

## Determination of the IC of epifadin

Due to the low stability of epifadin under standard culture conditions and resulting low yields after purification, minimal IC determination via standardized laboratory approaches such as microdilution or standard agar dilution was not feasible. Instead, a miniaturized assay was developed. Molten TSA, kept below 42 °C, was inoculated with different test strains at an OD$_{600}$ of 0.00125. Then, 17.5 ml was poured into Petri dishes, resulting in agar layers of ~3.16 mm thickness. After solidification, a cork borer was used to punch wells of 4-mm diameter into the agar. Thirty-five microlitres of purified epifadin in DMSO–PA

(2.4 µg, 1.2 µg, 0.6 µg, 0.3 µg, 0.15 µg and 0.075 µg) was pipetted into the wells, and agar plates were incubated for 24 h at 37 °C in the dark. Diameters of the zones of inhibition were measured and the volume containing the inhibitory epifadin concentration was calculated, using the following formula for cylinder volume:

$$V_c = \pi \times r^2 \times h$$

$V_c$ = cylinder volume

$$V_i = (\pi r_z^2 \times h_a) - (\pi r_w^2 \times h_a)$$

$V_i$ = inhibited volume

$r_z^2$ = radius (diameter/2) zone of inhibition

$r_w^2$ = radius (diameter/2) of well

$h_a$ = height of agar

The volume of agar showing inhibition of the test strain was then correlated with the included amount of epifadin, and the epifadin concentration necessary to inhibit test strain growth in a 1 ml volume was calculated to yield IC values. If, for example, 2.4 µg of epifadin lead to an inhibited volume of 480 µl, to inhibit a volume of 1 ml, 5 µg epifadin is needed, which corresponds to an IC of 5 µg ml⁻¹. These calculations were conducted for the lowest amount of epifadin that led to a zone of inhibition to minimize the necessary epifadin amounts and the calculation error due to the gradient within the inhibition zone. As a control, the same assay was used to determine the IC for vancomycin, daptomycin and lugdunin under these conditions.

## Minimal bactericidal concentration determination

Fresh TSB was inoculated 1:1,000 with an overnight culture of *S. aureus* USA300 LAC and incubated for 4 h at 37 °C and constant shaking at 160 rpm. Bacteria were centrifuged and washed once with PBS. Subsequently, cells were resuspended in PBS to an $OD_{600}$ 0.00125 ($1 \times 10^6$ colony-forming units (CFU) ml⁻¹) and different concentrations of purified epifadin in DMSO–PA were added. Bacteria were incubated for 4 h at 37 °C and shaking in the dark (500 rpm, thermomixer). Every 30 min, samples were taken, and serial dilutions were plated on TSA. After 24 h incubation at 37 °C, CFUs were counted.

## In vitro competition assay

*S. epidermidis* IVK83 wild type, *S. epidermidis* IVK83 Δ*efiTP*, *S. epidermidis* IVK83 Δ*efiTP* pRB474-83compl and a streptomycin-resistant *S. aureus* Newman strain were grown overnight in TSB at 37 °C and constant shaking at 160 rpm. The bacteria were washed once with TSB and adjusted to $5 \times 10^8$ CFU ml⁻¹ in TSB. For liquid culture-based competition, 10 ml of *S. aureus* and 10 ml *S. epidermidis* suspension were mixed in shaking flasks and incubated at 37 °C and constant shaking at 160 rpm. After 24 h of incubation, fresh TSB was inoculated to an $OD_{600}$ 0.5 with the previous culture (to ensure nutrient availability for antimicrobial production by IVK83) and further incubated at 37 °C; this procedure was repeated twice. Samples were taken at 0 h, 24 h, 48 h and 72 h and serial dilutions were plated on TSA and TSA supplemented with streptomycin for *S. aureus* Newman selection. After overnight incubation at 37 °C, CFUs were counted, and bacterial ratios were calculated.

For agar-based competition, overnight cultures of *S. aureus* and *S. epidermidis* were washed once with PBS and adjusted to $1 \times 10^8$ CFU ml⁻¹ in PBS. Five hundred microlitres of *S. aureus* and 500 µl of *S. epidermidis* were mixed, and $3 \times 20$ µl drops were spotted on TSA and incubated at 37 °C. After 24-h incubation, grown bacterial cells were scraped off from the three spots, individually resuspended in PBS and adjusted to an $OD_{600}$ 0.5 in 1 ml PBS, from which again $3 \times 20$ µl were spotted on fresh TSA and incubated at 37 °C; this procedure was repeated twice. Samples of bacteria were taken at 0 h, 24 h, 48 h and 72 h and resuspended, and serial dilutions were plated on TSA and TSA supplemented with streptomycin to select for *S. aureus*. After overnight incubation at 37 °C, CFUs were counted, and bacterial ratios were calculated.

## Animal models and ethics statement

All animal experiments were conducted in strict accordance with the German regulations of the Gesellschaft für Versuchstierkunde/Society for Laboratory Animal Science (GV-SOLAS) and the European Health Law of the Federation of Laboratory Animal Science Associations (FELASA) in accordance with German laws after approval by the local authorities (IMIT 1/15, Regierungspräsidium Tübingen). Animal studies were carried out at the University Tübingen and conformed to institutional animal care and use policies. No randomization or blinding was necessary for the animal colonization model, and no samples were excluded. Animal studies were performed with cotton rats of both genders, 8–12 weeks old, respectively.

## Colonization and co-colonization of cotton rat noses

Spontaneously streptomycin-resistant *S. epidermidis* IVK83 and *S. epidermidis* IVK83 Δ*efiTP* mutants were generated by incubating and passaging those strains on TSA supplemented with 250 µg ml⁻¹ streptomycin. For co-colonization of cotton rat noses, these strains and streptomycin-resistant *S. aureus* Newman were used. The cotton rat colonization model has been described previously[36]. We have shown that an inoculum of $1 \times 10^7$ CFU per nose is required to achieve stable colonization of $1 \times 10^3$–$1 \times 10^4$ CFU per nose for *S. aureus* Newman, while other staphylococcal species such as *S. lugdunensis* may require a higher inoculum[25]. Since the capability of *S. epidermidis* IVK83 and *S. epidermidis* IVK83 Δ*efiTP* to colonize cotton rat noses was unknown, the inoculum required to achieve stable colonization was determined first. In brief, overnight cultures were washed twice in PBS and inocula were adjusted to $1 \times 10^8$ CFU per 10 µl. Subsequently, cotton rats were anaesthetized with isoflurane and instilled intranasally with $1 \times 10^8$ CFU per nose. After 5 days, the cotton rats were euthanized, and the noses were removed and covered in 1 ml PBS. After heavy vortexing for 30 s, dilutions of the samples were plated on BM agar supplemented with 250 µg ml⁻¹ streptomycin and incubated overnight at 37 °C to obtain CFU per nose. Whereas an inoculum of $1 \times 10^7$ CFU per nose is sufficient for *S. aureus* Newman to establish colonization, *S. epidermidis* IVK83 and IVK83 Δ*efiTP* had to be instilled with $1 \times 10^8$ CFU per nose to obtain comparable colonization levels. Based on these parameters, co-colonization was performed with tenfold increased inoculum for IVK83 and IVK83 Δ*efiTP*.

For co-colonization experiments, cotton rats were instilled intranasally with a mixture of $1 \times 10^7$ CFU per nose of *S. aureus* Newman and either $1 \times 10^8$ CFU per nose IVK83 or IVK83 Δ*efiTP*. Five days after instillation, bacteria were extracted from cotton rat noses as described above and the bacterial ratio was calculated; *S. aureus* (yellow) and *S. epidermidis* (white) were distinguished by colour and colony size.

## Cytotoxicity assay in HeLa cells

The human cervical carcinoma HeLa cell line was cultivated in RPMI cell culture medium (Thermo Fisher Scientific) supplemented with 10% foetal bovine serum (Thermo Fisher Scientific) at 37 °C, 5% $CO_2$ and 95% relative humidity. A twofold serial dilution of epifadin in RPMI was prepared in a microtitre plate, and trypsinized HeLa cells were added at a final cell concentration of $1 \times 10^4$ per well. After 24-h incubation, 7-hydroxy-3*H*-phenoxazin-3-one-10-oxide (resazurin) was added at a final concentration of 200 µM and cells were again incubated for 24 h. Cell viability was evaluated by determining the reduction of resazurin to the fluorescent resorufin ($\lambda_{ex}$ = 560 nm, $\lambda_{em}$ = 600 nm) in relation to an untreated control in a TECAN Infinite M200. Cycloheximide served as a positive control.

## Experimental evolution of *S. aureus* USA300 JE2 to select for resistance to epifadin-producing *S. epidermidis* IVK83

*Staphylococcus* strains were cultured on (BHI) agar plates (LAB M) at 37 °C overnight before competition experiments. All strains were scraped off separate agar plates, suspended in 10 ml of PBS (Sigma-Aldrich) and vortexed to mix thoroughly. The bacterial

suspensions were diluted to $OD_{600}$ of 0.1 and the indicator and producer were then mixed together at a ratio of 1:1. The co-culture mixtures were vortexed, and 50 µl of cells (~$2.5 \times 10^6$ CFU) was spotted onto 25-ml BHI agar plates. Each species was also spotted separately as a control and plates were incubated at 37 °C for 24 h. The next day, the cultures were scraped from the agar and individually resuspended in 10 ml PBS before repeat culture as previous, followed by daily repeats for 8 days. Viable counts were used to enumerate each species, and the two bacteria were differentiated by their morphology and pigmentation.

## Time-lapse microscopy

For microscopic experiments, *S. aureus* USA300 JE2 cells were grown in TSB medium at 37 °C with agitation (180 rpm) overnight. The next day, cultures were diluted (1:100) in fresh medium and grown to an $OD_{600}$ of 0.4. A microscopy slide was covered with two small, separated agarose pads (1.2% agarose in TSB/PBS 1:10, including the fluorescent dyes FM4-64 (0.25 µg ml$^{-1}$, Thermo Fisher) and Sytox Green (0.25 µM, Thermo Fisher). The epifadin-containing extract obtained from the producer strain was dissolved at 50 mg ml$^{-1}$ in DMSO and centrifuged at maximal speed for 5 min, and 2 µl of the supernatant was spotted to the middle of one agarose pad. On the second agarose pad, 2 µl of the supernatant from the BGC deletion mutant was spotted in the same way. After 5 min, when the DMSO was evaporated, 1 µl of the bacterial culture, pre-stained with FM4-64 (20 µg ml$^{-1}$) for 10 min, was spotted onto each pad. Fifteen minutes after spotting the cells, image acquisition was started. Cells were monitored in a short distance to the extract spots, visualized by bright-field and fluorescence microscopy in a Zeiss Axio Observer Z1 LSM800 at $\lambda_{ex} = 506$ nm/$\lambda_{em} = 751$ nm (FM4-64) and at $\lambda_{ex} = 504$ nm/$\lambda_{em} = 524$ nm (Sytox Green) at 30 °C in 5-min intervals. Image processing was performed in FIJI[65].

## Membrane potential assay

The membrane potential assay was performed as described before[66]. *S. aureus* NCTC8325 was grown in LB medium at 37 °C, 200 rpm to an $OD_{600}$ of 0.75. Cells were pelleted and resuspended to an $OD_{600}$ of 0.5 in PBS and incubated with 30 µM 3,3′-diethyloxacarbocyanine iodide (DiOC$_2$(3)) (Molecular Probes, Fisher Scientific) for 15 min in the dark. The loaded cells were transferred to a black 96-well flat-bottom polystyrol microtitre plate (BRAND), and a base line measurement was taken for 2 min at an excitation wavelength of 485 nm and two emission wavelengths, 530 nm (green) and 630 nm (red), after which concentration series of the IVK83 wild type or Δ*efiTP* extract were added and the measurement continued as above for a total of 15 min, using a microplate reader (TECAN Spark). The protonophore carbonylcyanide-*m*-chlorphenylhydrazone (5 µM) was used as a positive control and DMSO (1%) as a negative control.

## Statistical analysis

Statistical analyses were performed using GraphPad Prism, version 8 (GraphPad Software, version 8). Data distribution was assumed to be normal, but this was not formally tested. Statistically significant differences were calculated by appropriate statistical methods as indicated. The number of replicates per experiment and the *P* values are indicated in the figures and legends. 'NS' indicates non-significance.

## Reporting summary

Further information on research design is available in the Nature Portfolio Reporting Summary linked to this article.

## Data availability

All data supporting the findings of this study are available within the paper, its extended data or supplementary information. Whole-genome sequencing data obtained for *S. epidermidis* IVK83 were deposited in the NCBI Sequence Read Archive (genome available under accession number CP088002, plasmid pIVK83 under CP088003). Sequence of strain *S. epidermidis* B155 (Liverpool, UK) was deposited as BioSample SAMEA12384066 (BioProject PRJEB50307). Representative microscopy images are included in the extended data figures and the supplementary videos, which were deposited at Figshare (https://doi.org/10.6084/m9.figshare.24125589). NMR data were deposited at nmrXive and are available under the project identifier NMRXIV:P18 (https://doi.org/10.57992/nmrxiv.p18; https://nmrxiv.org/P18). Source data are provided with this paper.

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

## Acknowledgements

We thank D. Belikova, V. Augsburger and J. Straetner for excellent technical support, M. Hamburger (Pharmaceutical Biology, University of Basel, Switzerland) for providing authentic sample of militarinone C, and A. Tooming-Klunderud (Center for Ecological and Evolutionary Synthesis, Department of Biosciences, University of Oslo, Norway) for PacBio sequencing of strain IVK83. The sequencing company MicrobesNG (Birmingham, UK) is supported by the Biotechnology and Biological Sciences Research Council; grant number BB/L024209/1). The authors' work is financed by grants from Deutsche Forschungsgemeinschaft (DFG) TRR261 (A.P., H.B.-O. and S.G.; project ID 398967434), GRK1708 (S.G., H.B.-O. and A.P.) and Cluster of Excellence EXC2124 Controlling Microbes to Fight Infection (CMFI, S.G., B.K., H.B.-O. and A.P.; project ID 390838134), TRR156 (A.P.; project ID 246807620), and ZUK 63 (N.A.S.); from the German Center of Infection Research (DZIF) to B.K., H.B.-O. and A.P.; from the Novo Nordisk Foundation (T.W., project ID NNF20CC0035580); from the German Ministry of Research and Education (BMBF) Culture Challenge to A.P.; and from the European Innovative Medicines Initiate IMI (COMBACTE) to A.P. We acknowledge support by the High Performance and Cloud Computing Group at the Zentrum für Datenverarbeitung of the University of Tübingen, the state of Baden-Württemberg through bwHPC and the German Research Foundation (DFG) through grant no. INST 37/935-1 FUGG.

## Author contributions

B.O.T.S. performed and analysed most of the bacteriological, molecular and compound isolation experiments with help by D.J., S.K. and B.K., who originally isolated strain IVK83; animal experiments were performed by B.O.T.S. and B.K.; T.D., N.A.S. and S.G. elucidated the structure of epifadin with support from J.M.B.-B.; T.D. performed total syntheses of all tetrapeptides, their purification and chemical analyses; A.B. analysed epifadin toxicity and membrane potential effects; J.B. performed all microscopic experiments; A.M.A.E. performed the bioinformatic search for epifadin-like BGCs; M. Li., M.J.H. and S.K. identified and provided epifadin-producing *S. epidermidis* strains.; N.A. and M.J.H. performed the experimental evolution and analysed the epifadin-resistant mutants; S.J.J. and M. Lämmerhofer confirmed the absolute configuration of the tetrapeptide via chiral HPLC; T.W. analysed the potential epifadin biosynthesis enzymes; H.B.-O., S.G., B.K. and A.P. supervised the experiments and wrote the manuscript.

## Competing interests

The authors declare no competing interests.

## Additional information

**Extended data** is available for this paper at https://doi.org/10.1038/s41564-023-01544-2.

**Correspondence and requests for materials** should be addressed to Stephanie Grond or Bernhard Krismer.

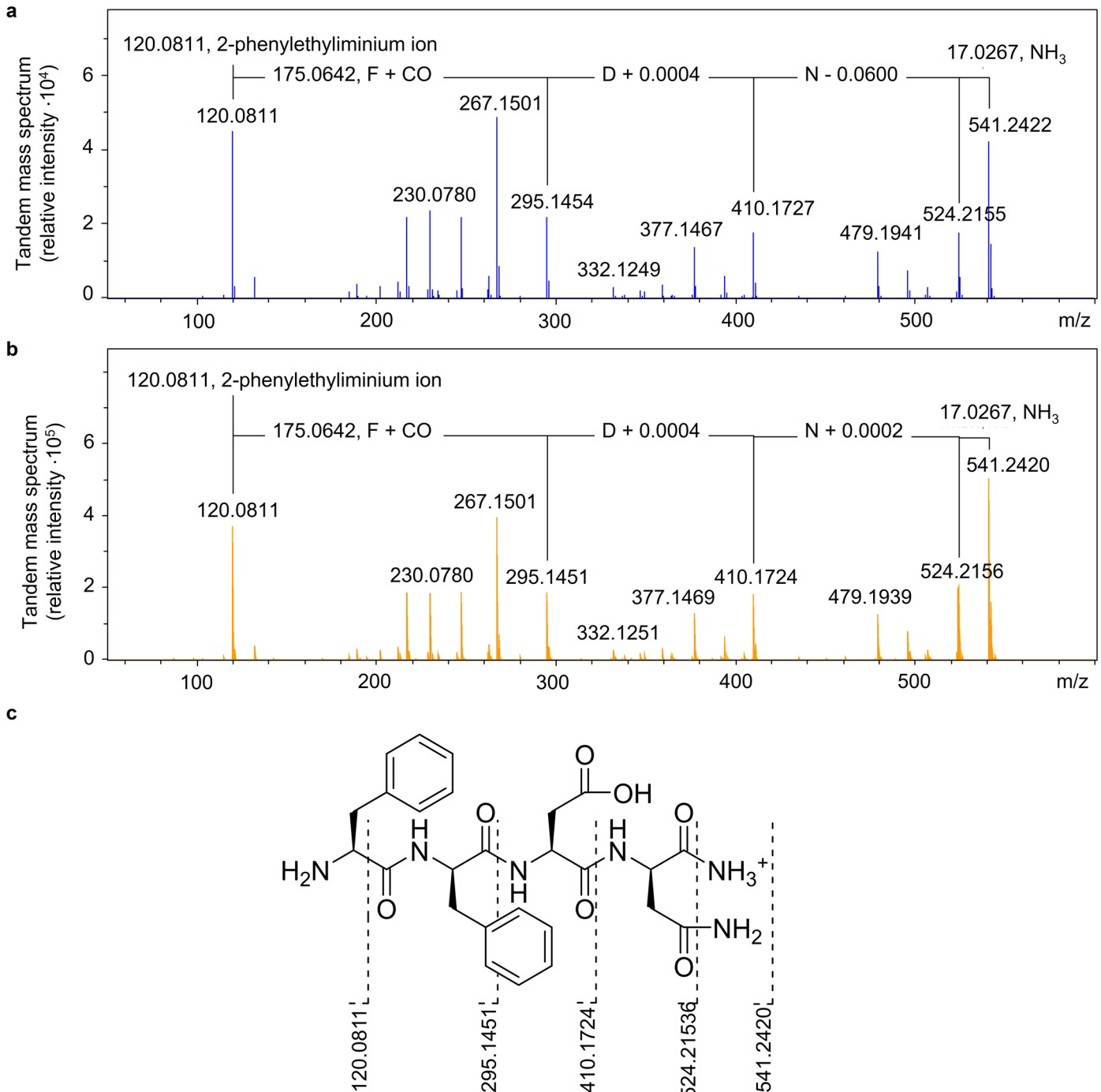

**Extended Data Fig. 1 | Comparison of MS/MS spectra of the synthetic and natural peptide amide fragments of epifadin.** *a*, MS/MS spectrum of the natural peptide amide after decomposition of epifadin. **b**, MS/MS spectrum of the synthetic peptide amide **2**. **c**, Fragmentation pattern of the synthetic and natural peptide amides **2**. Fragmentation pattern for the peptide amide **2** is shown in black. F, phenylalanine; D, aspartate; N, asparagine; CO, carbon monoxide; NH₃, ammonia.

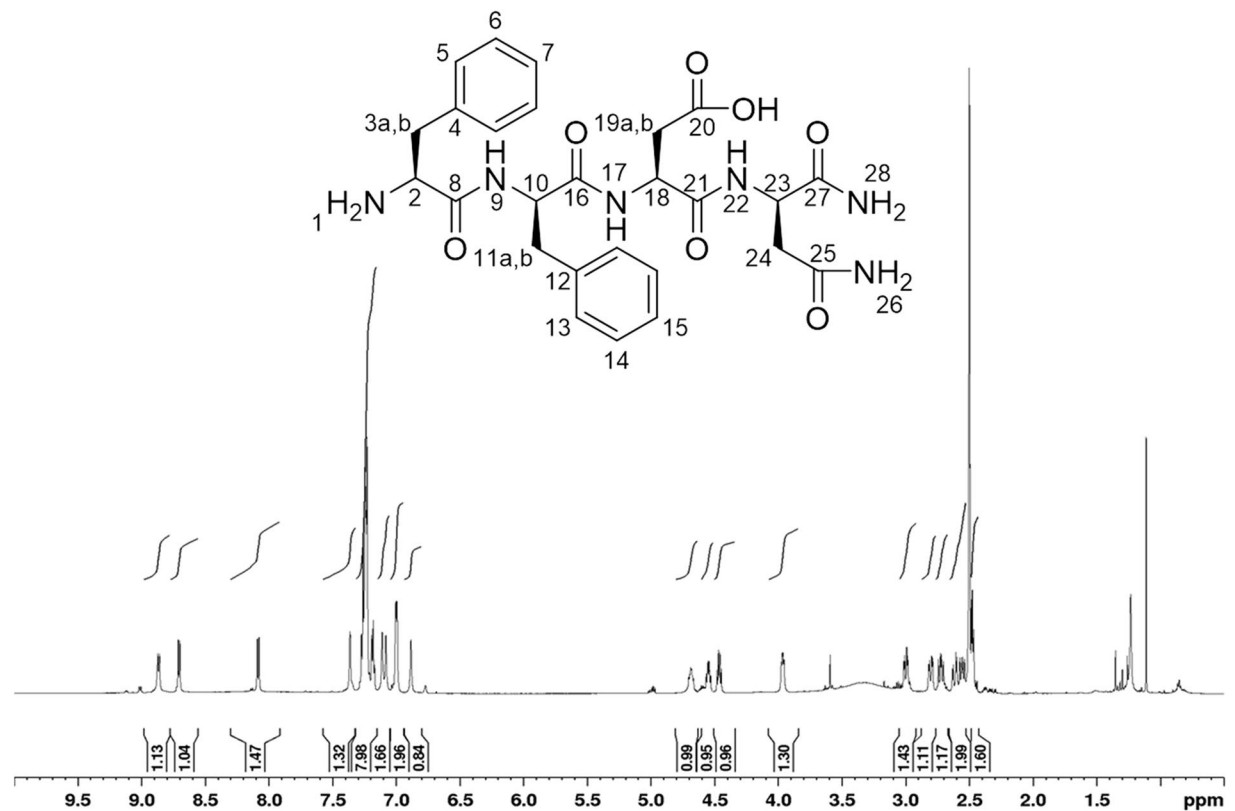

**Extended Data Fig. 2 | Proton NMR spectrum of the synthetic peptide amide FfDn-NH₂ (DMSO-d₆, 600MHz, 303K).** The integrals of the proton signals are depicted as black curves. The scale shows the chemical shift $\delta$ in parts per million (ppm).

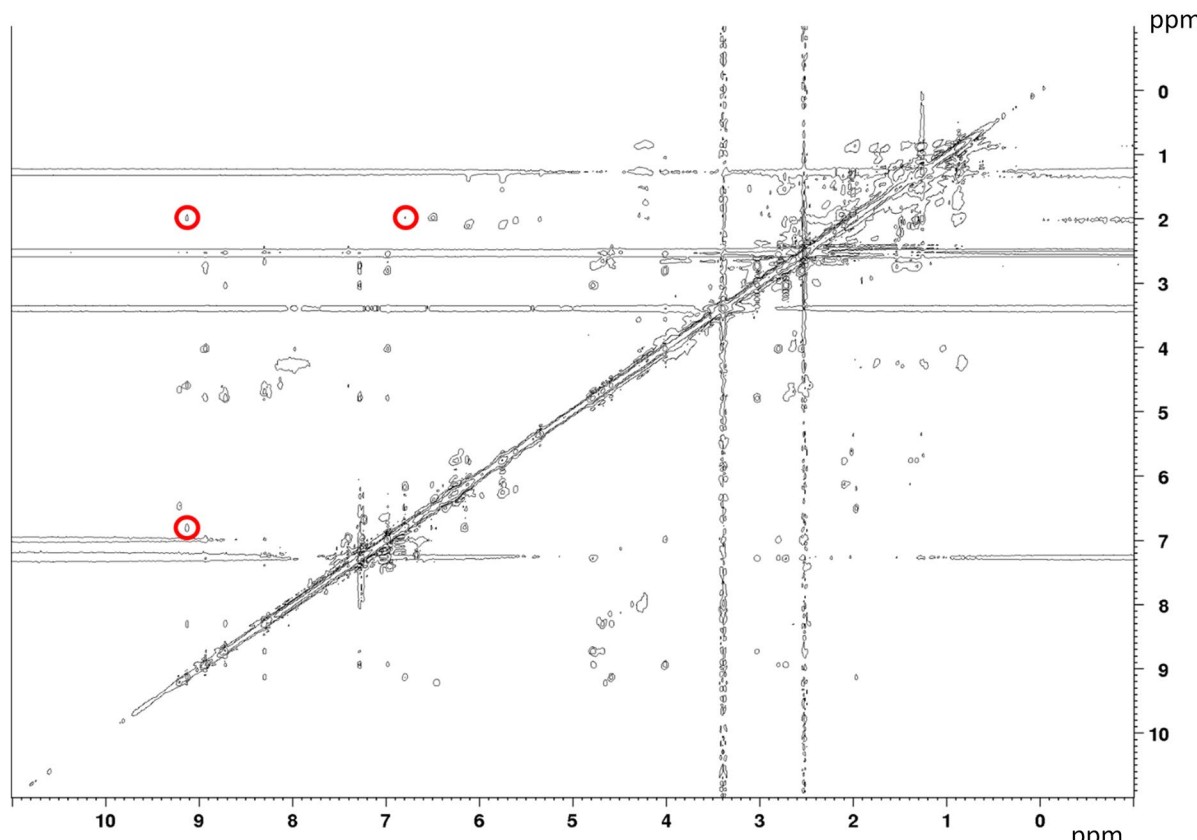

**Extended Data Fig. 3 | $^1$H-$^1$H-ROESY NMR spectrum of the purified epifadin in DMSO-d$_6$ (700 MHz, 303 K).** The red circles highlight coupling between the NH-proton and the protons of the methyl group of the alanine residue (9.14 ppm/1.95 ppm) and to the proton of the adjacent methine group (9.14 ppm/6.77 ppm). Also, the coupling of the protons of the methyl group from the alanine residue to the methine group is shown (6.77 ppm/1.95 ppm).

**a**

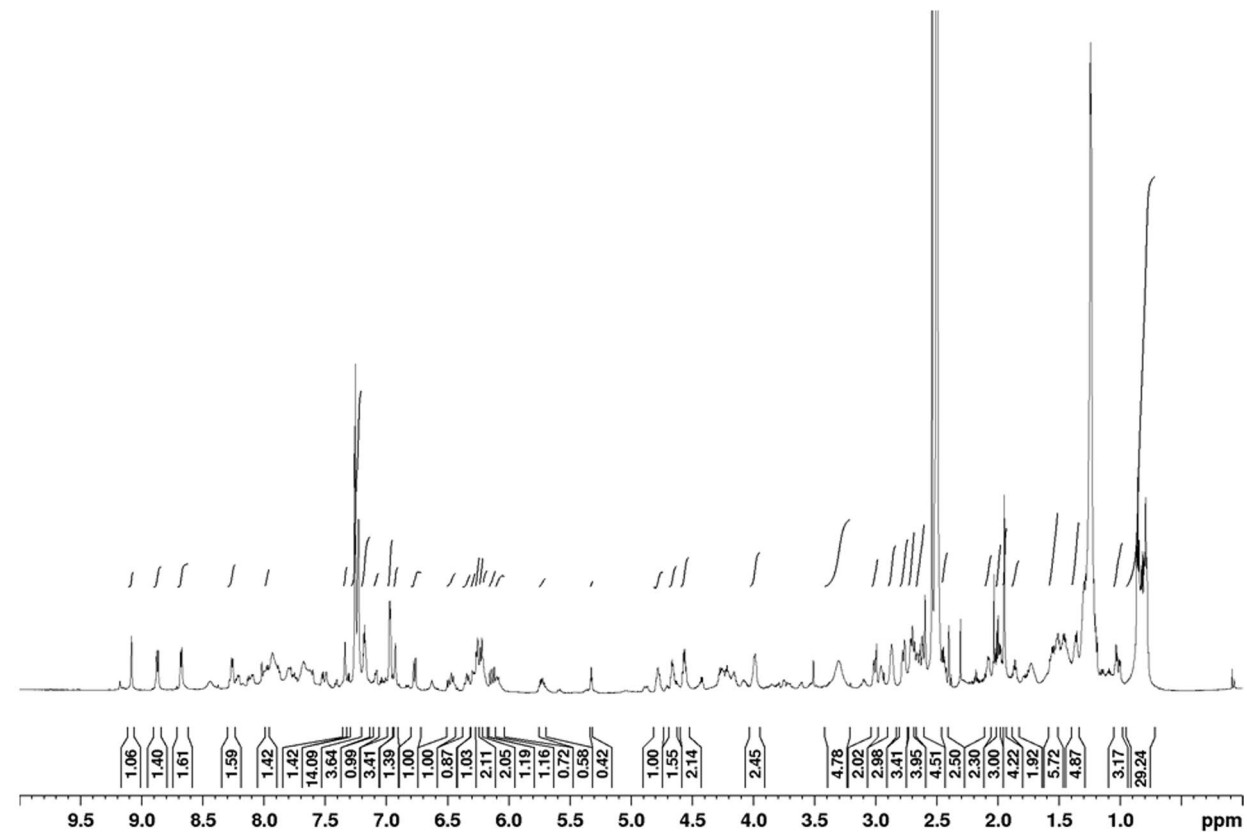

**b**

**c**

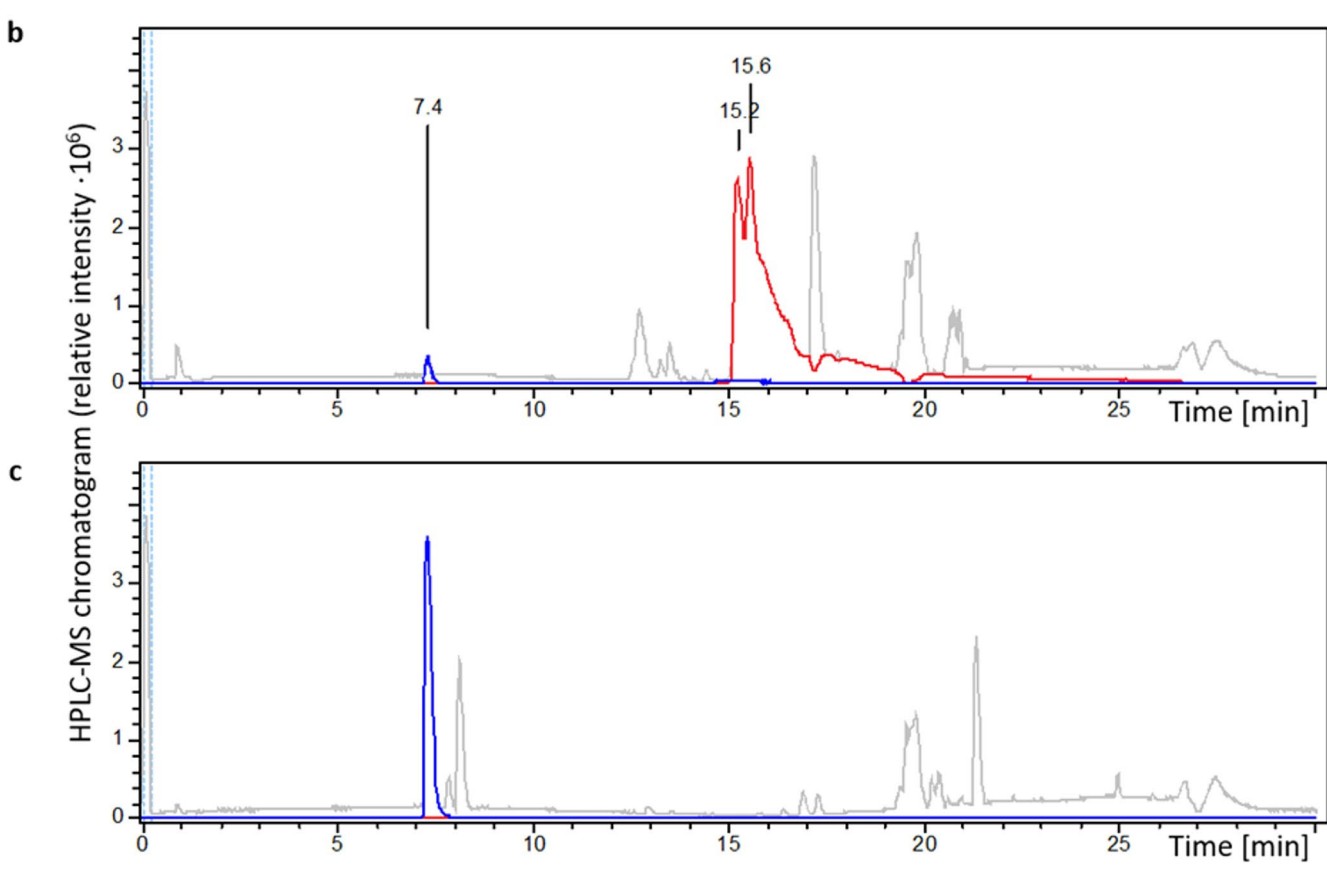

**Extended Data Fig. 4 | See next page for caption.**

**Extended Data Fig. 4 | $^{1}$H NMR spectrum (DMSO-d$_6$, 700 MHz, 303 K) of purified epifadin and its decomposition analyzed by HPLC-MS. a**, DMSO-d$_6$ signal at 2.50 ppm as reference. The integrals of the proton signals are depicted as black curves. The scale shows the chemical shift $\delta$ in parts per million (ppm). **b,c**, The epifadin-enriched material was dissolved in a mixture of acetonitrile and water (1:1) with 0.05% trifluoroacetic acid, resulting in a concentration of 0.2 mg/mL and analyzed by HPLC-ESI-TOF-high resolution MS. The extracted ion chromatograms (EICs) of epifadin ($C_{51}H_{61}N_7O_{12}$ [M+H]$^+$, $m/z$ 964.4451 ± 0.005) are depicted in red (retention time 15.2 min and 15.6 min) and the base peak chromatograms (BPCs) in gray. EICs of the peptide amide ($C_{26}H_{32}N_6O_7$ [M+H]$^+$, $m/z$ 541.2405 ± 0.005) are depicted in blue (retention time 7.4 min) accumulating by strong decomposition of epifadin in the mentioned solvent after storage at −20 °C. **b**, analyzed after purification. **c**, Analyzed after 14 days of storage at −20 °C.

**Extended Data Fig. 5 | Deduced fragmentation pattern for the peptide amide and the PKS/NRPS moiety.** From a six-membered transition state a rearrangement results in a neutral loss of the peptide amide moiety. The newly formed allene (*m/z* 424.2131) decomposes into further fragments.

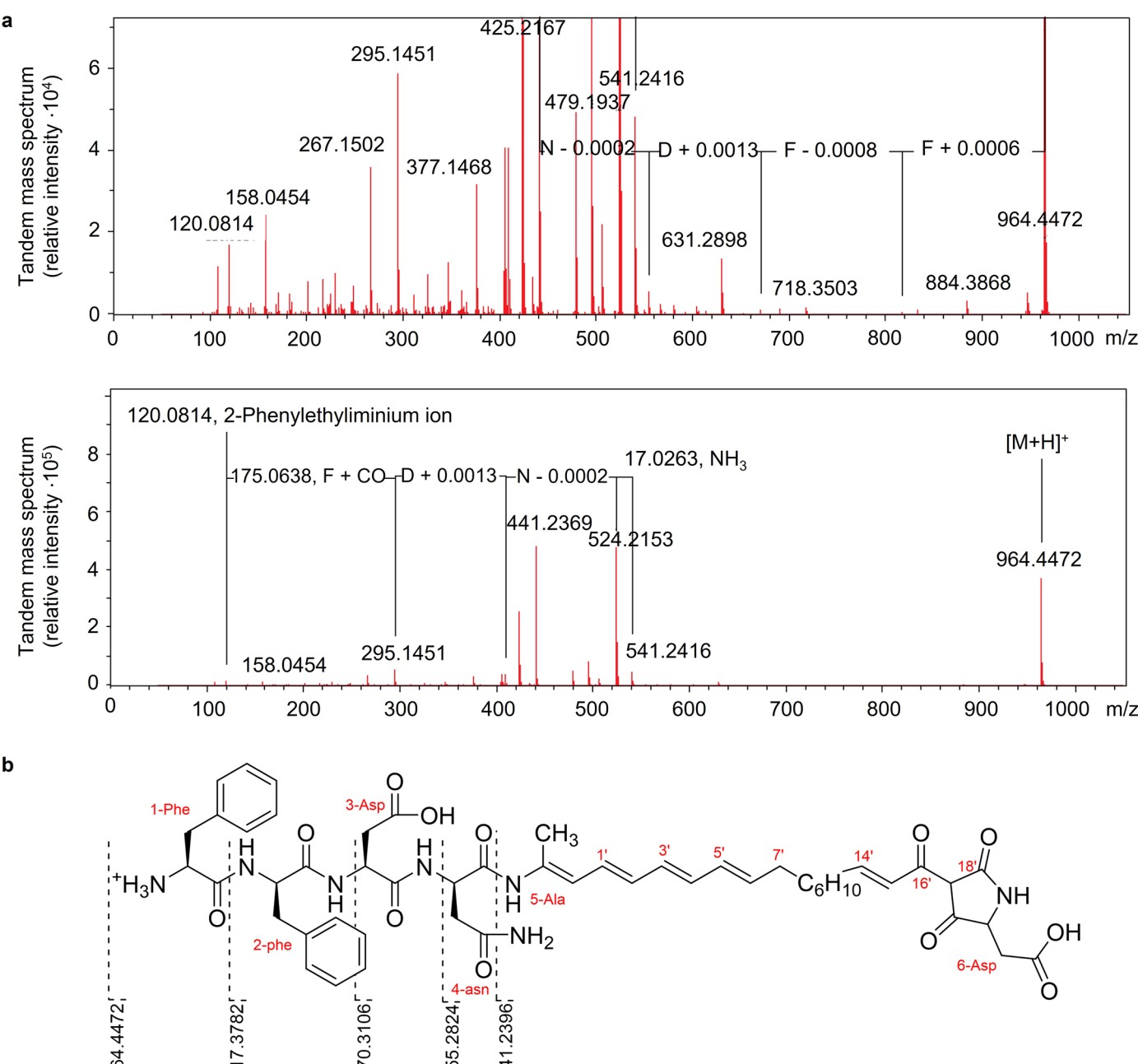

**Extended Data Fig. 6 | MS/MS spectra of epifadin showing fragmentation products from ionization in MS. a**, The mass of 964 Da corresponds to the intact proton adduct (m/z 964.4) of epifadin. 524 Da (m/z 524.2) corresponds to the proton adduct of the tetrapeptide EfiA product, and the mass of 441 Da (m/z 441.2) is assigned to the proton adduct of the EfiBCDE product (expansion shows also minor signals of peptide fragments). [M+H]⁺, monoisotopic positively charged ion; F, phenylalanine; D, aspartate; N, asparagine; CO, carbon monoxide. **b**, The fragmentation pattern for the peptide moiety in epifadin is shown. Numbering of amino acids and carbon atoms of PKS chain in red.

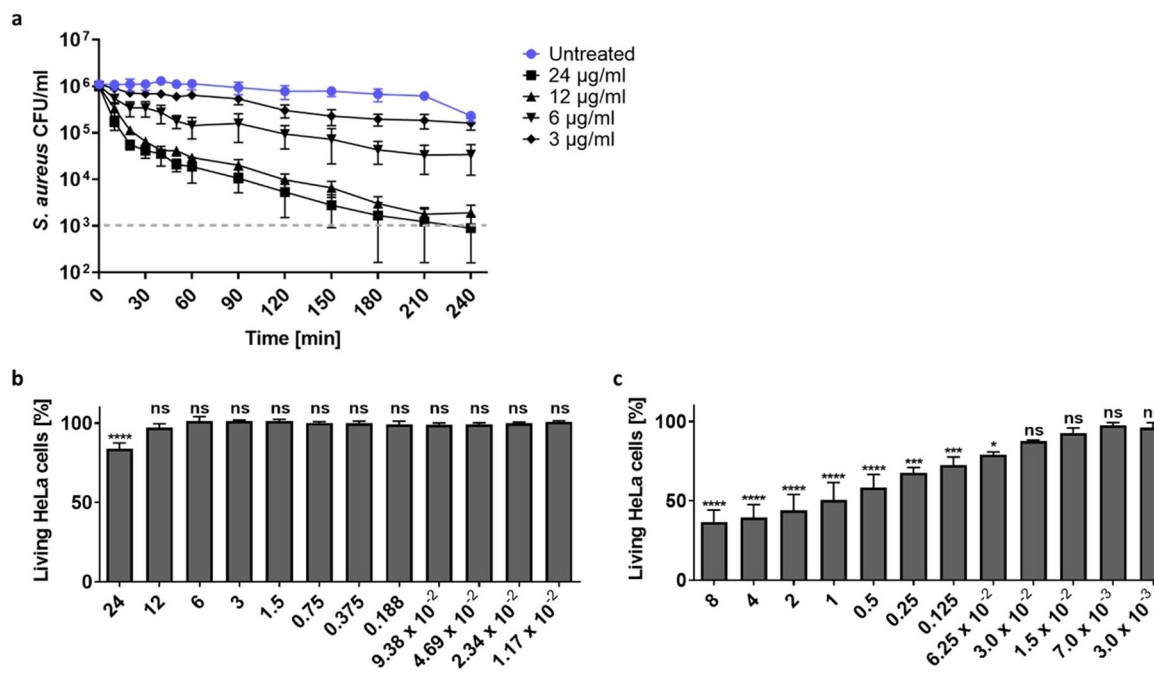

**Extended Data Fig. 7 | Epifadin is bactericidal for susceptible bacterial cells but does not inhibit mammalian cells. a**, Time-dependent elimination of S. aureus by epifadin. Incubation of S. aureus USA300 LAC with epifadin concentrations of 24 µg/mL and 12 µg/mL led to a fast decline of CFUs reaching the detection limit of $1 \times 10^3$ CFU/mL after 210 min. Data represent means with SEM of three independent experiments. **b,c**, Cell viability assay. HeLa cells incubated with epifadin do not show increased cell death compared to mock-treated cells (DMSO treatment set as 100%) even at high concentrations of 12 µg/mL. Cycloheximide (CHM) was included as a positive control. Only at concentration of 24 µg/mL, epifadin shows a significant effect on cell viability, still leaving 84% of HeLa cells intact. Data points represent the mean ± SD of three independent experiments. Significant differences between lowest compound concentrations and higher concentrations were analyzed by one-way ANOVA (*P < 0.05; **P < 0.01; ***P < 0.001; ****P < 0.0001). Exact p values for the CHM treated HeLa cells were 0.0192 ($6.25 \times 10^{-2}$ µg/ml), 0.0009 (0.125 µg/ml), 0.0001 (0.25 µg/ml).

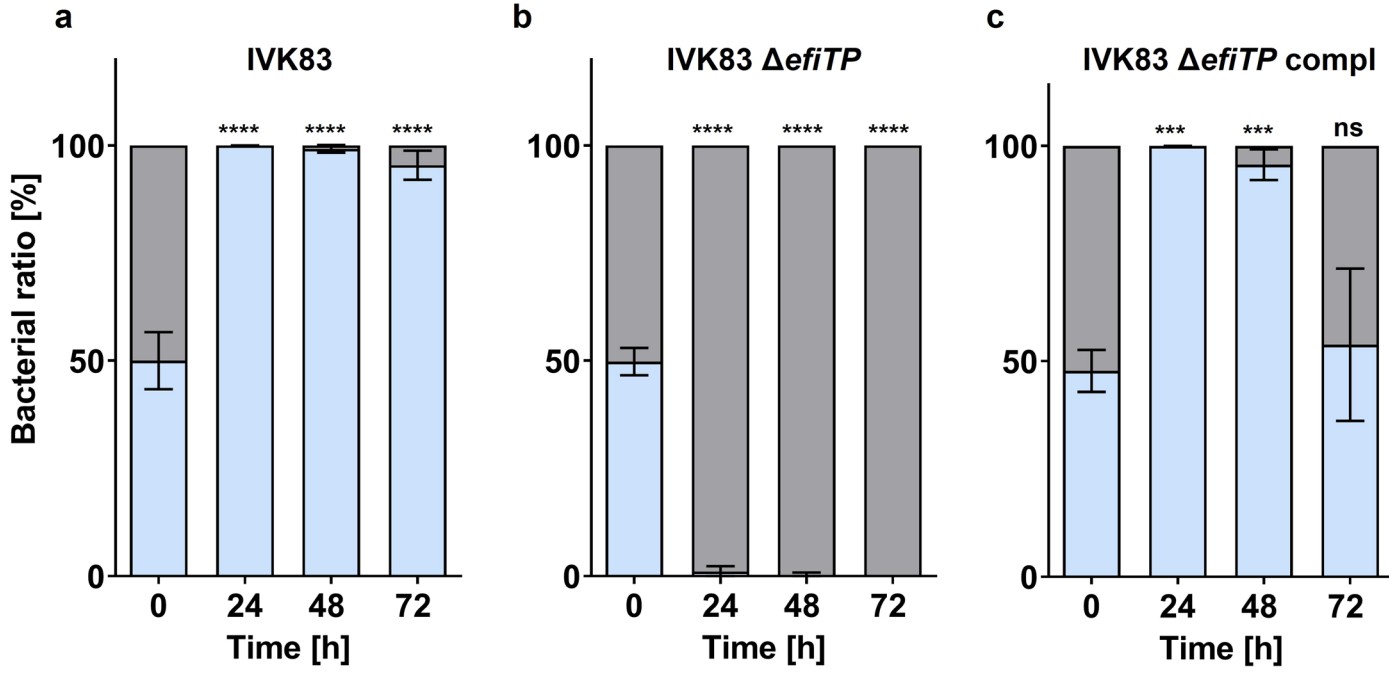

**Extended Data Fig. 8 | Epifadin-producing *S. epidermidis* IVK83 restricts *S. aureus* growth *in vitro*.** **a**–**c**, *in vitro* competition assays in TSB. **a**, *S. aureus* growth is inhibited by IVK83 wild type (grey or light blue bars, respectively) already after 24 h of incubation in TSB inoculated at ratios of ~50:50. **b**, in contrast, the mutant IVK83 Δ*efiTP* is overgrown by *S. aureus* over time when inoculated at a 50:50 ratio. **c**, Complemented strain overgrew *S. aureus* for 48 h, after 72 h, ratio of complemented strain and *S. aureus* were similar to starting conditions. Data points represent mean value ± SD of three independent experiments. Significant differences between the starting condition and the indicated time points were analyzed by one-way ANOVA (**P < 0.01; ***P < 0.001; ****P < 0.0001).

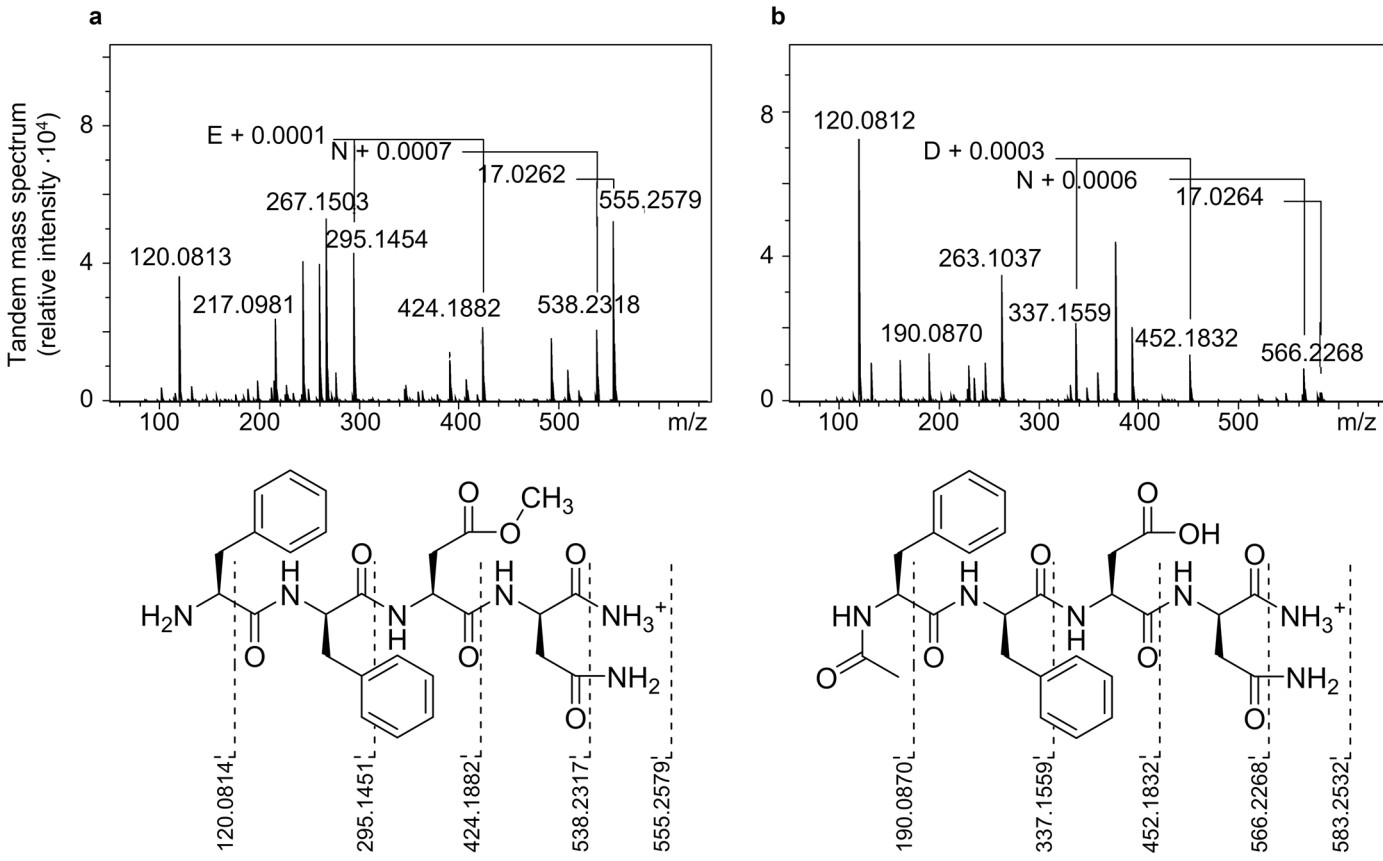

**Extended Data Fig. 9 | Structures of semi-synthetic derivatives of peptide amide and MS/MS spectra. a)** methyl ester of natural peptide amide and **b)** acetylated natural peptide amide.

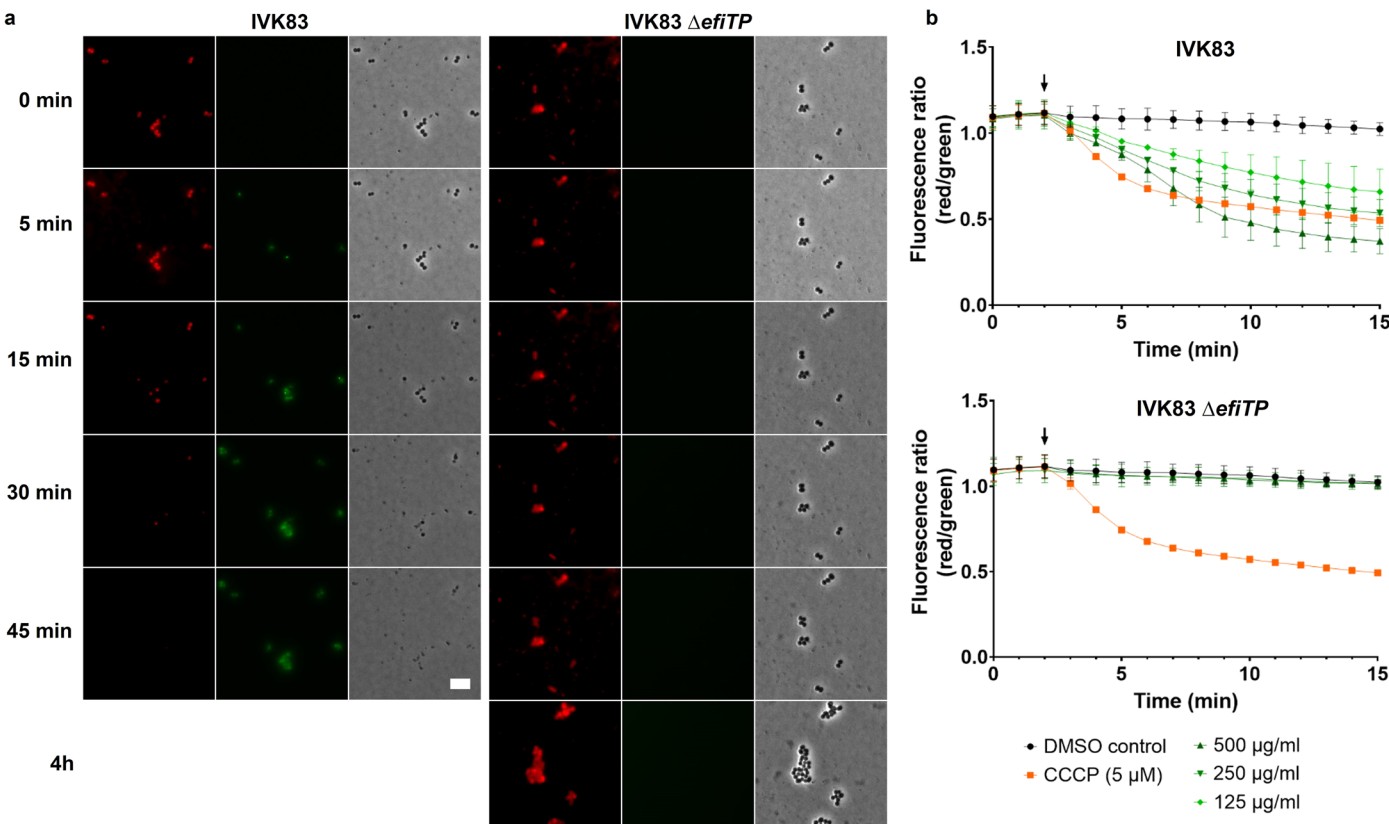

**Extended Data Fig. 10 | Epifadin leads to depolarization of the bacterial membrane and rapid cell lysis of *S. aureus*.** (**a**) *S. aureus* USA300 JE2 cells were applied to an agarose pad, onto which 2 μL of extracts (50 mg/mL) of the epifadin producer IVK83 (left) or Δ*efiTP* (right) had been previously spotted. Image acquisition was started in the surrounding of the respective extract spot 15 min after *S. aureus* application. The agarose contained FM4-64 (red, 0.25 μg/mL, membrane dye) and Sytox Green (green, 0.25 μM, only visible upon membrane barrier malfunction). Representative micrographs are depicted, all adjusted in the Sytox green channel to the same settings for qualitative comparison. White scale bar, 5μm. (**b**) Time-resolved effects of extracts of IVK83 wild type and Δ*efiTP* on the membrane potential of *S. aureus* NCTC8325 as monitored by DiOC$_2$(3) staining. CCCP (5 μM), positive control; DMSO, untreated control. Mean and s.d. of two biological with two technical replicates (n = 4). Black arrow, time point of compound addition.

# Reporting Summary

## Statistics

For all statistical analyses, confirm that the following items are present in the figure legend, table legend, main text, or Methods section.

| n/a | Confirmed | |
|---|---|---|
| ☐ | ☒ | The exact sample size (*n*) for each experimental group/condition, given as a discrete number and unit of measurement |
| ☐ | ☒ | A statement on whether measurements were taken from distinct samples or whether the same sample was measured repeatedly |
| ☐ | ☒ | The statistical test(s) used AND whether they are one- or two-sided<br>*Only common tests should be described solely by name; describe more complex techniques in the Methods section.* |
| ☒ | ☐ | A description of all covariates tested |
| ☒ | ☐ | A description of any assumptions or corrections, such as tests of normality and adjustment for multiple comparisons |
| ☐ | ☒ | A full description of the statistical parameters including central tendency (e.g. means) or other basic estimates (e.g. regression coefficient) AND variation (e.g. standard deviation) or associated estimates of uncertainty (e.g. confidence intervals) |
| ☐ | ☒ | For null hypothesis testing, the test statistic (e.g. *F*, *t*, *r*) with confidence intervals, effect sizes, degrees of freedom and *P* value noted<br>*Give P values as exact values whenever suitable.* |
| ☒ | ☐ | For Bayesian analysis, information on the choice of priors and Markov chain Monte Carlo settings |
| ☒ | ☐ | For hierarchical and complex designs, identification of the appropriate level for tests and full reporting of outcomes |
| ☒ | ☐ | Estimates of effect sizes (e.g. Cohen's *d*, Pearson's *r*), indicating how they were calculated |

*Our web collection on statistics for biologists contains articles on many of the points above.*

## Software and code

Policy information about availability of computer code

| Data collection | No commercial, open source or custom code was used to collect data for this study. |
|---|---|
| Data analysis | antiSMASH 5.0 (bacterial version) was used to analyse the epifadin biosynthetic gene cluster; Whole-genome sequence of S. epidermidis IVK83 was determined by Illumina short-read and PacBio long-read sequencing; Illumina reads were de-novo assembled by velvet (version 1.2.10); Alignment of the two de-novo assemblies with MAUVE (version 2.4.0) and subsequent manual curation allowed us to generate the final genome, which was confirmed by mapping the Illumina reads to the final assembly. The circular chromosome and the plasmid were annotated using the NCBI Prokaryotic Genome Annotation Pipeline (version 5.3) and deposited at NCBI. DNA analysis was performed with DNASTAR Lasergene (version 15); ImageJ software (version 1.8.0_112) was used to measure zones of inhibition for stability analysis of epifadin. GraphpadPrism 8 was used to generate figures and for statistical analyses. For WGS of isolates from experimental evolution, Trimmomatic (v0.39) was used to trim adapters and low-quality bases and read qualities were assessed using FastQC v0.11.7 (https://www.bioinformatics.babraham.ac.uk/projects/fastqc/) and MultiQC v1.0. Genome sequences were assembled de novo and annotated using Unicycler v 0.4.7 with default parameters, using SPAdes v 3.15.4, and Prokka v 1.14.6. |

For manuscripts utilizing custom algorithms or software that are central to the research but not yet described in published literature, software must be made available to editors and reviewers. We strongly encourage code deposition in a community repository (e.g. GitHub). See the Nature Portfolio guidelines for submitting code & software for further information.

## Data

Policy information about availability of data

All manuscripts must include a data availability statement. This statement should provide the following information, where applicable:

- Accession codes, unique identifiers, or web links for publicly available datasets
- A description of any restrictions on data availability
- For clinical datasets or third party data, please ensure that the statement adheres to our policy

All data supporting the findings of this study are available within the paper, its Extended data or Supplementary information. WGS data obtained for S. epidermidis IVK83 were deposited in the NCBI Sequence Read Archive (genome available under accession number CP088002, plasmid pIVK83 under CP088003). Sequence of strain S. epidermidis B155 (Liverpool, UK) was deposited as BioSample SAMEA12384066 (BioProject PRJEB50307). Representative microscopy images are included in the extended data figures and the supplementary videos, which were deposited at Figshare (doi.org/10.6084/m9.figshare.24125589). NMR data were deposited at nmrXive and are available under the project identifier NMRXIV:P18 (10.57992/nmrxiv.p18; https://nmrxiv.org/P18). Source data for experiments is provided.

# Field-specific reporting

Please select the one below that is the best fit for your research. If you are not sure, read the appropriate sections before making your selection.

☒ Life sciences        ☐ Behavioural & social sciences        ☐ Ecological, evolutionary & environmental sciences

For a reference copy of the document with all sections, see nature.com/documents/nr-reporting-summary-flat.pdf

# Life sciences study design

All studies must disclose on these points even when the disclosure is negative.

| | |
|---|---|
| Sample size | Epifadin instability testing, in vitro competition assays, minimal bactericidal concentration assay and cytotoxicity assay were chosen to have a sample size of 3 independent replicates, as this is the minimum number for statistical testing. The sample size of cotton rats used for S. epidermidis IVK83 wildtype and S. epidermidis ΔefiTP nasal colonization was chosen to be 11 animals (5 and 6 per group, respectively), as this number is necessary to determine the median colonization capability of bacterial strains in cotton rat noses. The sample size of cotton rats used for S. epidermidis IVK83 wildtype + S. aureus Newman and S. epidermidis ΔefiTP + S. aureus Newman nasal co-colonization was chosen to be 17 animals (9 and 8 per group, respectively), as this number was sufficient to observe that S. epidermidis IVK83 wildtype is capable to reduce S. aureus nasal colonization in contrast to S. epidermidis ΔefiTP. Here, we kept sample sizes identical or similar to previously published experiments (https://pubmed.ncbi.nlm.nih.gov/27466123) |
| Data exclusions | No data was excluded for any of the experiments. |
| Replication | Replication of epifadin instability testing, in vitro competition assays, minimal bactericidal concentration assay and cytotoxicity assay was performed with n=3 biological replicates and was successful each time. Epifadin inhibitory concentration determination was performed once, but in parallel with all strains, due to the limited availability of purified epifadin. Membrane depolarisation assays were performed with n=2 biological replicates with n=2 technical replicates, each. |
| Randomization | Randomization was only partially conducted since female cotton rats are usually kept in groups of two or three animals per cage. All animals from one cage had to be colonised with the same bacterial strain, or strain-combination. All male animals were kept alone in individual cages and were randomly colonised. |
| Blinding | Animal experiments could be performed only by two experimenters who had to work together, for which reason blinding in the animal facility was not possible. Sample plating was performed by another person who was not involved in the colonisation. Colony counting was performed by two methods, a semiautomatic camera-based, which resulted in absolute colony numbers, and a manual counting which distinguished between S. aureus and S. epidermidis. This differentiation between colony morphologies was totally objective. |

# Reporting for specific materials, systems and methods

We require information from authors about some types of materials, experimental systems and methods used in many studies. Here, indicate whether each material, system or method listed is relevant to your study. If you are not sure if a list item applies to your research, read the appropriate section before selecting a response.

## Materials & experimental systems

| n/a | Involved in the study |
|-----|----------------------|
| ☒ | Antibodies |
| ☐ | Eukaryotic cell lines |
| ☒ | Palaeontology and archaeology |
| ☐ | Animals and other organisms |
| ☒ | Human research participants |
| ☒ | Clinical data |
| ☒ | Dual use research of concern |

## Methods

| n/a | Involved in the study |
|-----|----------------------|
| ☒ | ChIP-seq |
| ☒ | Flow cytometry |
| ☒ | MRI-based neuroimaging |

# Eukaryotic cell lines

Policy information about cell lines

| | |
|---|---|
| Cell line source(s) | Human cervical carcinoma HeLa cell line (ATCC) |
| Authentication | Cell line was initially purchased by ATCC. The cell line was recently analysed by the DSMZ via DNA profiling using 17 different and highly polymorphic short tandem repeat (STR) loci. This confirmed the identity as HeLa without any doubt. |
| Mycoplasma contamination | Cell lines were not tested for mycoplasma contamination |
| Commonly misidentified lines (See ICLAC register) | In this study no commonly misidentified cell lines were used. |

# Animals and other organisms

Policy information about studies involving animals; ARRIVE guidelines recommended for reporting animal research

| | |
|---|---|
| Laboratory animals | In this study, cotton rats (Sigmodon hispidus) of both sexes, 8-12 weeks old, were used. |
| Wild animals | No wild animals were used in this study. |
| Field-collected samples | The study did not involve samples collected from the field. |
| Ethics oversight | All animal experiments were conducted in strict accordance with the German regulations of the Gesellschaft für Versuchstierkunde/ Society for Laboratory Animal Science (GV-SOLAS) and the European Health Law of the Federation of Laboratory Animal Science Associations (FELASA) in accordance with German laws after approval by the local authorities (IMIT 1/15, Regierungspräsidium Tübingen) |

Note that full information on the approval of the study protocol must also be provided in the manuscript.

