## [Peer Review File · Nature Microbiology]

Peer Review Information

Journal: Nature Microbiology

Manuscript Title: Commensal production of a broad spectrum and short-lived antimicrobial peptide polyene eliminates nasal *Staphylococcus aureus*

Corresponding author name(s): Dr Bernhard Krismer

Reviewer Comments & Decisions:

Decision Letter, initial version:

Message: 7th April 2022

Dear Dr. Krismer,

Thank you for your patience while your manuscript "Extremely short-lived peptide-polyene antimicrobial enables nasal commensal to eliminate *Staphylococcus aureus*" was under peer-review at Nature Microbiology. It has now been seen by 4 referees, whose expertise and comments you will find at the end of this email. Although they find your work of some potential interest, they have raised a number of concerns that will need to be addressed before we can consider publication of the work in Nature Microbiology.

In particular, reviewer #1 asks you to search for epifadin BGC in human skin and nasal metagenomic datasets and to consider epifadin stability in human physiological conditions. Reviewer #2 asks you to provide additional information on the basis of differential susceptibilities to epifading of *S. epidermidis* strains, and to potentially extend the co-culture experiments to determine resistance suppressor mutants. Reviewer #3 raises issues regarding the elucidation of epifadin structure by NMR, pointing at the need to strengthen it and provide more detailed bioinformatic data. This reviewer also criticizes the microbiological aspect of the study as not being fully conclusive. Reviewer #4, in agreement with reviewers #1 and #2, comments on the lack of experimental evidence for the mechanism of action of epifadin, and suggests to perform experiments in order to determine the target of epifadin (for example, obtain resistant mutants and perform WGS). Should further experimental data allow you to address these criticisms, we would be happy to look at a revised manuscript.

We strongly support public availability of data. Please place the data used in your paper into a public data repository, if one exists, or alternatively, present the data as Source Data or Supplementary Information. If data can only be shared on request, please explain why in your Data Availability Statement, and also in the correspondence with your editor. For some data types, deposition in a public repository is mandatory - more information on our data deposition policies and available repositories can be found at <https://www.nature.com/nature-research/editorial-policies/reporting->

2standards#availability-of-data.

Please include a data availability statement as a separate section after Methods but before references, under the heading "Data Availability". This section should inform readers about the availability of the data used to support the conclusions of your study. This information includes accession codes to public repositories (data banks for protein, DNA or RNA sequences, microarray, proteomics data etc...), references to source data published alongside the paper, unique identifiers such as URLs to data repository entries, or data set DOIs, and any other statement about data availability. At a minimum, you should include the following statement: "The data that support the findings of this study are available from the corresponding author upon request", mentioning any restrictions on availability. If DOIs are provided, we also strongly encourage including these in the Reference list (authors, title, publisher (repository name), identifier, year). For more guidance on how to write this section please see: <http://www.nature.com/authors/policies/data/data-availability-statements-data-citations.pdf>

* If you have not done so already we suggest that you begin to revise your manuscript so that it conforms to our Article format instructions at <http://www.nature.com/nmicrobiol/info/final-submission>. Refer also to any guidelines provided in this letter.

When submitting the revised version of your manuscript, please pay close attention to our [href="https://www.nature.com/nature-research/editorial-policies/image-integrity">Digital Image Integrity Guidelines](https://www.nature.com/nature-research/editorial-policies/image-integrity) and to the following points below:

2Please use the link below to submit a revised paper:

[redacted]

Note: This url links to your confidential homepage and associated information about manuscripts you may have submitted or be reviewing for us. If you wish to forward this e-mail to co-authors, please delete this link to your homepage first.

Nature Microbiology is committed to improving transparency in authorship. As part of our efforts in this direction, we are now requesting that all authors identified as 'corresponding author' on published papers create and link their Open Researcher and Contributor Identifier (ORCID) with their account on the Manuscript Tracking System (MTS), prior to acceptance. This applies to primary research papers only. ORCID helps the scientific community achieve unambiguous attribution of all scholarly contributions. You can create and link your ORCID from the home page of the MTS by clicking on 'Modify my Springer Nature account'. For more information please visit www.springernature.com/orcid.

If you wish to submit a suitably revised manuscript we would hope to receive it within 6 months. If you cannot send it within this time, please let us know. We will be happy to consider your revision, even if a similar study has been accepted for publication at Nature Microbiology or published elsewhere (up to a maximum of 6 months).

Yours sincerely,

[redacted]

Reviewer Expertise:

Referee #1: nasal microbiome, bacterial interactions
Referee #2: Staphylococcus biology, interactions
Referee #3: natural microbial products as antimicrobials, microbial interactions
Referee #4: natural microbial products as antimicrobials

Reviewer Comments:

Reviewer #1 (Remarks to the Author):

This is a well written manuscript supported by well-designed experimentation that describes the detection, purification, estimation of the structure, and the functional analysis of a new antimicrobial encoded on a plasmid present in some strains of the common human nasal and skin microbiont *Staphylococcus epidermidis*. Epifadin is a new peptide-polyene-tetramic acid molecule and has a broad range of antimicrobial activity but with variable

3

strain level susceptibility among many of the genera and species tested (Fig. 5a). This testing of a broad phylogenetic range of species is a strength of the analysis. Epifadin exhibits instability under standard laboratory conditions, e.g., light and pH 7. However, it has increased stability under the physiological acidic and temperature conditions of the human skin and nasal vestibule, primary colonization sites of *S. epidermidis*. The competition assays demonstrate that an intact epifadin BGC is required for *S. epidermidis* IVK83 to out compete *S. aureus* Newman during cocultivation in vitro (including the wt, an efiTP deficient mutant and a complemented efiTP mutant) and cocolonization in vivo in a cotton rat model. There are an appropriate number of independent experiments and standard deviation (or standard error of the mean) is indicated for most graphs along with appropriate statistical testing. This work will be of broad interest to both microbiome, microbe-microbe interaction and chemical biology researchers. However, there are several issues that if addressed would strengthen the manuscript.

Suggestions for the authors.

1. How often is the epifadin BGC detected in existing, and publicly available, skin and nasal metagenomic datasets? Detection of the epifadin BGC in these metagenomic datasets would strengthen the argument that this molecule plays a role in shaping microbiota composition on humans and would increase the impact of this manuscript.

2. In lines 94-96, was a search performed to detect the epifadin BGC among all existing bacterial genomes, rather than only among *Staphylococcus* species?

3. Regarding Figure 2, line 125 and lines 264-265, pH 7 and 37°C are inconsistent with in vivo conditions in the human nasal vestibule or on human skin where both the pH and temperature are lower, e.g., pH ~5.5-6.5 and temperature range 30-34°C. In this respect, epifadin stability may be tuned for the conditions where *S. epidermidis* primary resides, since *S. epidermidis* is broadly distributed across human skin sites as well as in the human nasal passages. Revised wording in both Figure 2 legend and in the text to more accurately reflect human physiology is recommended.

4. Regarding Figure 6e and lines 227-229, please comment on the apparent bimodal distribution of the percentage of *S. aureus* recovered from the cotton rat nose with wt *S. epidermidis* IVK82. Does this suggest some other mode of competition between the two inoculated species during colonization? Also, this graph would benefit from a box and whiskers plot based on the spread of data points for *S. aureus* + IVK83.

5. Given the unusual structure of epifadin and the identification of both susceptible and resistant isolates of *S. epidermidis*, are there clues to the mechanism of action? (This may be beyond the scope of the current manuscript.)

6. Line 209: A more cautious interpretation of epifadin toxicity to HeLa cells at 24 ug/ml seems appropriate given that topical use would require demonstration of lack of toxicity to that skin and nasal epithelial cells and systemic administration would require a more thorough assessment of potential toxicity than HeLa cells.

Minor suggestions

1. Line 244: Given the unusual structure of epifadin with the "alternation of a polar peptide,

a non-polar polyene, and, again, a polar and charged amino acid-derived building block" is it possible that that target in bacteria differs from the target in fungi?

2. Line 246: Is the reduced susceptibility of HeLa cells compared to the two yeast tested related to stability of epifadin under the differing cultivation conditions, i.e., was the pH in the yeast culture lower increasing epifadin's stability?

3. Is the IC listed in Fig. 5b the minimal inhibitory concentration? If yes, it would be clearer to list as MIC rather than IC.

4. Please correct the typo in Figure 5a which should read *Corynebacterium pseudodiphtheriticum* with an h after the first t.

5. The *S. epidermidis* strain info is missing on panel C of extended data Fig. 8.

Reviewer #2 (Remarks to the Author):

This important paper describes a new class of antimicrobial combining a non-ribosomally synthesised peptide and a polyketide called epifadin. Epifadin is produced by *S. epidermidis* strain IVK83 and has broad spectrum activity against *S. aureus* and many other bacteria, but a very short half life. The authors propose that this represents not only a new type of antimicrobial, but also a novel strategy for some *S. epidermidis* strains to eliminate skin and nasal microbiome competitors nearby without impacting potentially important bacterial mutualists & host cells further away.

Using transposon mutagenesis, the biosynthetic gene cluster responsible for the antimicrobial activity of IVK83 was identified on a plasmid. This cluster and plasmid was also identified in several other *S. epidermidis* isolates from diverse geographical locations. Interesting it is absent in submitted *S. epidermidis* genomes in the databases, indicating that it is an uncommon accessory element. Isolation of the antibacterial activity proved difficult and the authors are to be commended on progress in this area and showing that the compound has an extremely short half life. An elegant chemical characterisation of the unique structure of epifadin is described.

Using the small amounts of available purified epifadin under controlled conditions demonstrated its broad antibacterial activity.

This is an important and well written manuscript that provides novel insights into interbacterial interactions in the nasal microbiome and the identification of a novel antimicrobial agent with a short half life. An impressive body of research is presented and the work is suitable for publication in *Nature Microbiology*. Perhaps the genetics of epifadin biosynthesis could have been developed further but this is a minor quibble, particularly given that manipulation of a >40Kb gene cluster is technically challenging in *S. epidermidis*. As referred to in the discussion, analysis of the biosynthetic gene cluster did not reveal any insights on the mechanism of action, but of interest there were differences in susceptibility among the *S. epidermidis* nasal isolates tested, with strain 9/16 being completely uninhibited by epifadin (Fig. 5), while the authors also indicate that 5 isolates of *S. epidermidis* are susceptible (line 194). Can the authors provide any additional insights on the basis for these different susceptibilities based on genomic/phenotypic comparisons between resistant and susceptible isolates? Future studies to identify the mechanism of

5

resistance will be important. Did the authors consider extension of the co-culture experiments between IVK83 and a susceptible *S. epidermidis* isolate (perhaps a rifampicin resistant derivative) to determine if resistant suppressor mutants can be isolated?.

Reviewer #3 (Remarks to the Author):

Comments to Authors

Manuscript number: NMICROBIOL-22020429

Manuscript Type: Article

Title: Extremely short-lived peptide-polyene antimicrobial enables nasal commensal to eliminate *Staphylococcus aureus*.

Overview and general opinion

This manuscript reports the discovery of a novel antibiotic produced by the common human nose isolate *S. epidermidis* IVK83, which drew the attention of the research group by displaying strong antibacterial activity against many strains from their bacterial collection. By way of a transposon mutagenesis, this bioactivity could be traced to a hybrid PKS-NRPS biosynthetic gene cluster. The associated antibacterial compound, named epifadin, is highly unstable; nevertheless, the authors deduced the structure of epifadin by a combination of HRMS, MS/MS, NMR, tetrapeptide synthesis, and bioinformatic analysis. They also proposed a biosynthetic pathway based on the structure and bioinformatics. Epifadin has broad range antibacterial activity, including against a number of potential competitors of the producer. Among them, the strongest activity is observed against the facultative pathogen *S. aureus*, with only weak toxicity observed against human HeLa cells. Experiments using *S. epidermidis* IVK83 and an epifadin non-producing strain showed that epifadin production enables *S. epidermidis* to outcompete *S. aureus* in vivo. The authors propose that the short half-life of epifadin avoids collateral damage to beneficial members of the microbiota, thereby helping to maintain balance of the nasal microbiota.

Overall, I believe that the new information described in the manuscript is significant given our limited understanding of competitive interactions among members of the human microbiota. I believe the manuscript presents a story that would be of interest to the readership of Nature Microbiology. However, it suffers from two major flaws that mean I cannot recommend this manuscript for publication in Nature Microbiology in its current state (further elaborated below):

1. While the storyline is well-organized, the manuscript would benefit from extensive editing, and some paragraphs are difficult to understand and/or are missing detailed explanation.
2. The structure determination and bioinformatic parts are weak so could be improved or toned down in favour of the biological significance.
3. The microbiology part and the biological activities are not fully conclusive.

Major Comments

Structural Elucidation/Bioinformatics

Unfortunately, the structure of epifadin is not yet elucidated completely by NMR spectral data because of instability. The authors showed only the N-terminal tetrapeptide unit clearly, while the other parts are deduced by MS and bioinformatic analysis. The authors emphasize that epifadin is novel and the first hybrid NRP-PK from a member of the human

6microbiota. Thus, more precise structure elucidation using NMR is pivotal to support the structure. Various highly instable compounds have already been purified and measured by NMR. The authors could use either 600 or 700 MHz NMR (with a cryoprobe), so that authors consume 0.5 mg per measurement at $-20\text{ }^{\circ}\text{C}$. It would not be essential to show pure NMR spectra, but the authors have to identify the main signals.

At least in page 7, line 162, authors describe "... ^{13}C NMR...tetramic moiety", but the ^{13}C NMR spectrum is not shown. Although the authors explain the proposed structure and biosynthetic pathway, there are no detailed bioinformatic data including adenylation domain codes. KR domain fingerprinting analysis also may show geostereochemistry of double bonds, but the authors do not explain the reason why all double bonds are trans configurations (perhaps ROESY correlations, but it is not explained at all).

From a structural point of view, the combination of three different units may be new, but each moiety is not really new. Therefore, the authors should show other groups' compounds to highlight this. For example, reutericyclin/mutanocyclin, which have been isolated from *Lactobacillus* in human gut/nasal/oral cavity, possess a tetramic moiety and an acyl chain. Also, mutanofactin has been isolated from human oral *Streptococcus* and consists of an unsaturated fatty acyl chain and peptide. These examples are only from human microbiota so there could be other similar compounds known.

The authors suggest that the tetrapeptide amino acid sequence is L-Phe-D-Phe-L-Asp-D-Asn, but the 4 NRPS modules of EfiA indicate D-Phe-L-Phe-D-Asp-L-Asn. The first module has a sequence of domain A-C-T-E, which is bit unusual because the usual domain sequence is A-T-E-C. The downstream 3 modules are normal type, but judging from this module composition, D-L-D-L is a true sequence.

The obtained shunt tetrapeptide does not have antibacterial activity. However, the scenario cannot be excluded that all three units are important to pass the cell membrane. The authors should discuss. An L-D-L-D peptide very likely makes a helix like gramicidin. For example, after epifadin enters into cell, tetrapeptides could be activated by aggregation. Has a tetramic acid moiety alone been reported to have any antibacterial activity?

Microbiology

To strengthen the relevance of the results, the authors should make clear the scale of the problem of *S. aureus* as an opportunistic pathogen and risk of *S. aureus* infection.

The authors are experts of antibacterial activities, but readers of this journal come from many different fields, including both students and scientists. Thus, authors must better explain how epifadin is a potent antibiotic. In Figure 5, vancomycin and daptomycin are used as famous and clinical antibiotic controls. However, what about a comparison to lugdunin, which has been reported by the authors' group? This peptide also inhibits the growth of *S. aureus* and could be further discussed from an ecology perspective.

The rat model should be briefly explained for the unfamiliar reader. For example, I assume it is not germ-free, which would mean that it could be said to replicate a somewhat real-world habitat. This would mean that the bioactivity shown in this section is perhaps the most convincing of all the results of this section could be "hyped up" accordingly. Can the authors also comment on the apparent "split" in efficacy of *S. aureus* inhibition by the WT, as observed in Fig. 6e? Could it be that the microbiota of some individuals differs meaning that the compound is "mopped up" by other susceptible organisms, thereby being diverted from affecting *S. aureus* so profoundly? But of course, this would not be seen in this experimental set-up seeing as microbiota-level dynamics weren't tracked (the overall composition is unknown), just the competing *S. epidermidis* and *S. aureus* strains.

7

Could the authors comment on whether the stability of epifadin could be better in vivo in some way i.e. is possible that the in vivo environment is less harsh? I know of at least one case where a natural product (streptolysin S) cannot be extracted in a bioactive (and presumably intact) form without the presence of carrier molecule, and it has been proposed that in vivo this problem is overcome by the toxin being cell-bound until it is deposited upon the surface of target cells by direct contact.

I find interesting the idea put forward by the authors that the highly instable nature of the compound might minimize collateral damage. This is one of the big impacts of the report but is not shown so can the authors at least speculate how it would be possible to provide support for this (future directions)?

Minor comments

Main Text

Please unify "-" and "-" throughout.

L22 and throughout: Should be microbiota (all the microorganisms that are found within a specific environment) not microbiome (the collection of genomes from all the microorganisms in a given environment).

L24: This sentence is phrased strangely – sounds like separate things. Maybe change to "It has an unprecedented architecture consisting of a non-ribosomally synthesized peptide component, a polyketide component, and a terminal modified amino acid moiety."

L35: Insert reference(s) (review on the topic?) to support "crucial roles".

L38: Unnecessary comma after "both".

L42: "S. epidermis can...competitors" Recommend to split sentence.

L51: In what way is bacteriocin production beneficial? Presumably when competitors are eliminated.

L60: Change to "lifetimes".

I would assume "distance from" rather than "distance to" since bacteriocins originate from the producer and diffuse away.

Also, why not producer in general rather than "producing micro-colony"? If this is a technical term, explain since "colony" makes me think of in an artificial lab set-up.

L64: "Such a strategy can only work if..." Explain why this is the case as not obvious.

L65: Reference to support "...which is usually the case in skin..."

L69: If it is the first example of this strategy, wouldn't it be unique so far, not "unusual"? I would delete as "previously unrecognized" is sufficient.

L68: Suggest replacing "reflecting" with "representing".

L69: I would rather say "first discovered".

L71: Should be NRPSs and PKSs.

L72: Suggest "their shared habitat".

I would not consider "in vitro" a habitat (i.e. the natural home of an organism). I would separate these points – shown both in a lab-based setup and, more significantly, in vivo.

L81: Highlight results by beginning sentence as follows "We determined that the transposon had..."

Is *S. aureus* USA300 an MRSA strain? If so, probably of interest to state.

L82: Suggest to delete "operon" as superfluous.

L83: Why did you deem this unusual? Hybrid NRPS-PKS biosynthetic gene clusters encode NRPSs and PKSs. Is a set of NRPS-transAT iterative PKS-tetramic acid forming NRPS unusual?

L84: Suggest insertion "...set of putative biosynthetic enzymes, namely a putative NRPS

(EfiA)...” Also, I would use “hybrid PKS-NRPS” (enzymes) or “hybrid PK-NRP” (natural product) here and throughout since this is standard in the literature; important also to hyphenate as “/” suggests “or”.

L85: Suggest “biosynthesis”.

L91: What is meant by screening? First identified those with BGC then checked for similar antimicrobial activity?

L94: More accurate to describe something like “Homologous BGCs were absent from all *S. epidermidis* genomes...”

L100: Suggest replacing “contained” with “possessed” or “displayed”.

L110: Suggest replacing “faded” with “diminished”.

L114: Edit for clarity. “Reversed-phase high-performance liquid chromatography with ultraviolet light detection (RP-HPLC-UV) of DMSO-PA extracts of *S. epidermidis* IVK83 and *S. epidermidis* IVK83 Δ efiTP, obtained from acid-precipitated culture supernatants, revealed no obvious difference between the IVK83 wild type and efiTP mutant strain at the peptide-bond specific wavelength of 215 nm, although the NRPS-encoding genes for NRPS pointed to an antimicrobial product with the presence of multiple peptide bonds.” Also recommend to split sentence.

L118: Edit for clarity. “However, the HPLC profile of the wild type and mutant differed in displayed a major and in some adjacent minor peaks at 383 nm that were absent in that of the mutant, indicating a product possessing the presence of an expanded unsaturated system with double bonds (Figs. 3a-c).”

L129: Suggest insertion. “...predicted the product of the cluster to have a three-partite composition”.

L134: Re-phrase to clarify if this means there were multiple purifications from one 100 L culture, or multiple 100 L cultures?

L141: Remove unnecessary comma after “Marfey`s analysis”.

L151: Split sentence. “Two-dimensional NMR ... Extended Data Fig. 4a).”

L177: Suggest replacing “combines” with “is derived from a combination of”.

L 179: “...multidimensional NMR analyses...”, but Extended Figure 3 and 4a shows only ROESY and ¹H NMR of purified epifadin, respectively. It seems that none of the NMR spectra of the synthetic tetrapeptide are shown elsewhere.

L181: “Moreover... instability of epifadin.” Recommend sentence as better fit to the Discussion.

Consider replacing “Moreover” with “In the future”.

L182: Suggested simplification. “which will require further efforts though and remains challenging...”

L183: Incorrect usage of “By all means”. Suggest changing as follows: By all means Interestingly, we demonstrate that epifadin is the first...and is the founding member...”.

L187: Wild type and mutant strains are not referred to consistently e.g. wild type variously written as *S. epidermidis* IVK83, IVK83, IVK83 wild type; mutant as efiTP mutant strain, Δ efiTP, *S. epidermidis* Δ efiTP, IVK83 Δ efiTP. For simplicity, always give the full names *S. epidermidis* IVK83 and *S. epidermidis* IVK83 Δ efiTP, with the occasional use of wild type and mutant when clear by context (e.g. full names used in previous sentence).

L189: This sentence should be moved to the paragraph starting L202 for clarity, i.e. keep producer/purified compound experiments separate.

L196: Consider “...nystatin A1 contain similar polyene moieties similar to that of as epifadin”

- L204: Consider "...MIC values (between 3.7 and 8.6 µg/mL)..." for ease of reading.
- L208: "Epifadin targets..." Sentence seems out of place as no context for the point is given.
- L209: 20 times which MIC? Unclear.
- L210: Consider "...suggesting thea lack...".
- L213: Consider "...raised the question ifof whether..."
- L226: Consider "...nasally instilled into the noses..."
- L228: Split sentence e.g. "... (Fig. 6e),. In this way, we could demonstrate indicating that epifadin-producing *S. epidermidis* can effectively interfere with *S. aureus* nasal colonization.
- L235: Unnecessary comma. "...both, NRP and PK moieties..."
- L236: "new compound class" – name it.
- L239: "...rely on..." For what?
- L242: Use of "again" is confusing as there has not been a prior single polar charged amino acid.
- L243: Redundant as implied already by "remains". "...remains to be elucidated in the future,..."
- "...the molecular target appears to be shared by bacterial and fungal cells..." How do you conclude that? Since there is antibiotic activity towards both? I think too speculative as, in theory, it could have a different target in both to exert activity.
- L246: How do you know it is taken up? Could act at the membrane for example. If this is based on the known behaviour of similar molecules, provide reference(s).
- L263: It might initially At first thought, it may seem paradoxical and a waste of energy for a bacterium to waste energy to by produceing an antimicrobial with such a short lifespan.
- L315 and throughout: Should be "...10 µl werewas..." since it is a collective noun so counted as a single entity.
- L359: Fix typo. "...the genes Δ efiTP were amplified..."
- L394: Fix typo. "Sodium formiate ..."
- L471: How were the CFUs of the two strains differentiated? Later described but should be mentioned here too.
- L485: "...inoculum of 1×10^7 CFU/nose isare required..."

Figures

- Figure 1: Increase font sizes in Part C.
- L694: "DMSO extract..." Split sentence.
- Figure 3: The quality of chromatograms is not high enough.
- Figure 4: Peak labels difficult to read in Part A. Keep font-size consistent throughout all figures.
- Figure 5: Strain names should not be italicized in Part B. Addition of MIC values in µM (standard) would allow direct comparison to potency of controls.
- Extended Figure 5: The quality of structures is not high enough.
- Extended Data Figure 7: Is the last bar in Panel b&c an untreated control?
- Extended Data Figure 8: Label above barchart missing in Part C
- Extended Figure 9: The quality of mass spectra is not high enough.

Supplementary

- In general, some amino acid stereochemistry did not use lower case (eg. line 42: L-Ala). Please unify them.
- L24: Missing abbreviation of PA in DMSO-PA, even if it is described in a main text.

10

Reviewer #4 (Remarks to the Author):

General assessment:

The manuscript "Extremely short-lived peptide-polyene antimicrobial enables nasal commensal to eliminate *Staphylococcus aureus*" presents the discovery of the NRPS-PKS based peptide-polyene epifadin from the nasal commensal *S. epidermidis* IVK83. Epifadin features not only moderate antimicrobial activity against numerous pathogens including *Candida albicans*, *Saccharomyces cerevisiae* and different *S. aureus* strains but also interesting physiochemical characteristics like its unique pentapeptide polyene tetramic acid scaffold and the extraordinary instability of this natural product class. Due to this instability, Epifadin was purified from *S. epidermidis* IVK83 culture supernatant under acidic conditions with argon atmosphere, by preparative RP-HPLC in the dark and fractionation at -77°C . In addition, the authors investigated not only the abundance of the identified BGC in different genome sequences but also propose the biosynthetic pathway leading to the formation of this new natural product class. The corresponding BGC was identified previously and published including a knock-out. The key finding of the current manuscript is the structure elucidation which is indeed an important milestone to understand the consequences of epifadin formation in the microbiota.

The laboratory work and manuscript are of good quality and finally provide a new perspective of natural product formation in the human microbiome. The authors did not only describe a new compound class from *S. epidermidis* IVK83 in the context of environmental and host-associated bacterial communities, but also put a lot of effort to obtain this highly instable natural product in contrast to previous discoveries of new microbiome-derived stable compounds. The herein presented study highlights an interesting concept for the exploration of new antimicrobials from nasal commensals, which might also support the discovery of new bioactive natural products from conventional habitats. Unfortunately, the authors were not able to reveal or provide any experimental information considering the mode-of-action of epifadin. Therefore, it is highly recommended to perform at least an additional experiment to obtain in vitro Epifadin-resistant mutants (see below). Furthermore, the authors could provide more information considering a potential molecular target of Epifadin.

Nevertheless, the manuscript should become suitable once the issues raised are addressed for Nature Microbiology, since it provides new insights to the microbial production of short-lived natural products and is interesting for the broad community of scientist dedicated to the exploration of natural products from the human microbiome.

Major suggestions:

Is it possible to generate spontaneous resistant mutants of *Staphylococcus aureus* USA300LAC/JE2 (or another suitable sensitive strain) in vitro by applying lower antibiotic pressure and repeated passaging of incrementally resistant colonies on rising Epifadin concentrations? Subsequent genomic DNA isolation and whole-genome sequencing from all spontaneous resistant mutants could help to reveal the molecular target of Epifadin; and even if the mutation would only result in the overexpression of an ABC transporter (which seems also to be the encoded self-resistance mechanism of the producer strains), this outcome might still be useful for future investigations.

11

P8/9 L194–195

“Notably, all tested *S. aureus* strains were susceptible to epifadin, whereas 9 of 16 tested nasal *S. epidermidis* isolates were resistant.”

Any idea, what is the genetic difference between an Epifadin-resistant and non-resistant *S. epidermidis* isolate? Maybe the presence of distinct ABC transporters? Please comment and provide more information to this observation.

P8/9 L248–250

“However, the epifadin BGC encodes only the two ABC transporter components EfiFG as potential resistance proteins, which may act as drug exporters and do not shed light on the antibacterial target”

In order to prove this statement experimentally, it would be interesting to generate in addition to the mutant *S. epidermidis* IVK83 Δ efiTP, the mutant *S. epidermidis* IVK83 Δ efiFGTP. Subsequent comparison of both Epifadin-nonproducers against the epifadin-producing isolate *S. epidermidis* IVK83 could reveal the role of both genes.

Minor suggestions:

Title

Have you considered implementing the name of the natural product in the title e.g.

“The extremely short-lived peptide-polyene antimicrobial epifadin enables nasal commensal to eliminate *Staphylococcus aureus*”

Abstract:

P1, L24:

“...*Staphylococcus epidermidis*.” Could you please add the exact strain name

“*Staphylococcus epidermidis* IVK83?”

Introduction:

P3, L45-46 and L47:

“---- and may impair the fitness of its major competitors.” and “....activity against potential target species”.

Could you please mention one or two relevant examples of major competitors and target species (Probably *S. aureus* as mentioned on P4, L72) ?

P3, L67:

“Here we present a novel bacteriocin-like antimicrobial, epifadin.....”

This statement is ambiguous, since the bacteriocins are RiPPs and epifadin is a natural product produced by a hybrid NPRS-PKS assembly line. Furthermore, the structural architecture of epifadin is clearly distinct from typical RiPPs. Only the biological function of epifadin is similar to previously described bacteriocins. Please change the wording accordingly.

Result:

P6, L123:

“This formula was unknown in scientific literature”.

This statement is according to SciFinder database true for natural products, but not for synthetic compounds, although those have only been described in patents. Nevertheless, please rephrase this statement accordingly.

P8, L183–185:

“By all means, we demonstrate that epifadin is the first NRP/PK antimicrobial isolated from the human microbiome and the founding member of a new structural class of peptide-polyene-tetramic acids.”

This statement is highly questionable, since the reutericyclins from *Streptococcus mutans* B04Sm5 (X. Tang, Y. Kudo, J. L. Baker, S. LaBonte, P. A. Jordan, S. M. K. McKinnie, J. Guo, T. Huan, B. S. Moore, A. Edlund, ACS infectious diseases 2020, 6, 563.) are previously isolated NRP/PKs with antimicrobial activity. Please rephrase the wording ((also in L69–72 (P2, 3) and L234–236 (P10)).

Discussion:

P12, L288–291: “Related gene clusters have also been found in *Lactococcus lactis* and *Streptococcus mutans*. However, antimicrobial activities have not been reported for these strains. Moreover, these two BGCs lack EfiO and the gene order is different from the epifadin BGC suggesting that the product is not identical to epifadin”.

Could you please provide in the SI a comparison of these BGCs, since you claim that the associated natural products are probably not identical to epifadin? It would be in particular interesting which part of the secondary metabolite might deviate from epifadin. If necessary, you could also move this part to the Result section, since it would fit quite well to the description and analysis of the biosynthetic pathway.

Author Rebuttal to Initial comments

Revision of NMICROBIOL-22020429; Torres-Salazar et al., point-by-point response to reviewers

Thank you for editing our manuscript. We are glad to see that the reviews were overall positive and contained several valuable comments for improving the manuscript. In this revised version we added additional experimental work in new Figure 2, Extended Data Figures 10, Supplementary Figures 1-5, and Supplementary Videos 1-2, and addressed all the concerns as advised.

The following changes were included:

Reviewer #1 (Remarks to the Author):

This is a well written manuscript supported by well-designed experimentation that describes the detection, purification, estimation of the structure, and the functional analysis of a new antimicrobial encoded on a plasmid present in some strains of the common human nasal and skin microbiont *Staphylococcus epidermidis*. This work will be of broad interest to both microbiome, microbe-microbe interaction and chemical biology researchers.

13RE: We thank the reviewer for this positive assessment.

However, there are several issues that if addressed would strengthen the manuscript.

Suggestions for the authors.

1. How often is the epifadin BGC detected in existing, and publicly available, skin and nasal metagenomic datasets? Detection of the epifadin BGC in these metagenomic datasets would strengthen the argument that this molecule plays a role in shaping microbiota composition on humans and would increase the impact of this manuscript.

RE: This is an important point, which is addressed in the revised manuscript. We performed an in-depth analysis of the eHOMD database (expanded Human Oral Microbiome Database), which contains microbiomes of the human mouth and aerodigestive tract (including the nose). In the eHOMD microbiome data sets neither the *S. epidermidis* nor the highly similar *S. saccharolyticus* BGC were detected. In contrast, less similar but related *Lactococcus lactis* and *Streptococcus mutans* BGCs (Supplementary Figure S1) could be identified. These findings are described in the Results sections (lines 313-315) and discussed (lines 352-358):

2. In lines 94-96, was a search performed to detect the epifadin BGC among all existing bacterial genomes, rather than only among *Staphylococcus* species?

RE: To answer this question, we performed in-depth analyses of the NCBI database, allowing the identification of the BGC in individual isolates. This approach enabled us to identify 10 additional individual *S. epidermidis* isolates encoding the epifadin BGC, indicating that epifadin production is not an exceptional feature. We also could identify three more *S. saccharolyticus* isolates with almost identical BGCs. As described in the discussion (lines 288ff of the original manuscript), we had identified similar BGCs in *Streptococcus mutans* and *Lactococcus lactis* via antiSMASH. The new in-depth analysis revealed even more bacterial species containing BGCs with related composition but with certain genes lacking or additional genes included. Of note, no function has been assigned to and no antimicrobial activities have been documented for these BGCs. The most abundant related BGCs could be identified in *S. mutans* genomes, and in genomes of three species of the genus *Bacillus* (*B. amyloliquefaciens*, *B. thuringiensis*, and *B. velezensis*). In members of the *Paenibacillus*, *Clostridium*, and *Lachnospira* genera BGCs with similar genetic organization could be identified albeit at very low frequencies. These results were now added as Supplementary Fig. 1 and are described in lines 313-315 and 358-363 of the revised

manuscript:

3. Regarding Figure 2, line 125 and lines 264-265, pH 7 and 37°C are inconsistent with *in vivo* conditions in the human nasal vestibule or on human skin where both the pH and temperature are lower, e.g., pH ~5.5-6.5 and temperature range 30-34°C. In this respect, epifadin stability may be tuned for the conditions where *S. epidermidis* primary resides, since *S. epidermidis* is broadly distributed across human skin sites as well as in the human nasal passages. Revised wording in both Figure 2 legend and in the text to more accurately reflect human physiology is recommended.

RE: We agree that our experimental conditions did not reflect well *in-vivo* like conditions. Therefore, we repeated the experiment shown in Figure 2b under more relevant conditions including lower pH and temperature values as suggested. Even at lower pH values and temperatures, reflecting human skin and nasal conditions, epifadin was found to be highly unstable (see new Fig. 2b). The altered experimental conditions are mentioned in the text (lines 122-125):

4. Regarding Figure 6e and lines 227-229, please comment on the apparent bimodal distribution of the percentage of *S. aureus* recovered from the cotton rat nose with wt *S. epidermidis* IVK83. Does this suggest some other mode of competition between the two inoculated species during colonization? Also, this graph would benefit from a box and whiskers plot based on the spread of data points for *S. aureus* + IVK83.

RE: Figure 6e was modified as advised and a sentence was added to address the bimodal distribution of recovered *S. aureus*. We point out that cotton rats are not available from inbred colonies and that genetic or cage-specific differences may impact on nasal microbiome conditions in ways that modulate the competition between *S. epidermidis* IVK83 and *S. aureus*. Nevertheless, the overall difference between animals with wild-type IVK83 or with the deletion mutant is highly significant. This bimodal distribution did not result from spontaneous resistance development by *S. aureus* during colonization of these animals, which was verified by testing epifadin susceptibility of recovered nasal isolates. This point is addressed now in the Results section (lines 265-270):

5. Given the unusual structure of epifadin and the identification of both susceptible and resistant isolates of *S. epidermidis*, are there clues to the mechanism of action? (This may be beyond the scope of the current manuscript.)

15RE: This is an important point we addressed with extensive additional experimental work. Using *Bacillus subtilis* strains with reporter genes that respond to inhibition of known antibacterial targets, epifadin production did not activate any of the reporters. But the inhibitory activity on *B. subtilis* was generally low (see figure below), for which reason we did not include these results in the manuscript. We provide new experiments demonstrating that epifadin leads to rapid depolarization of the cytoplasmic membrane, to membrane damage, and to disruption of bacterial cells (new Extended Data Fig. 10). These data are described at lines 225-233:

In order to understand why several *Staphylococcus* strains are not susceptible to epifadin, an epifadin-susceptible *S. aureus* strain was co-cultivated with the epifadin producer *S. epidermidis* IVK83 for several days and the evolution of spontaneous resistance was monitored. Whole genome sequencing of five resistant and five control clones revealed characteristic resistance-specific mutations in the *desK* gene, encoding the kinase of the two-component system DesKR with unknown function. Mutated DesK harbored one of two independent point mutations, leading to amino acid exchanges (A162V or AV95DG) (Supplementary Figure 5 and Supplementary Table S4). The regulatory consequences of these mutations on the *desKR* regulon, which includes several genes of unknown function, will be part of a separate manuscript. The new data are described at (line 234-246):

6. Line 209: A more cautious interpretation of epifadin toxicity to HeLa cells at 24 ug/ml seems appropriate given that topical use would require demonstration of lack of toxicity to that skin and nasal epithelial cells and systemic administration would require a more thorough assessment of potential toxicity than HeLa cells.

RE: We agree that our data on toxicity to human cells are limited and should be discussed cautiously. The text and Discussion (lines 301-302) were modified accordingly in the Results (lines 220-224): “Purified epifadin was bactericidal - a tenfold IC reduced the number of viable *S. aureus* cells by three orders of magnitude within four hours of incubation (Extended Data Fig. 7a). In contrast, epifadin impaired human HeLa cells only at an IC ca. 20-fold higher than that for *S. aureus* (Extended Data Fig. 7b), suggesting minor effects on viability of human cells”

Minor suggestions

1. Line 244: Given the unusual structure of epifadin with the “alternation of a polar peptide, a non-polar

polyene, and, again, a polar and charged amino acid-derived building block” is it possible that that target in bacteria differs from the target in fungi?

RE: This is something we cannot exclude but, since we cannot provide detailed information on the mode of action, we feel it would be better not to speculate too much. Nevertheless, we removed our statement in the Discussion that bacteria and fungi may share the same epifadin target.

2. Line 246: Is the reduced susceptibility of HeLa cells compared to the two yeast tested related to stability of epifadin under the differing cultivation conditions, i.e., was the pH in the yeast culture lower increasing epifadin’s stability?

RE: This question is difficult to answer, since the test conditions were different. Activity against yeasts was tested on TSB agar by spotting *S. epidermidis* IVK83 on a layer of yeast, which resulted in growth inhibition zones. HeLa cells were tested in liquid RPMI with purified epifadin. Since purified epifadin has limited stability, especially in liquid environment, we cannot exclude that faster degradation impacts the results with HeLa cells. A potentially different level of toxicity of epifadin for other human cell types is discussed now (lines 301-302):

3. Is the IC listed in Fig. 5b the minimal inhibitory concentration? If yes, it would be clearer to list as MIC rather than IC.

Because of the very limited availability of pure epifadin, we could not perform standard MIC assays (serial dilutions in broth). Our alternative test setting is described in the Methods section. We refrain, therefore, from using the term ‘MIC’ and referred to ‘inhibitory concentration’ (IC).

4. Please correct the typo in Figure 5a which should read *Corynebacterium pseudodiphtheriticum* with an h after the first t.

RE: corrected.

5. The *S. epidermidis* strain info is missing on panel C of extended data Fig. 8.

RE: figure was amended.

Reviewer #2 (Remarks to the Author):

This important paper describes a new class of antimicrobial combining a non-ribosomally synthesised peptide and a polyketide called epifadin. ...This is an important and well written manuscript that provides novel insights into interbacterial interactions in the nasal microbiome and the identification of a novel antimicrobial agent with a short half-life. An impressive body of research is presented and the work is suitable for publication in Nature Microbiology.

Perhaps the genetics of epifadin biosynthesis could have been developed further but this is a minor quibble, particularly given that manipulation of a >40Kb gene cluster is technically challenging in *S. epidermidis*.

RE: We added an additional Supplementary Figure 1) showing the comparison of BGSs related to that found in *S. epidermidis* IVK83. However, a detailed characterization of the various genes was beyond the scope of this study but will be part of future studies.

As referred to in the discussion, analysis of the biosynthetic gene cluster did not reveal any insights on the mechanism of action, but of interest there were differences in susceptibility among the *S. epidermidis* nasal isolates tested, with strain 9/16 being completely uninhibited by epifadin (Fig. 5), while the authors also indicate that 5 isolates of *S. epidermidis* are susceptible (line 194). Can the authors provide any additional insights on the basis for these different susceptibilities based on genomic/phenotypic comparisons between resistant and susceptible isolates? Future studies to identify the mechanism of resistance will be important.

RE: We added several additional experiments to address the the mode of action of epifadin and potential reasons for different susceptibilities. We performed reporter strain experiments to identify the type of stress that sensitive strains experience. However, none of the classical responses monitored in this assay was detectable pointing to an uncommon antimicrobial mechanism (Wex KW, et al.; Bioreporters for direct mode of action-informed screening of antibiotic producer strains. Cell Chem Biol. 2021;28(8):1242-1252.e4). It should be mentioned that the reporter strain *Bacillus subtilis* was only weakly inhibited by epifadin, hence we will not present this experiment in the manuscript. For the completeness of information, we show the result in this response letter:

Agar-based bioreporter screening. Reporter enzyme induction pattern elicited by epifadin (top) or by reference antibiotics (bottom) in a *Bacillus subtilis* bioreporter strain panel submerged in agar. Listed are the reporter promoters that, when induced by the indicated kinds of stress, drive the expression of beta-galactosidase. The epifadin extract was spotted directly onto the agar, reference antibiotics were spotted onto filter discs. Growth inhibition by epifadin production is hardly visible.

We nevertheless provide evidence that epifadin compromises cytoplasmic membrane potential and integrity (New Ext. Fig. 10). Moreover, experimental evolution of epifadin resistance led to the identification of the DesKR two-component regulation system whose mutation was found to be involved in resistance development in *S. aureus*. The regulatory consequences of the described *desK* point mutations will be studied in the future. A detailed elucidation of mechanisms will need extensive further analyses of the various genes of the *desKR* regulon. Nevertheless, we hope the presented data are sufficient for this first description of epifadin and its interaction with bacterial cells. Additional data on the mode of action of epifadin are shown in new Extended Data Fig. 10 and described in the text at lines 225-233. The new data are also addressed in the discussion at lines 294-297:

Did the authors consider extension of the co-culture experiments between IVK83 and a susceptible *S. epidermidis* isolate (perhaps a rifampicin resistant derivative) to determine if resistant suppressor mutants can be isolated?

RE: As described above, such experiments with *S. aureus* are now included in the revised manuscript. They indicate that point mutations in *desK* can indeed lead to reduced susceptibility, albeit, without pointing to a specific potential resistance factor or mechanism as the DesKR regulon encompasses many

19genes of virtually unrelated and often unclear function..

Reviewer #3 (Remarks to the Author):

This manuscript reports the discovery of a novel antibiotic produced by the common human nose isolate *S. epidermidis* IVK83, which drew the attention of the research group by displaying strong antibacterial activity against many strains from their bacterial collection. ... I believe the manuscript presents a story that would be of interest to the readership of Nature Microbiology. However, it suffers from two major flaws that mean I cannot recommend this manuscript for publication in Nature Microbiology in its current state (further elaborated below):

1. While the storyline is well-organized, the manuscript would benefit from extensive editing, and some paragraphs are difficult to understand and/or are missing detailed explanation.

RE: We are grateful for all the suggestions given below, which were all addressed to improve the readability.

2. The structure determination and bioinformatic parts are weak so could be improved or toned down in favour of the biological significance.

RE: We performed additional experiments to confirm the stated absolute configuration of the N-terminal epifadin peptide amide. To this end, the peptide amide was synthesized in the two possible conformations (LDLD-NH₂ or DLDL-NH₂), and comparison with the natural peptide amide on a chiral HPLC column clearly confirmed the LDLD conformation (Supplementary Figure 2). This was also predicted by AntiSmash. The here presented structure is entirely based on unambiguous data obtained by HPLC-MS/MS and NMR. The only remaining uncertainty is the absolute configuration of the C6H10 fragment in the polyene moiety of the molecule. The additional experiments are described in the Results at (lines 159-161) and in the Supplementary Information.

3. The microbiology part and the biological activities are not fully conclusive.

RE: Please find our responses below to the specific points.

Major Comments

Structural Elucidation/Bioinformatics

20Unfortunately, the structure of epifadin is not yet elucidated completely by NMR spectral data because of instability. The authors showed only the N-terminal tetrapeptide unit clearly, while the other parts are deduced by MS and bioinformatic analysis. The authors emphasize that epifadin is novel and the first hybrid NRP-PK from a member of the human microbiota. Thus, more precise structure elucidation using NMR is pivotal to support the structure. Various highly instable compounds have already been purified and measured by NMR. The authors could use either 600 or 700 MHz NMR (with a cryoprobe), so that authors consume 0.5 mg per measurement at $-20\text{ }^{\circ}\text{C}$. It would not be essential to show pure NMR spectra, but the authors have to identify the main signals.

At least in page 7, line 162, authors describe "... ^{13}C NMR...tetramic moiety", but the ^{13}C NMR spectrum is not shown. Although the authors explain the proposed structure and biosynthetic pathway, there are no detailed bioinformatic data including adenylation domain codes. KR domain fingerprinting analysis also may show geostereochemistry of double bonds, but the authors do not explain the reason why all double bonds are trans configurations (perhaps ROESY correlations, but it is not explained at all).

RE: We would like to emphasize that the original manuscript already contained the requested NMR data. As highlighted in Figure 4, the peptide moiety was elucidated by NMR and MS², the entire polyene structure was elucidated by NMR (not by MS or bioinformatic prediction), and the tetramic acid moiety was clarified by single NMR and MS, also supported by bioinformatic analyses. For the full picture of performed experiments we now included the ^1H - ^1H -COSY and ^1H - ^{13}C -HMBC spectra (Supplementary Figures 3 and 4) with an explanation of the found correlations. We hope that these data now convince this reviewer that the presented structure of epifadin is beyond doubt.

From a structural point of view, the combination of three different units may be new, but each moiety is not really new. Therefore, the authors should show other groups' compounds to highlight this. For example, reutericyclin/mutanocyclin, which have been isolated from *Lactobacillus* in human gut/nasal/oral cavity, possess a tetramic moiety and an acyl chain. Also, mutanofactin has been isolated from human oral *Streptococcus* and consists of an unsaturated fatty acyl chain and peptide. These examples are only from human microbiota so there could be other similar compounds known.

RE: We are thankful for this comment. We were aware of reutericyclin, which shares the tetramic acid moiety with epifadin and antimicrobial activity. Nevertheless, the peptide moiety was found to be essential for epifadin's activity. We now included a paragraph, which considers the mentioned compounds, but we would like to stress that despite intense search, epifadin remains the only known peptide-polyene-tetramic acid molecule. The additional points are mentioned in the discussion at lines 303-315:

The authors suggest that the tetrapeptide amino acid sequence is L-Phe-D-Phe-L-Asp-D-Asn, but the 4 NRPS modules of EfiA indicate D-Phe-L-Phe-D-Asp-L-Asn. The first module has a sequence of domain A-C-T-E, which is bit unusual because the usual domain sequence is A-T-E-C. The downstream 3 modules are normal type, but judging from this module composition, D-L-D-L is a true sequence.

RE: We agree with the reviewer that the domain sequence A-C-T-E-(C) is not the expected order. However, only this order can explain the predicted and also confirmed LDLD conformation of the N-terminal peptide. The first A-domain with specificity for phenylalanine is responsible for the activation of the first two amino acids, the first of which is incorporated in its L- and the second is epimerized to yield D-conformation. In our recent publication, where we identified the structure of the cyclic NRPS peptide lugdunin, produced by *Staphylococcus lugdunensis*, a single A-domain is responsible for the incorporation of three consecutive valine residues in L-, D-, and again L-conformation. The domain order in the lugdunin BGC is A-T-C-T-E-C-T, which is also very uncommon. To unequivocally confirm the LDLD configuration of the N-terminal peptide of epifadin, we synthesized both versions (FfDn-NH₂ and fFDN-NH₂) and compared their elution behavior with that of the natural peptide on a chiral HPLC column. These data clearly demonstrate that the epifadin peptide has a LDLD configuration (see new Supplementary Fig. 2 and text at lines 159-161 and in the Supplementary Information).

The obtained shunt tetrapeptide does not have antibacterial activity. However, the scenario cannot be excluded that all three units are important to pass the cell membrane. The authors should discuss. An L-D-L-D peptide very likely makes a helix like gramicidin. For example, after epifadin enters into cell, tetrapeptides could be activated by aggregation. Has a tetramic acid moiety alone been reported to have any antibacterial activity?

RE: The acyl-tetramic acid-containing reutericylin, has indeed antimicrobial activity, suggesting that the tetramic acid group may play a critical role in the antimicrobial mechanism. Reutericyclin and its acyl-free variant mutanocyclin, which lacks antimicrobial activity, are mentioned now in the discussion (lines 304-310). We can at the moment not exclude that, after incorporation of epifadin into the target cell, the tetrapeptide is released and aggregates but this is pure speculation, and we suggest to refrain from discussing this point further. Gramicidin D, which forms helices in the membrane, is much bigger (15 aa) and able to span the membrane thereby acting as an ionophore. It is hard to imagine how tetrapeptides (FfDn-NH₂) should aggregate and form channels. Time lapse microscopy indicates membrane disruption as a consequence of epifadin treatment and gives first hints on the mode of action. This has been included in the manuscript (Extended Data Figure 10).

Microbiology

To strengthen the relevance of the results, the authors should make clear the scale of the problem of *S. aureus* as an opportunistic pathogen and risk of *S. aureus* infection.

RE: We added more background to the importance of controlling *S. aureus* colonization for the prevention of severe infections in the introduction (lines 42-43).

The authors are experts of antibacterial activities, but readers of this journal come from many different fields, including both students and scientists. Thus, authors must better explain how epifadin is a potent antibiotic. In Figure 5, vancomycin and daptomycin are used as famous and clinical antibiotic controls. However, what about a comparison to lugdunin, which has been reported by the authors' group? This peptide also inhibits the growth of *S. aureus* and could be further discussed from an ecology perspective.

RE: The IC values for lugdunin were added to Fig. 5b as suggested. In the text we included a comparison with lugdunin (lines 364-369): "Whereas the highly stable lugdunin acts as a protonophore leading to the breakdown of the membrane potential and subsequent slow killing of bacterial cells, the unstable epifadin seems to instantly disrupt the membrane of *S. aureus* leading to fast cell lysis, highlighting two different strategies to combat *S. aureus* in the human microbiome."

The rat model should be briefly explained for the unfamiliar reader. For example, I assume it is not germ-free, which would mean that it could be said to replicate a somewhat real-world habitat. This would mean that the bioactivity shown in this section is perhaps the most convincing of all the results of this section could be "hyped up" accordingly. Can the authors also comment on the apparent "split" in efficacy of *S. aureus* inhibition by the WT, as observed in Fig. 6e? Could it be that the microbiota of some individuals differs meaning that the compound is "mopped up" by other susceptible organisms, thereby being diverted from affecting *S. aureus* so profoundly? But of course, this would not be seen in this experimental set-up seeing as microbiota-level dynamics weren't tracked (the overall composition is unknown), just the competing *S. epidermidis* and *S. aureus* strains.

RE: The cotton rat model of nasal colonization is indeed not based on germ-free animals. As only a small number of laboratories around the world use these animals, inbred animals with controlled or standardized microbial colonization are not available. As outlined in the response to reviewer 1, we assume that individual genetic polymorphisms or subtle differences in the exposure to other

microorganisms in the animal facility may have led to the dichotomous behavior of animals colonized with *S. aureus* and the epifadin-producing *S. epidermidis* strain. However, despite the uneven distribution of nasal bacterial counts, the differences are highly significant. We added a statement addressing this point in the Results section (lines 265-270).

Could the authors comment on whether the stability of epifadin could be better *in vivo* in some way i.e. is possible that the *in vivo* environment is less harsh? I know of at least one case where a natural product (streptolysin S) cannot be extracted in a bioactive (and presumably intact) form without the presence of carrier molecule, and it has been proposed that *in vivo* this problem is overcome by the toxin being cell-bound until it is deposited upon the surface of target cells by direct contact.

RE: This is indeed an interesting possibility. As of now, we do not have evidence for higher stability of epifadin or for binding to carrier molecules *in vivo*. We speculate on such possibilities now in the Discussion section (lines 372-373): "It remains to be clarified how unstable epifadin is in an *in vivo* setting where mucus or mucosal components could influence its stability."

I find interesting the idea put forward by the authors that the highly instable nature of the compound might minimize collateral damage. This is one of the big impacts of the report but is not shown so can the authors at least speculate how it would be possible to provide support for this (future directions)?

RE: We extended the last paragraph of our Discussion section to devise future research directions and potential ways to elucidate how the instability of epifadin may help to preserve overall microbiome integrity (lines 373-374): "Mass spectrometry-based imaging combined with organoid-based microbiome models, which more mimic *in vivo* physiology, could help clarify this question."

Minor comments

Main Text

Please unify "-" and "–" throughout.

RE: changed as advised.

L22 and throughout: Should be microbiota (all the microorganisms that are found within a specific environment) not microbiome (the collection of genomes from all the microorganisms in a given environment).

RE: It is obvious that the original distinction between the terms microbiome and microbiota has become more and more blurred in the scientific community. The combined genetic material of microorganisms in a particular environment is now increasingly described with the term 'metagenome'. In contrast, the term 'microbiome' is more commonly used than the term 'microbiota' to describe microbial communities. A recent consensus paper suggested the use of 'microbiota' to describe only the member organisms of a community while 'microbiome' should include the member organisms plus their 'theatre of activities and molecules' (Berg et al, 2020, Microbiome, 8:103). Since we refer almost always to microbial communities in the full context of their activities and molecules, we feel it is more appropriate to use the term 'microbiome'. We are open though for any other nomenclature suggested by the reviewers and the editor.

L24: This sentence is phrased strangely – sounds like separate things. Maybe change to “It has an unprecedented architecture consisting of a non-ribosomally synthesized peptide component, a polyketide component, and a terminal modified amino acid moiety.”

RE: changed as advised.

L35: Insert reference(s) (review on the topic?) to support “crucial roles”.

RE: Literature cited (Bomar, L., Brugger, S. D. & Lemon, K. P. Bacterial microbiota of the nasal passages across the span of human life. *Curr Opin Microbiol* 41, 8-14 (2018). <https://doi.org/10.1016/j.mib.2017.10.023>

L38: Unnecessary comma after “both”.

RE: changed as advised.

L42: “S. epidermis can...competitors” Recommend to split sentence.

RE: Sentence was split.

L51: In what way is bacteriocin production beneficial? Presumably when competitors are eliminated.

RE: We extended the sentence to better explain the point (lines 57-59): “Elimination of competitors by bacteriocin production is often beneficial but can also cause collateral damage to the producer when other bacteria, which are necessary for the function of mutualistic networks, are also killed.”

L60: Change to “lifetimes”.

RE: changed as advised.

I would assume “distance from” rather than “distance to” since bacteriocins originate from the producer and diffuse away.

RE: changed as advised.

Also, why not producer in general rather than “producing micro-colony”? If this is a technical term, explain since “colony” makes me think of in an artificial lab set-up.

RE: changed as advised.

L64: “Such a strategy can only work if...” Explain why this is the case as not obvious.

RE: We modified the sentence to better explain the point (lines 70-72): “Such a strategy can only work if community density is relatively low, and the microbiome composition and structure are sufficiently stable in time and space, which is usually the case in skin and anterior nares habitats.”

L65: Reference to support “...which is usually the case in skin...”

RE: We added new reference (No. 23)

L69: If it is the first example of this strategy, wouldn't it be unique so far, not “unusual”? I would delete as “previously unrecognized” is sufficient.

RE: changed as advised.

L68: Suggest replacing “reflecting” with “representing”.

RE: changed as advised.

L69: I would rather say “first discovered”.

RE: changed as advised.

L71: Should be NRPSs and PKSs.

RE: changed as advised.

L72: Suggest “their shared habitat”.

RE: changed as advised.

I would not consider “in vitro” a habitat (i.e. the natural home of an organism). I would separate these points – shown both in a lab-based setup and, more significantly, in vivo.

RE: changed as advised.

L81: Highlight results by beginning sentence as follows “We determined that the transposon had...”
Is *S. aureus* USA300 an MRSA strain? If so, probably of interest to state.

RE: changed as advised.

L82: Suggest to delete “operon” as superfluous.

RE: changed as advised.

L83: Why did you deem this unusual? Hybrid NRPS-PKS biosynthetic gene clusters encode NRPSs and PKSs. Is a set of NRPS-transAT iterative PKS-tetramic acid forming NRPS unusual?

RE: changed as advised.

L84: Suggest insertion "...set of putative biosynthetic enzymes, namely a putative NRPS (EfiA)..." Also, I would use "hybrid PKS-NRPS" (enzymes) or "hybrid PK-NRP" (natural product) here and throughout since this is standard in the literature; important also to hyphenate as "/" suggests "or".

RE: changed as advised.

L85: Suggest "biosynthesis".

RE: changed as advised.

L91: What is meant by screening? First identified those with BGC then checked for similar antimicrobial activity?

RE: changed to (lines 99-100): "Whole genome sequencing (WGS) of various other inhibitory *S. epidermidis* strains from different strain collections identified four additional isolates, ..."

L94: More accurate to describe something like "Homologous BGCs were absent from all *S. epidermidis* genomes..."

RE: changed to: "Analysis of the NCBI database revealed the presence of further ten *S. epidermidis* isolates containing an identical BGC, indicating that the ability to produce epifadin might be a sporadic but consistent trait among *S. epidermidis* strains. Additionally, a nearly identical BGC, which differed only in the length of the initial NRPS gene, was also present in five *Staphylococcus saccharolyticus* genomes indicating that it is also an infrequent accessory genetic element in this species (Supplementary Fig. 1).

L100: Suggest replacing "contained" with "possessed" or "displayed".

RE: changed as advised.

L110: Suggest replacing "faded" with "diminished".

RE: changed as advised.

L114: Edit for clarity. "Reversed-phase high-performance liquid chromatography with ultraviolet light

detection (RP-HPLC-UV) of DMSO-PA extracts of *S. epidermidis* IVK83 and *S. epidermidis* IVK83 Δ efiTP, obtained from acid-precipitated culture supernatants, revealed no obvious difference between the IVK83 wild type and efiTP mutant strain at the peptide-bond specific wavelength of 215 nm, although the NRPS-encoding genes for NRPS pointed to an antimicrobial product with the presence of multiple peptide bonds.” Also recommend to split sentence.

RE: Sentence was split and clarified.

L118: Edit for clarity. “However, the HPLC profile of the wild type and mutant differed in displayed a major and in some adjacent minor peaks at 383 nm that were absent in that of the mutant, indicating a product possessing the presence of an expanded unsaturated system with double bonds (Figs. 3a-c).”

RE: changed as advised.

L129: Suggest insertion. “...predicted the product of the cluster to have a three-partite composition”.

RE: changed as advised.

L134: Re-phrase to clarify if this means there were multiple purifications from one 100 L culture, or multiple 100 L cultures?

RE: changed as advised.

L141: Remove unnecessary comma after “Marfey`s analysis”.

RE: changed as advised.

L151: Split sentence. “Two-dimensional NMR ... Extended Data Fig. 4a).”

RE: changed as advised.

L177: Suggest replacing “combines” with “is derived from a combination of”.

RE: changed as advised.

L 179: "...multidimensional NMR analyses...", but Extended Figure 3 and 4a shows only ROESY and ¹H NMR of purified epifadin, respectively. It seems that none of the NMR spectra of the synthetic tetrapeptide are shown elsewhere.

RE: Extended Data Figure 2 shows NMR of the synthetic peptides; ¹H-¹H-COSY and ¹H-¹³C-HMBC spectra were added as new Supplementary Figures 3 and 4.

L181: "Moreover... instability of epifadin." Recommend sentence as better fit to the Discussion. Consider replacing "Moreover" with "In the future".

RE: Sentence was moved and modified as advised (now lines 283-285): "We will focus on a chemical total synthesis in the future to study structural details of the C₆H₁₀ bridge in the polyene moiety, which remains challenging due to the extraordinary instability of epifadin."

L182: Suggested simplification. "which will require further efforts though and remains challenging..."

RE: This sentence was rephrased as described for L181.

L183: Incorrect usage of "By all means". Suggest changing as follows: By all means Interestingly, we demonstrate that epifadin is the first...and is the founding member..."

RE: changed to (lines 197-199): "In conclusion, we demonstrate that epifadin is the first NRP-PK antimicrobial isolated from the human microbiome and is the founding member of a new structural class of peptide-polyene-tetramic acids."

L187: Wild type and mutant strains are not referred to consistently e.g. wild type variously written as S. epidermidis IVK83, IVK83, IVK83 wild type; mutant as efiTP mutant strain, ΔefiTP, S. epidermidis ΔefiTP, IVK83 ΔefiTP. For simplicity, always give the full names S. epidermidis IVK83 and S. epidermidis IVK83 ΔefiTP, with the occasional use of wild type and mutant when clear by context (e.g. full names used in previous sentence).

RE: We use strain designations more consistently in the revised manuscript. We suggest keeping the short forms where suitable because the full names would drastically increase the word count.

L189: This sentence should be moved to the paragraph starting L202 for clarity, i.e. keep producer/purified compound experiments separate.

RE: changed as advised.

L196: Consider "...nystatin A1 contain similar polyene moieties similar to that of as epifadin"

RE: changed as advised.

L204: Consider "...MIC values (between 3.7 and 8.6 µg/mL)..." for ease of reading.

RE: changed as advised. "MIC" was replaced by "IC" to indicate that an alternative method was applied

L208: "Epifadin targets..." Sentence seems out of place as no context for the point is given.

RE: this sentence was deleted

L209: 20 times which MIC? Unclear.

RE: has been clarified.

L210: Consider "...suggesting thea lack...".

RE: has been changed

L213: Consider "...raised the question ifof whether..."

RE: changed as advised.

L226: Consider "...nasally instilled into the noses..."

RE: changed as advised.

L228: Split sentence e.g. “...(Fig. 6e),. In this way, we could demonstrate indicating that epifadin-producing *S. epidermidis* can effectively interfere with *S. aureus* nasal colonization.

RE: We changed to: “...(Fig. 6e). This indicates that epifadin-producing *S. epidermidis*...”

L235: Unnecessary comma. “...both, NRP and PK moieties...”

RE: changed as advised.

L236: “new compound class” – name it.

RE: This new compound class will get a specific name in the future when we gain more chemical knowledge of the chemical structures of further class members. We prefer to coin the name only in a future study, when a currently running chemical total synthesis program delivers epifadin analogues, which will teach us on the requirements of (1)peptide- (2)alkyl/aryl - (3)polar moieties for bioactivity and mode of action. The name will then comprise the 'chemical-antimicrobial concept' of our here presented new compound class.

L239: “...rely on...” For what?

RE: changed to “are based on”

L242: Use of “again” is confusing as there has not been a prior single polar charged amino acid.

RE: “again” was deleted

L243: Redundant as implied already by “remains”. “...remains to be elucidated in the future,...” “...the molecular target appears to be shared by bacterial and fungal cells...” How do you conclude that? Since there is antibiotic activity towards both? I think too speculative as, in theory, it could have a different target in both to exert activity.

RE: this sentence was deleted

L246: How do you know it is taken up? Could act at the membrane for example. If this is based on the known behaviour of similar molecules, provide reference(s).

RE: Our new experiments indicate that epifadin damages bacterial cytoplasmic membranes, a process involving the uptake into the membrane compartment. Therefore, we suggest not to change the wording.

L263: It might initially At first thought, it may seem paradoxical and a waste of energy for a bacterium to waste energy to by produceing an antimicrobial with such a short lifespan.

RE: changed as advised.

L315 and throughout: Should be "...10 µl werewas..." since it is a collective noun so counted as a single entity.

RE: changed as advised.

L359: Fix typo. "...the genes ΔefiTP were amplified..."

RE: changed as advised.

L394: Fix typo. "Sodium formiate ..."

RE: changed as advised.

L471: How were the CFUs of the two strains differentiated? Later described but should be mentioned here too.

RE: explanation added.

L485: "...inoculum of 1 x 10⁷CFU/nose isare required..."

RE: all the above-listed points were addressed, and sentences were amended as suggested.

Figures

Figure 1: Increase font sizes in Part C.

RE: changed as advised.

L694: "DMSO extract..." Split sentence.

RE: changed as advised.

Figure 3: The quality of chromatograms is not high enough.

RE: quality was adjusted

Figure 4: Peak labels difficult to read in Part A. Keep font-size consistent throughout all figures.

RE: changed as advised.

Figure 5: Strain names should not be italicized in Part B. Addition of MIC values in μM (standard) would allow direct comparison to potency of controls.

RE: changed as advised.

Extended Figure 5: The quality of structures is not high enough.

RE: quality was adjusted and original ChemDraw files are submitted to the Journal

Extended Data Figure 7: Is the last bar in Panel b&c an untreated control?

RE: We are sorry for this unprecise presentation. The last bar is not the untreated control but is missing the significance label. All bars represent the viability of HeLa cells at the indicated compound concentrations. Data was normalized to the DMSO treated cells, which was set to 100%. This is now explained in the corresponding Figure legend.

Extended Data Figure 8: Label above bar chart missing in Part C

RE: changed as advised.

Extended Figure 9: The quality of mass spectra is not high enough.

RE: quality was adjusted

Supplementary

In general, some amino acid stereochemistry did not use lower case (eg. line 42: L-Ala). Please unify them.

RE: changed as advised.

L24: Missing abbreviation of PA in DMSO-PA, even if it is described in a main text.

RE: changed as advised.

Reviewer #4 (Remarks to the Author):

General assessment:

The manuscript “Extremely short-lived peptide-polyene antimicrobial enables nasal commensal to eliminate *Staphylococcus aureus*” presents the discovery of the NRPS-PKS based peptide-polyene epifadin from the nasal commensal *S. epidermidis* IVK83. ... The herein presented study highlights an interesting concept for the exploration of new antimicrobials from nasal commensals, which might also support the discovery of new bioactive natural products from conventional habitats.

Unfortunately, the authors were not able to reveal or provide any experimental information considering the mode-of-action of epifadin. Therefore, it is highly recommended to perform at least an additional experiment to obtain in vitro Epifadin-resistant mutants (see below). Furthermore, the authors could provide more information considering a potential molecular target of Epifadin.

RE: Epifadin-resistant *S. aureus* were generated by experimental evolution during serial co-cultivation of epifadin-susceptible *S. aureus* and epifadin-producing *S. epidermidis* (Supplementary Figure 5). Subsequent whole genome sequencing identified mutations in *desK* (encoding the kinase of the DesKR two-component system; Supplementary Table 4) potentially involved in resistance development in *S. aureus*. The cellular consequences of the described *desK* point mutations, which probably affect the

35entire DesKR regulon including several genes of unknown function will be studied in the future. Time lapse microscopy and membrane potential measurement indicated the cellular membrane as target since *S. aureus* is immediately disrupted after contact with epifadin extract (Extended Data Figure 10). The experiments were added to the manuscript and described in the text as outlined.

Nevertheless, the manuscript should become suitable once the issues raised are addressed for Nature Microbiology, since it provides new insights to the microbial production of short-lived natural products and is interesting for the broad community of scientist dedicated to the exploration of natural products from the human microbiome.

Major suggestions:

Is it possible to generate spontaneous resistant mutants of *Staphylococcus aureus* USA300LAC/JE2 (or another suitable sensitive strain) in vitro by applying lower antibiotic pressure and repeated passaging of incrementally resistant colonies on rising Epifadin concentrations? Subsequent genomic DNA isolation and whole-genome sequencing from all spontaneous resistant mutants could help to reveal the molecular target of Epifadin; and even if the mutation would only result in the overexpression of an ABC transporter (which seems also to be the encoded self-resistance mechanism of the producer strains), this outcome might still be useful for future investigations.

RE: We performed such an experiment and found a series of epifadin-resistant mutants. As outlined above, whole genome sequencing of such mutants identified *desK* as a key factor of resistance development but with no obvious hints towards the resistance mechanism.

P8/9 L194–195

“Notably, all tested *S. aureus* strains were susceptible to epifadin, whereas 9 of 16 tested nasal *S. epidermidis* isolates were resistant.”

Any idea, what is the genetic difference between an Epifadin-resistant and non-resistant *S. epidermidis* isolate? Maybe the presence of distinct ABC transporters? Please comment and provide more information to this observation.

RE: We compared genomes of epifadin-susceptible and -resistant isolates carefully but, because the isolates are from different clonal backgrounds, the genomic differences are extensive, and we could not identify a potential resistance factor.

P8/9 L248–250

“However, the epifadin BGC encodes only the two ABC transporter components EfiFG as potential resistance proteins, which may act as drug exporters and do not shed light on the antibacterial target”
In order to prove this statement experimentally, it would be interesting to generate in addition to the mutant *S. epidermidis* IVK83 Δ efiTP, the mutant *S. epidermidis* IVK83 Δ efiFGTP. Subsequent comparison of both Epifadin-nonproducers against the epifadin-producing isolate *S. epidermidis* IVK83 could reveal the role of both genes.

RE: This is certainly an interesting point. However, investigations about resistance in *S. epidermidis* are part of ongoing work, which will be published in a separate manuscript with a focus on the ABC transporter EfiFG.

Minor suggestions:

Title

Have you considered implementing the name of the natural product in the title e.g.

“The extremely short-lived peptide-polyene antimicrobial epifadin enables nasal commensal to eliminate *Staphylococcus aureus* “

RE: we now included the name in the title as suggested.

Abstract:

P1, L24:

“...*Staphylococcus epidermidis*.” Could you please add the exact strain name “*Staphylococcus epidermidis* IVK83”

RE: Strain name was added.

Introduction:

P3, L45-46 and L47:

“---- and may impair the fitness of its major competitors.” and “....activity against potential target species”.

Could you please mention one or two relevant examples of major competitors and target species (Probably *S. aureus* as mentioned on P4, L72) ?

RE: some examples were added

P3, L67:

“Here we present a novel bacteriocin-like antimicrobial, epifadin.....”

This statement is ambiguous, since the bacteriocins are RiPPs and epifadin is a natural product produced by a hybrid NPRS-PKS assembly line. Furthermore, the structural architecture of epifadin is clearly distinct from typical RiPPs. Only the biological function of epifadin is similar to previously described bacteriocins. Please change the wording accordingly.

RE: Sentence was modified (lines 73-75): “Here we present a novel antimicrobial small molecule compound, epifadin, produced by certain *S. epidermidis* strains, that combines an exceptionally wide target range with a very short half-life thereby representing a previously unrecognized antimicrobial strategy.”

Result:

P6, L123:

“This formula was unknown in scientific literature”.

This statement is according to SciFinder database true for natural products, but not for synthetic compounds, although those have only been described in patents. Nevertheless, please rephrase this statement accordingly.

RE: statement was clarified (lines 136-138): “This formula was unknown in publicly available literature, indicating that IVK83 produces a novel compound....”

P8, L183–185:

“By all means, we demonstrate that epifadin is the first NRP/PK antimicrobial isolated from the human microbiome and the founding member of a new structural class of peptide-polyene-tetramic acids.”

This statement is highly questionable, since the reutericyclins from *Streptococcus mutans* B04Sm5 (X. Tang, Y. Kudo, J. L. Baker, S. LaBonte, P. A. Jordan, S. M. K. McKinnie, J. Guo, T. Huan, B. S. Moore, A. Edlund, ACS infectious diseases 2020, 6, 563.) are previously isolated NRP/PKs with antimicrobial activity. Please rephrase the wording ((also in L69–72 (P2, 3) and L234–236 (P10)).

RE: As described above, reutericyclin, mutanocyclin and mutanofactin share some structural similarities with epifadin, but are composed of an acyl tetramic acid (reutericyclin) or acyl-free tetramic acid (mutanocyclin) or a peptide-polyene structure (mutanofactin). None of them and to our knowledge no other molecule in the SciFinder database is composed of peptide-polyene-tetramic acid moieties. This point is outlined in more detail in the discussion now (lines 303-315).

Discussion:

P12, L288–291: “Related gene clusters have also been found in *Lactococcus lactis* and *Streptococcus mutans*. However, antimicrobial activities have not been reported for these strains. Moreover, these two BGCs lack EfiO and the gene order is different from the epifadin BGC suggesting that the product is not identical to epifadin”.

Could you please provide in the SI a comparison of these BGCs, since you claim that the associated natural products are probably not identical to epifadin? It would be in particular interesting which part of the secondary metabolite might deviate from epifadin. If necessary, you could also move this part to the Result section, since it would fit quite well to the description and analysis of the biosynthetic pathway.

RE: An additional Supplementary Figure 1 was added with the comparison of the BGCs and this point is more extensively discussed now (lines 352-363).

Further changes:

The names of five more scientists, who contributed significantly to the additional experiments, were added to the author list.

We hope that our manuscript is now suitable for publication in Nat Microbiol.

Decision Letter, first revision:

Message: 10th January 2023

Dear Dr Krismer,

39Thank you for your patience while your manuscript "The extremely short-lived peptide-polyene antimicrobial epifadin enables nasal commensal to eliminate *Staphylococcus aureus*" was under peer-review at Nature Microbiology. It has now been seen by 4 referees, whose expertise and comments you will find at the of this email. You will see from their comments below that while they find your work of interest, some important points are raised. We are very interested in the possibility of publishing your study in Nature Microbiology, but would like to consider your response to these concerns in the form of a revised manuscript before we make a final decision on publication.

In particular, you will see that referees #3 and #4 both request 2D NMR data for the tetramate and we will require this for publication. If you cannot provide these data please let us know and we would be happy to consult with our colleagues at Nature Communications to see if they would be interested in taking the paper without these data. The rest referees' reports are clear and the remaining issues should be straightforward to address.

If you have not done so already please begin to revise your manuscript so that it conforms to our Article format instructions at <http://www.nature.com/nmicrobiol/info/final-submission/>

The usual length limit for a Nature Microbiology Article is six display items (figures or tables) and 3,000 words. We have some flexibility, and can allow a revised manuscript at 3,500 words, but please consider this a firm upper limit. There is a trade-off of ~250 words per display item, so if you need more space, you could move a Figure or Table to Supplementary Information.

Some reduction could be achieved by focusing any introductory material and moving it to the start of your opening 'bold' paragraph, whose function is to outline the background to your work, describe in a sentence your new observations, and explain your main conclusions. The discussion should also be limited. Methods should be described in a separate section following the discussion, we do not place a word limit on Methods.

Nature Microbiology titles should give a sense of the main new findings of a manuscript, and should not contain punctuation. Please keep in mind that we strongly discourage active verbs in titles, and that they should ideally fit within 90 characters each (including spaces).

40Please include a data availability statement as a separate section after Methods but before references, under the heading "Data Availability". This section should inform readers about the availability of the data used to support the conclusions of your study. This information includes accession codes to public repositories (data banks for protein, DNA or RNA sequences, microarray, proteomics data etc...), references to source data published alongside the paper, unique identifiers such as URLs to data repository entries, or data set DOIs, and any other statement about data availability. At a minimum, you should include the following statement: "The data that support the findings of this study are available from the corresponding author upon request", mentioning any restrictions on availability. If DOIs are provided, we also strongly encourage including these in the Reference list (authors, title, publisher (repository name), identifier, year). For more guidance on how to write this section please see: <http://www.nature.com/authors/policies/data/data-availability-statements-data-citations.pdf>

To improve the accessibility of your paper to readers from other research areas, please pay particular attention to the wording of the paper's opening bold paragraph, which serves both as an introduction and as a brief, non-technical summary in about 150 words. If, however, you require one or two extra sentences to explain your work clearly, please include them even if the paragraph is over-length as a result. The opening paragraph should not contain references. Because scientists from other sub-disciplines will be interested in your results and their implications, it is important to explain essential but specialised terms concisely. We suggest you show your summary paragraph to colleagues in other fields to uncover any problematic concepts.

If your paper is accepted for publication, we will edit your display items electronically so they conform to our house style and will reproduce clearly in print. If necessary, we will re-size figures to fit single or double column width. If your figures contain several parts, the parts should form a neat rectangle when assembled. Choosing the right electronic format at this stage will speed up the processing of your paper and give the best possible results in print. We would like the figures to be supplied as vector files - EPS, PDF, AI or postscript (PS) file formats (not raster or bitmap files), preferably generated with vector-graphics software (Adobe Illustrator for example). Please try to ensure that all figures are non-flattened and fully editable. All images should be at least 300 dpi resolution (when figures are scaled to approximately the size that they are to be printed at) and in RGB colour format. Please do not submit Jpeg or flattened TIFF files. Please see also 'Guidelines for Electronic Submission of Figures' at the end of this letter for further detail.

Figure legends must provide a brief description of the figure and the symbols used, within 350 words, including definitions of any error bars employed in the figures.

When submitting the revised version of your manuscript, please pay close attention to our [href="https://www.nature.com/nature-research/editorial-policies/image-integrity">Digital Image Integrity Guidelines. and to the following points below:](https://www.nature.com/nature-research/editorial-policies/image-integrity)

-- that unprocessed scans are clearly labelled and match the gels and western blots presented in figures.

41- that control panels for gels and western blots are appropriately described as loading on sample processing controls
- all images in the paper are checked for duplication of panels and for splicing of gel lanes.

Please include a statement before the acknowledgements naming the author to whom correspondence and requests for materials should be addressed.

Finally, we require authors to include a statement of their individual contributions to the paper -- such as experimental work, project planning, data analysis, etc. -- immediately after the acknowledgements. The statement should be short, and refer to authors by their initials. For details please see the Authorship section of our joint Editorial policies at http://www.nature.com/authors/editorial_policies/authorship.html

- * include a point-by-point response to any editorial suggestions and to our referees. Please include your response to the editorial suggestions in your cover letter, and please upload your response to the referees as a separate document.

- * ensure it complies with our format requirements for Letters as set out in our guide to authors at www.nature.com/nmicrobiol/info/gta/

- * state in a cover note the length of the text, methods and legends; the number of references; number and estimated final size of figures and tables

- * resubmit electronically if possible using the link below to access your home page:

[redacted]

- *This url links to your confidential homepage and associated information about manuscripts you may have submitted or be reviewing for us. If you wish to forward this e-mail to co-authors, please delete this link to your homepage first.

Please ensure that all correspondence is marked with your Nature Microbiology reference number in the subject line.

Nature Microbiology is committed to improving transparency in authorship. As part of our efforts in this direction, we are now requesting that all authors identified as 'corresponding author' on published papers create and link their Open Researcher and Contributor Identifier (ORCID) with their account on the Manuscript Tracking System (MTS), prior to acceptance. This applies to primary research papers only. ORCID helps the scientific community achieve unambiguous attribution of all scholarly contributions. You can create

and link your ORCID from the home page of the MTS by clicking on 'Modify my Springer Nature account'. For more information please visit www.springernature.com/orcid.

We hope to receive your revised paper within three weeks. If you cannot send it within this time, please let us know.

Yours sincerely,

[redacted]

Reviewer Expertise:

Referee #1: nasal microbiome, microbial interactions

Referee #2: Staphylococcus

Referee #3: natural products

Referee #4: natural products

Reviewers Comments:

Reviewer #1 (Remarks to the Author):

In this revised manuscript, the authors have addressed the majority of the review comments well resulting in an improved manuscript that will have a broad audience and is likely to be highly cited. Below are some suggestions for minor edits.

Minor correction that is strongly suggested to enhance accuracy

Line 123. Change "anterior nares" to "nasopharyngeal" because the pH 7, 34°C condition more closely simulates conditions in the nasopharynx or posterior nasal passages. The temperature at the anterior nares close room/ambient temperature whereas it is 34°C in the posterior nasal passage/nasopharynx (Keck et al. Laryngoscope, 2000; PMID:10764013).

Minor corrections to consider

Line 105. "might be a sporadic but consistent trait"... Since the epifadin BGC is on the mobile genetic element pIVK83, finding it in a small number of strains of diverse STs suggests it has been transferred sporadically across the *S. epidermidis* phylogeny, as well as to some *S. saccharolyticus* strains. Not sure this interesting MGE-mediate transfer is equated as a consistent trait.

Figure 5b. Of note, the methicillin-resistant *S. aureus* strain JE2 was derived from the

methicillin-resistant *S. aureus* strain LAC by curing the latter of two plasmids, p01 and p03 (Fey et al mBio 2013), so these are very highly similar strains. Recommend mentioning this to avoid the impression that they are distinctly different strains of *S. aureus*.

Line 235, Please indicate the strain of *S. aureus* USA300 used, e.g., *S. aureus* USA300 JE2 or *S. aureus* JE2.

Line 263. Would changing this line to read "the median number of *S. aureus* cells was significantly reduced..." more accurately reflects that sometimes the number was little reduced and other times greatly reduced, which might be the case based on the ratio shown.

Reviewer #2 (Remarks to the Author):

The revised manuscript is substantially improved and the new data arising from co-culture of epifadin-producing *S. epidermidis* IVK83 with a susceptible *S. aureus* to identify a mutation in *desK* from the DesKR two component system in increased resistance to epifadin is a significant addition. I look forward to future work on the role of DesKR in epifadin resistance and indeed the function(s) of the DesKR regulon. The inclusion of new analysis on the prevalence of the epifadin BGC in other strains of *S. epidermidis*, CoNS and other bacteria is also an important addition.

Reviewer #3 (Remarks to the Author):

While the authors have adequately addressed most of the reviewers' requests, unfortunately, the revised manuscript still suffers from an incomplete structure elucidation of epifadin. As it stands, this part of the paper does not meet the high technical requirements of a Nature journal, and it would not be publishable in any reputed Chemistry journal, either.

Specifically, solid experimental support of the proposed tetramic acid moiety is missing. The authors propose the tetramate structure solely based on MS/MS data and AntiSmash prediction, which is not at all sufficient. In our earlier evaluation, we have requested compelling 2D NMR data, but the authors still do not show any tetramate-related NMR signals.

On page 74, line 1241, the authors stated that "MS analyses delivered a characteristic signal pattern for the tetramic acid moiety". However, no references are given. Even so, fragmentation patterns are not sufficient to deduce the molecular structure of the presumed heterocycle.

Re: page 74, line 1246, ("Unfortunately, ...") It is clear that tetramates are prone to tautomerization, thus hampering clear NMR assignments. However, there are various ways to prevent tautomerization, e.g. by methylation.

Overall, without rigorous ^1H and ^{13}C NMR chemical shift assignments of the assumed tetramate moiety, it is not appropriate to claim a full structure elucidation.

Again, while this journal is not chemical journal, the structure of epifadin is central to this study (which is even emphasized in the title). The accepted standards of structure elucidation must be met. It is not acceptable to circumvent this and imply that structure elucidation does not require NMR assignment.

Apart from this sticking point regarding the structural elucidation, the authors should be commended for the manner in which they addressed the remaining comments. All queries were answered thoroughly and thoughtfully, including further convincing experimental data where requested. It should be recognized that as a result of this effort, the manuscript has been greatly improved.

On another note, the authors should comment on the iteration (or 2 x Phe) of the first module in Fig. 1.

Reviewer #4 (Remarks to the Author):

The authors have nicely addressed most of the points raised.

However, the revised manuscript still suffers from some flaws in the structure elucidation of epifadin. The paper would currently not be accepted in a major chemistry journal. 2D NMR data for the tetramate should be provided. Otherwise the structure cannot be claimed as fully elucidated. As there are chemical ways to prevent tautomerization this reviewer would expect that these data are included for publication in a nature journal, even for Nature Microbiology, as the structure represents a major finding. If the authors cannot present such data the wording and the title should at least be changed correspondingly and the structure should be labelled as a likely suggestion.

Author Rebuttal, first revision:

We resubmit the manuscript with extensive additional experimental data, which we generated and analyzed during the last months. In this revised version we also addressed all the concerns raised by reviewers as outlined below in our point-by-point response:

Summary for the reviewers:

Based on new experimental work we present

45A) obtained NMR-characteristics of especially the tetramic acid moiety of epifadin and the

B) assigned HR- MS-MS signals, for the first time to tetramate structures by us as well as

C) chemical reaction behaviour for derivatization of the 5-membered ring tetramic acid as requested by the reviewers.

We add four important full 2D-NMR-spectra and - for the convenience of the readership - selected expansions of the key correlation signals to the supporting information.

We now present an unambiguous structure elucidation of the epifadin molecule (**1**) with physicochemical data from analyses, as far as the amphiphilic molecule delivers them. In detail, structure assignment was achieved with the different methods of organic chemistry based essentially on analyses of epifadin's tetrapeptide (also L-D-L-D-stereochemistry from total synthesis), polyene (also with 1D-, 2D-NMR-/ROESY-data), and free NH-aspartate tetramic acid (key data from 1D-, 2D-NMR experiments, HR-MS, MS-MS). We now even achieved some support from the chemical synthesis towards the tetramate of epifadin (data not in this manuscript). Additionally, the structure of **1** is comprehensively supported by genetic information and comparison with the confirmed structure of malonomycin (thanks to fruitful discussions with leading experts Tilmann Weber, Copenhagen, and Jörn Piel, Zurich).

Reviewer #1 (Remarks to the Author):

In this revised manuscript, the authors have addressed the majority of the review comments well resulting in an improved manuscript that will have a broad audience and is likely to be highly cited. Below are some suggestions for minor edits.

RE: We are grateful for this positive reply.

Minor correction that is strongly suggested to enhance accuracy

Line 123. Change "anterior nares" to "nasopharyngeal" because the pH 7, 34°C condition more closely simulates conditions in the nasopharynx or posterior nasal passages. The temperature at the anterior nares close room/ambient temperature whereas it is 34°C in the posterior nasal passage/nasopharynx (Keck et al. Laryngoscope, 2000; PMID:10764013).

RE: We have changed the term to "nasopharyngeal".

Minor corrections to consider

46Line 105. “might be a sporadic but consistent trait”... Since the epifadin BGC is on the mobile genetic element pIVK83, finding it in a small number of strains of diverse STs suggests it has been transferred sporadically across the *S. epidermidis* phylogeny, as well as to some *S. saccharolyticus* strains. Not sure this interesting MGE-mediate transfer is equated as a consistent trait.

RE: Because of the wide geographic distribution of the epifadin-producing isolates we consider the production of epifadin as a regularly, albeit sporadically found trait. We changed the wording to “might be a sporadic but regularly found trait”...

Figure 5b. Of note, the methicillin-resistant *S. aureus* strain JE2 was derived from the methicillin-resistant *S. aureus* strain LAC by curing the latter of two plasmids, p01 and p03 (Fey et al mBio 2013), so these are very highly similar strains. Recommend mentioning this to avoid the impression that they are distinctly different strains of *S. aureus*.

RE: We clarified this point in the legend of Fig. 5b: by adding “*S. aureus* JE2 is a plasmid-cured descendant of *S. aureus* USA300 LAC.”

Line 235, Please indicate the strain of *S. aureus* USA300 used, e.g., *S. aureus* USA300 JE2 or *S. aureus* JE2.

RE: We indicated that we used strain JE2.

Line 263. Would changing this line to read “the median number of *S. aureus* cells was significantly reduced...” more accurately reflects that sometimes the number was little reduced and other times greatly reduced, which might be the case based on the ratio shown.

RE: We have changed the sentence as suggested.

Reviewer #2 (Remarks to the Author):

The revised manuscript is substantially improved and the new data arising from co-culture of epifadin-producing *S. epidermidis* IVK83 with a susceptible *S. aureus* to identify a mutation in *desK* from the *DesKR* two component system in increased resistance to epifadin is a significant addition. I look forward to future work on the role of *DesKR* in epifadin resistance and indeed the function(s) of the *DesKR* regulon. The inclusion of new analysis on the prevalence of the epifadin BGC in other strains of *S. epidermidis*, *CoNS* and other bacteria is also an important addition.

RE: We are grateful for this positive reply.

Reviewer #3 (Remarks to the Author):

While the authors have adequately addressed most of the reviewers' requests, unfortunately, the revised manuscript still suffers from an incomplete structure elucidation of epifadin. As it stands, this part of the paper does not meet the high technical requirements of a Nature journal, and it would not be publishable in any reputed Chemistry journal, either.

Specifically, solid experimental support of the proposed tetramic acid moiety is missing. The authors propose the tetramate structure solely based on MS/MS data and AntiSmash prediction, which is not at all sufficient. In our earlier evaluation, we have requested compelling 2D NMR data, but the authors still do not show any tetramate-related NMR signals.

RE: We are sorry that the presented data in our previous submission did not cover all the details of the tetramic acid moiety, which was a consequence of the extraordinary instability and complexity of the molecule. We put enormous efforts into more detailed 2D-NMR data, which are based on several NMR experiments of the isolated natural compound from multiple cultivation and isolation experiments (new Suppl. Figures 3-4 and new Suppl. Information).

We have now added a structural scheme of epifadin (**1**) with the 2D-NMR data (chemical shifts, HMBC, HSQC, assigned with arrows), gained from an additional series of natural product isolation and NMR studies to the supporting information.

Supplementary Figure 3A: Correlations of 2D-NMR experiments of epifadin (**1**) with focus on its tetramic acid.

Here, we are convinced, that the tetramic acid (TA) of epifadin is an unprecedented structural moiety in an exceptional structural environment, which delivers the unusual character of epifadin (**1**) in the NMR experiments. The unique epifadin tetramic acid consists of a combination of the (1.) aspartic acid residue, (2.) the unprotected NH-amide within the tetramic heterocycle, and the (3.) amphiphilic long polyene-tetrapeptide chain.

Hitherto, not all NMR-signals and therefore chemical shifts of the tetramic acid of epifadin could be observed. The missing NMR-signals are a typical tetramic acid phenomenon, e.g., militarinone C (**2**)/ Brückner group (doi.org/10.1021/acs.orglett.0c00431), pyranonigrin J (**3**) and I (**4**)/ Brückner group (doi.org/10.1002/ejoc.202101053) do also not show the respective C-3, C-4, C-8. However, our epifadin molecule luckily delivers key ^1H -, ^{13}C -NMR assignments of its TA but also reproducibly lacks distinct NMR signals, which often occurs for tetramates. (See discussion of more NMR-details below.)

With the new 2D-NMR data we are confident to provide sufficient evidence for the presence of a tetramic acid moiety in epifadin as outlined now in the revised manuscript (lines 182-191).

New text in the manuscript: “Typical for tetramic acids, 2D-NMR-spectroscopy yielded solid key correlations for characteristic tetramate signals in epifadin. However, some distinct signals of the tetramic heterocycle were not detected, probably as a consequence of the unique assembly of the amphiphilic tetrapeptide-polyene-(Asp)tetramate (Supplementary Figs. 3–4). Chemical derivatization attempts did not provide stable epifadin tautomers (Supplementary Figs. 5-7). The tetramic acid moiety is further supported by the predicted adenylation domain specificity of A₅ (Asp) and the presence of a particular C-terminal condensation-like (C_T) domain in EfiE, which is involved in cyclization reactions²⁹. Such a domain has been proposed to be responsible for the tetramic acid formation in the bacterial antibiotic malonomycin³⁰ and in fungal PKS-NRPS derived products such as cAATrp³¹.”

On page 74, line 1241, the authors stated that “MS analyses delivered a characteristic signal pattern for the tetramic acid moiety”. However, no references are given. Even so, fragmentation patterns are not sufficient to deduce the molecular structure of the presumed heterocycle.

RE: We are sorry for the misleading word “characteristic”, since there is no published reference for properly assigned MS-fragments of tetramic acids, to the best of our knowledge. We have changed the sentence to “MS analyses with MS-MS fragmentation yielded a comprehensive mass signal pattern, which we assigned to distinct fragment ions. We state that these ion structures originate from a 5-membered tetramate heterocycle (e.g., [M+H]⁺, *m/z* 158.0454, assigned to C₆H₈NO₄⁺ (*m/z* 158.0448); [M+H]⁺, *m/z* 140.0348, for a C₆H₆NO₃⁺ moiety (*m/z* 140.0342) (Fig. 4a; Extended Data Fig.5; Supplementary Table S1).”

Re: page 74, line 1246, (“Unfortunately, ...”) It is clear that tetramates are prone to tautomerization, thus hampering clear NMR assignments. However, there are various ways to prevent tautomerization, e.g. by methylation.

Overall, without rigorous ¹H and ¹³C NMR chemical shift assignments of the assumed tetramate moiety, it is not appropriate to claim a full structure elucidation.

RE: Unfortunately, different attempts of chemical derivatization aiming at methylation, e.g. TMS-CHN₂, methyl iodide, or acetylation, e.g. acetic acid anhydride/ pyridine, did not yield the desired tetramic acid derivatives of neither reutericyclin (**5**) as a model compound nor of epifadin (**1**). However, methylation reactions of the 6-membered pyridone ring in kirromycin (**6**) readily yielded the mono- and dimethylated kirromycin derivatives in our hands (successfully proven by HPLC-UV-(+) ESI-mass spectrometry).

With only the natural product epifadin (1) for the NMR-experiments at hand, we tackled the chemical total synthesis of further fragments of epifadin (1) (and, as a future goal, especially the synthesis of non-natural chemical analogues with higher chemical stability!). Current result:

In our hands, the synthesized, still fully protected synthetic aspartate-tetramic acid (7) shows clear NMR ¹H and ¹³C NMR-signals (deprotection experiments did not result in the 'free-NH-tetramic acid' and are subject to future work. Currently the deprotection regime, and Goldberg-polyene fragment connection are studied. Notably, we believe, that the currently running experiments of the total chemical synthesis of epifadin will further support the here given structure (data not shown in the manuscript).

Tetramic acid fragment towards epifadin from our total chemical synthesis

Again, while this journal is not chemical journal, the structure of epifadin is central to this study (which is even emphasized in the title). The accepted standards of structure elucidation must be met. It is not acceptable to circumvent this and imply that structure elucidation does not require NMR assignment.

RE: The structure of epifadin is indeed an important part of our manuscript and we are confident that our additional experimental data confirm the presented structure. Nevertheless, a major focus are the unique properties of epifadin. This novel type of microbiome-derived antimicrobial can be regarded as a new paradigm of antagonistic interactions in human microbiomes. Since we agree that the manuscript profits from more NMR evidence, we added the solid 2D-NMR data set (see above).

Apart from this sticking point regarding the structural elucidation, the authors should be commended for the manner in which they addressed the remaining comments. All queries were answered thoroughly and thoughtfully, including further convincing experimental data where requested. It should be recognized that as a result of this effort, the manuscript has been greatly improved.

RE: We are grateful for this positive reply.

On another note, the authors should comment on the iteration (or 2 x Phe) of the first module in Fig. 1.

RE: We have included the following information in the legend of Figure 1: “in which the first A-domain is responsible for the iterative integration of two phenylalanine residues in L- and D-conformation.”

Reviewer #4 (Remarks to the Author):

The authors have nicely addressed most of the points raised.

However, the revised manuscript still suffers from some flaws in the structure elucidation of epifadin. The paper would currently not be accepted in a major chemistry journal. 2D NMR data for the tetramate should be provided. Otherwise the structure cannot be claimed as fully elucidated. As there are chemical ways to prevent tautomerization this reviewer would expect that these data are included for publication in a nature journal, even for Nature Microbiology, as the structure represents a major finding. If the authors cannot present such data the wording and the title should at least be changed correspondingly and the structure should be labelled as a likely suggestion.

RE: We are thankful for this comment. As explained in response to Reviewer #3 (see above), we here provide additional 2D-NMR data strongly supporting the tetramic acid structure. For a full explanation of our additional data and comparison with tetramic acids from literature see the information above, below, and the revised manuscript with its Supplementary Information.

Appendix with more NMR-details

+ With only the unmodified natural product at hand, we state, that we face unusual chemical shifts in all of our NMR experiments performed over the years (2019-2023) compared to the rare literature-cited tetramic acids with an aspartic acid. With respect to the amphiphilic peptide-polyene-tetramic acid, we chose the different solvents and temperatures for NMR studies, since we aimed to achieve one single set of signals (in combinations: DMSO- d_6 , CD_3OD , dimethylformamide, CH_3CN , $CDCl_3$; with or w/o trifluoroacetic acid, formic acid, Zn^{2+} -ions).

+ Only one recent publication shows a quite similar polyene-'free NH'-aspartate-tetramic acid **8**. The fungal metabolite MCA 17-1 (**8**) (Yi Zang and Xu-Ming Mao) obviously delivers NMR spectra with the rare observation of one set of signals in DMSO (doi:[10.1021/acs.jafc.1c03639](https://doi.org/10.1021/acs.jafc.1c03639)). It appears that the challenging NMR studies of epifadin (**1**) can be attributed to its complex molecular structure, specifically its amphiphilic character and the unique combination of polar/non-polar tetrapeptide, long polyene, and 'free NH'-aspartate tetramic acid.

+ Hitherto, the molecule has not been stable or soluble enough to obtain one clear set of all the epifadin (**1**) signals in 1D- and 2D-NMR-spectra. We hypothesize, that the unusual high-field shifts, especially of C-3 and C-6 of TA, occur presumably due to strong atomic interaction with pi-electrons of the polyene and aspirated-C=O (due to models).

+ Additional full 1H - ^{13}C -NMR-HMBC spectra and figures of selected correlations are presented in the Supporting Information in addition to the already given NMR-figures and data.

Decision Letter, second revision:

Message: Our ref: NMICROBIOL-22020429B

5311th August 2023

Dear Dr. Krismer,

Thank you for submitting your revised manuscript "The extremely short-lived peptide-polyene antimicrobial epifadin enables nasal commensal to eliminate *Staphylococcus aureus*" (NMICROBIOL-22020429B). It has now been seen by the original referees and their comments are below. The reviewers find that the paper has improved in revision, and therefore we'll be happy in principle to publish it in *Nature Microbiology*, pending minor revisions to satisfy the referees' final requests and to comply with our editorial and formatting guidelines.

We are now performing detailed checks on your paper and will send you a checklist detailing our editorial and formatting requirements in about two weeks. Please do not upload the final materials and make any revisions until you receive this additional information from us.

Thank you again for your interest in *Nature Microbiology*. Please do not hesitate to contact me if you have any questions.

Sincerely,

Emily

Emily White, PhD
Locum Chief Editor
Nature Microbiology

4 Crinan Street, London, UK, N1 9XW
+44 207 418 5601
emily.white@nature.com

orcid.org/0000-0002-2314-5718

Reviewer #3 (Remarks to the Author):

The authors have addressed the reviewer comments and managed to provide additional support for the proposed structure of epifadin. I am just surprised that the tetramic acid structure is drawn with three carbonyls. Typically, one of these is in the enol form (conjugation). Please also adhere to the IUPAC standards and put the D/L discriminators in small capitals. Overall, this is a nice piece of work that is now suitable for publication.

54Reviewer #4 (Remarks to the Author):

The authors have done a great job in addressing the remaining points raised by the reviewers. With respect to structure elucidation the provided interpretation of the data is sound and deserves publication in conjunction with this very nice piece of work. Of course final prove of structural assignment can only be achieved by comparison to a synthetic product (which is clearly beyond the limits of this study).

Final Decision Letter:

Message 3rd November 2023

:
Dear Bernhard,

I am pleased to accept your Article "Commensal production of a broad spectrum and short-lived antimicrobial peptide polyene eliminates nasal *Staphylococcus aureus*" for publication in *Nature Microbiology*. Thank you for having chosen to submit your work to us and many congratulations.

Over the next few weeks, your paper will be copyedited to ensure that it conforms to *Nature Microbiology* style. We look particularly carefully at the titles of all papers to ensure that they are relatively brief and understandable.

Acceptance of your manuscript is conditional on all authors' agreement with our publication policies (see <https://www.nature.com/nmicrobiol/editorial-policies>). In particular your manuscript must not be published elsewhere and there must be no announcement of the work to any media outlet until the publication date (the day on which it is uploaded onto our website).

55Please note that *Nature Microbiology* is a Transformative Journal (TJ). Authors may publish their research with us through the traditional subscription access route or make their paper immediately open access through payment of an article-processing charge (APC). Authors will not be required to make a final decision about access to their article until it has been accepted. [Find out more about Transformative Journals](https://www.springernature.com/gp/open-research/transformative-journals)

Authors may need to take specific actions to achieve [compliance](https://www.springernature.com/gp/open-research/funding/policy-compliance-faqs) with funder and institutional open access mandates. If your research is supported by a funder that requires immediate open access (e.g. according to [Plan S principles](https://www.springernature.com/gp/open-research/plan-s-compliance)) then you should select the gold OA route, and we will direct you to the compliant route where possible. For authors selecting the subscription publication route, the journal's standard licensing terms will need to be accepted, including [self-archiving policies](https://www.nature.com/nature-portfolio/editorial-policies/self-archiving-and-license-to-publish). Those licensing terms will supersede any other terms that the author or any third party may assert apply to any version of the manuscript.

To assist our authors in disseminating their research to the broader community, our SharedIt initiative provides you with a unique shareable link that will allow anyone (with or without a subscription) to read the published article. Recipients of the link with a

subscription will also be able to download and print the PDF.

With kind regards,

[redacted]